# MECHANISTIC MODE CONNECTIVITY

## ABSTRACT

Neural networks are known to be biased towards learning mechanisms that help identify *spurious attributes*, yielding features that do not generalize well under distribution shifts. To understand and address this limitation, we study the geometry of neural network loss landscapes through the lens of *mode connectivity*, the observation that minimizers of neural networks are connected via simple paths of low loss. Our work addresses two questions: (i) do minimizers that encode dissimilar mechanisms connect via simple paths of low loss? (ii) can fine-tuning a pretrained model help switch between such minimizers? We define a notion of *mechanistic similarity* and demonstrate that lack of linear connectivity between two minimizers implies the corresponding models use dissimilar mechanisms for making their predictions. This property helps us demonstrate that naïve fine-tuning can fail to eliminate a model's reliance on spurious attributes. We thus propose a method for altering a model's mechanisms, named *connectivity-based fine-tuning*, and validate its usefulness by inducing models invariant to spurious attributes.

## 1 INTRODUCTION

Deep neural networks (DNNs) suffer from various robustness problems, learning representations that fail to generalize well beyond the given training distribution (D'Amour et al., 2020; Teney et al., 2022; Geirhos et al., 2020; Recht et al., 2019; Taori et al., 2020; Jacobsen et al., 2018). This lack of robustness is generally a consequence of models learning mechanisms that rely on *spurious* attributes in the training data for making their predictions. Such attributes–even if not perfectly predictive–tend to be *simpler* to represent according to the model's inductive biases (Nakkiran et al., 2019; Valle-Perez et al., 2018; Hu et al., 2020; Shah et al., 2020) and commonly emerge due to sampling biases and hidden confounders in static datasets (Kaur et al., 2022; Lee et al., 2022). For example, in most vision datasets, backgrounds are correlated with object categories–a sampling bias (Beery et al., 2018; Xiao et al., 2020). Consequently, a model can learn to predict the correct category of an object by learning mechanisms to identify either its background or its shape; however, only models that rely on shape are likely to generalize robustly (Geirhos et al., 2018; Dittadi et al., 2020). Indeed, Scimeca et al. (2021); Hermann & Lampinen (2020) show that using different datasets for a task, standard training pipelines can induce models that use *entirely distinct* mechanisms for making their predictions, performing equally well in-distribution, but vastly differently out-of-distribution. Recent works on improving neural networks robustness thus advocate a need for modeling the *causal mechanisms* underlying the data-generating process (Arjovsky et al., 2019; Krueger et al., 2021; Lu et al., 2021), promoting representations *invariant* to spurious attributes.

**This Work:** In this paper, we introduce the idea of *mechanistic similarity* (Sec. 3) to assess whether two models rely on the same input attributes for making their predictions. Specifically, we call two models mechanistically similar if they exhibit invariance to the same attributes of an input, but may otherwise produce different representations for it. Our motivating question is whether a model can be fine-tuned to alter its mechanisms, i.e., to learn different invariances; we call this the problem of *mechanistic fine-tuning* (Sec. 5). For instance, if a model has learned to rely on a spurious attribute in its training data, can that reliance be eliminated by training it on a minimal set of "clean" samples that do not contain the spurious attribute? This problem is of practical value because curating a large, clean dataset and training from scratch on it in the first place can be expensive.

We consider the problem above through the lens of *mode connectivity* in neural networks, which refers to the phenomenon that neural network minimizers identified via training on the *same* dataset for a task tend to be connected via relatively simple paths of low loss in the model's loss landscape (e.g.,

linear or quadratic splines) (Garipov et al., 2018; Draxler et al., 2018; Frankle et al., 2020; Entezari et al., 2021; Kuditipudi et al., 2019; Nguyen et al., 2021). In particular, we analyze connectivity of *mechanistically dissimilar* models that are induced via training on *different* datasets for a task and can hence learn to rely on *entirely distinct* attributes of an input to make their predictions (see Fig. 1).

We induce such mechanistically dissimilar models by embedding synthetic cues in existing datasets (see Fig. 2) and training models on these datasets under different proportions of samples with spurious attributes (see Fig. 4); given easily separable cues, such manipulated data allows training of models that learn mechanisms to preferentially identify the synthetic cue over natural data attributes (Shah et al., 2020). Our extensive analysis (Sec. 4) shows that *if two models lack linear connectivity in the landscape, they must be mechanistically dissimilar*; that is, increase in loss as we linearly move between two models implies they have learned different invariances. This result holds implications for naïve fine-tuning of pretrained networks, which often yields models linearly connected with the original pretraining minimizer (Neyshabur et al., 2020) and can hence be insufficient for altering a model's prediction mechanisms (Fig. 5). We thus propose a technique, named *Connectivity-Based Fine-Tuning (CBFT)*, that exploits lack of linear connectivity between mechanistically dissimilar models to induce networks that follow our desired mechanisms (Sec. 5). Our extensive experimental results show CBFT is more effective at reducing a model's sensitivity to spurious attributes than recent techniques (Kirichenko et al., 2022b; Kumar et al., 2022).

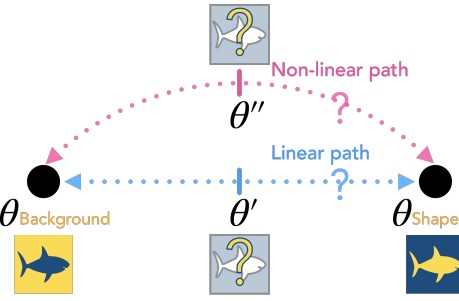

Figure 1: **Mechanistic Lens on Mode connectivity.** Consider two sets of parameters that minimize loss using background $\theta_{\text{background}}$ and object shape $\theta_{\text{shape}}$ as the input attributes for prediction, respectively. Are such *mechanistically dissimilar* minimizers connected via paths of low loss in the landscape? Does the dissimilarity of these mechanisms affect the simplicity of their connectivity paths? Can we exploit this connectivity to switch between minimizers that use our desired mechanisms?

## 2 PRELIMINARIES

**Notations.** Consider a neural network $f : \mathbb{R}^n \times \mathbb{R}^d \rightarrow [K]$ that takes $n$-dimensional inputs $x \in \mathcal{X} \subset \mathbb{R}^n$, has parameters $\theta \in \mathbb{R}^d$, and produces an output $f(x; \theta) \in [K]$. We say $\theta$ "induces the model" $f(.; \theta)$. Loss on a dataset $\mathcal{D} \in \mathcal{X} \times [K]$ for parameters $\theta$ is denoted using a non-negative function $\mathcal{L}(f(\mathcal{D}; \theta))$ and $\theta$ is called a minimizer for that dataset if $\mathcal{L}(f(\mathcal{D}; \theta)) < \epsilon$, where $\epsilon$ is some small scalar. In the following, our focus will be minimizers retrieved by running SGD on a dataset's loss. Assume there is a latent space $\mathcal{Z} \subset \mathbb{R}^m$ with $z$ sampled from some factorial distribution, $P(z) = \prod_i P(z_i)$. These latent variables are assumed to generate samples in the dataset via $\mathcal{G} : \mathcal{Z} \rightarrow \mathcal{X} \times [K]$; $(x, y) := \mathcal{G}(z)$, with $x$ and $y$ conditionally independent given $z$. If $\mathcal{G}_X, \mathcal{G}_Y$ define the components of $\mathcal{G}$ producing $x$ and $y$, we assume $\mathcal{G}_X(.)$ has a valid left-inverse $\mathcal{G}_X^{-1} : \mathcal{X} \rightarrow \mathcal{Z}$, i.e., $x$ contains all of the information necessary to recover the true settings of the (independent) latent variables, $z$. These assumptions are standard in literature on disentanglement (Locatello et al., 2019; 2020; Gresele et al., 2020; 2021; Von Kügelgen et al., 2021) and Independent Component Analysis (ICA) (Hyvarinen & Morioka, 2016; 2017; Khemakhem et al., 2020; 2021).

### 2.1 MODE CONNECTIVITY

We denote a continuous path between two sets of parameters $\theta_1, \theta_2$ as $\gamma_{\theta_1 \rightarrow \theta_2}(t)$, where $\gamma_{\theta_1 \rightarrow \theta_2}(0) = \theta_1$ and $\gamma_{\theta_1 \rightarrow \theta_2}(1) = \theta_2$. At times, we say a path lacks *loss barriers* on a dataset, if moving along the path never yields increase in loss on that dataset. Using the above notations, we now formalize the notion of mode connectivity, in line with previous work (Garipov et al., 2018; Draxler et al., 2018).

**Definition 1.** *(Mode Connectivity Along a Path.) Minimizers $\theta_1, \theta_2$ of loss $\mathcal{L}(f(\mathcal{D}; \theta))$ on a dataset $\mathcal{D}$ are called mode connected along the path $\gamma_{\theta_1 \rightarrow \theta_2}(t)$ if moving along the path never increases loss. Formally, $\forall t \in [0, 1], \mathcal{L}(f(\mathcal{D}, \gamma_{\theta_1 \rightarrow \theta_2}(t))) \leq \min\{\mathcal{L}(f(\mathcal{D}; \theta_1)), \mathcal{L}(f(\mathcal{D}; \theta_1))\}.$*

As mentioned in Sec. 1, prior work has shown that minimizers of modern neural networks exhibit mode connectivity along rather simple paths in the landscape. This property was first illustrated in

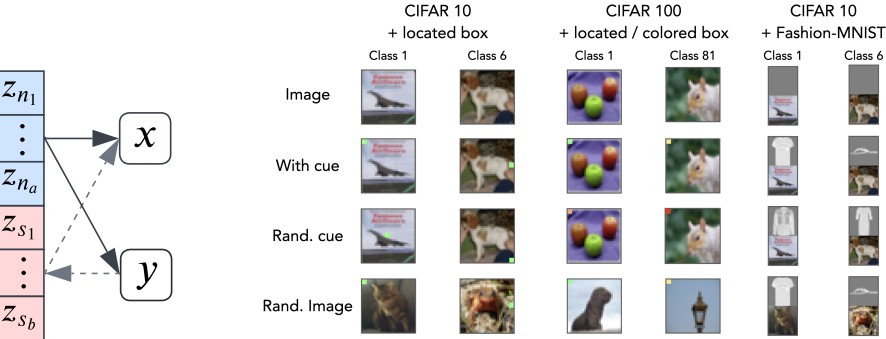

Figure 2: **Data-Generating Process (left).** We augment the natural latents $\{z_n\}$ of a data-generating process with a set of synthetic latents $\{z_s\}$. The attributes induced in the input by these synthetic latents are called *cues*. Conditioning (grey, dotted line) the value of a synthetic latent on the target label ($y$), we can induce correlation between its corresponding cue and the desired model output. If the cue is made easily separable, a neural network will preferentially learn mechanisms to use the cue for making its predictions (Shah et al., 2020; Sagawa et al., 2020). This allows us to controllably embed spurious attributes in the dataset. **Synthetic Datasets (right).** Following the protocol above, we embed synthetic cues (denoted *With cue*) in three existing dataset: (1) CIFAR-10 with $3 \times 3$ box cues whose locations depend on the target label; (2) CIFAR-100 with $3 \times 3$ box cues colored according to the first digit of the object label, and located according to the second digit; and (3) Dominoes (Shah et al., 2020), where CIFAR-10 images are concatenated with Fashion-MNIST images of the same class. We analyze counterfactual datasets that involve randomizing the cue (denoted *Rand. cue*) or natural image (denoted *Rand. image*) to break their correlation with the target label. This helps us ascertain the extent to which the model's prediction relies on natural vs. spurious attributes.

concurrent works by Garipov et al. (2018) and Draxler et al. (2018) and has been further expanded by Frankle et al. (2020); Entezari et al. (2021); Benton et al. (2021); Wortsman et al. (2021). Correspondingly, we will restrict our experiments to the following two forms of paths:

$$\text{Linear: } \gamma_{\theta_1 \rightarrow \theta_2}(t) = (1-t)\theta_1 + t\theta_2, \quad \text{and} \tag{1}$$

$$\text{Quadratic: } \gamma_{\theta_1 \rightarrow \theta_2}(t) = (1-t)^2\theta_1 + 2t(1-t)\theta_{12} + t^2\theta_2, \tag{2}$$

where $\theta_{12}$ denotes a set of parameters that is explicitly optimized to identify a quadratic path connecting the two minimizers $\theta_1$ and $\theta_2$ for that dataset. Thus, quadratic paths are a function of the dataset used for their identification. We also note that recent work hypothesizes two linearly disconnected minimizers of a dataset can be linearly connected given an appropriate permutation of neurons (Entezari et al., 2021); specifically, *one that maintains the model outputs, but aligns them in layer-wise activations/weights*. To demonstrate the robustness of our claims, we follow the alignment method by Singh & Jaggi (2020); Ainsworth et al. (2022) and include experiments on models that have been permuted to match in activations (see App. B.2). We call these paths *Linear (permuted)*.

## 2.2 EXPERIMENTAL SETUP

Since our goal is to assess the role of mechanistic similarity on connectivity of pairs of minimizers, we must design models that use different mechanisms for making predictions. To this end, we propose to use easily manipulable synthetic datasets. Such datasets have been used by prior works for better understanding several important topics, such as transfer learning (Dittadi et al., 2020), domain generalization (Wiles et al., 2021; Van Steenkiste et al., 2019), disentanglement (Higgins et al., 2017; Klindt et al., 2020), self- and semi-supervised learning (Von Kügelgen et al., 2021; Trivedi et al., 2022; Locatello et al., 2020), and inductive biases of neural networks (Hermann et al., 2020; Hermann & Lampinen, 2020; Ritter et al., 2017). Our data-generating process is illustrated in Fig. 2 and involves augmenting the natural latents of the process with synthetic ones that are conditioned on the target label. We refer to the attributes induced in the input by such latents as *cues*. We intentionally choose cues that are easily separable to cause the model to prefer them over natural attributes for making its predictions (Shah et al., 2020). Such low-complexity cues can be viewed as stand-ins for spurious or shortcut attributes that are commonplace in realistic settings (Beery et al., 2018; Geirhos et al., 2020), allowing us to determine whether minimizers that induce models reliant on either spurious or

non-spurious attributes exhibit mode connectivity. Training curves for different models are shown in App. A and clearly demonstrate that models trained without cues entirely ignore them for making their predictions; meanwhile, under perfect correlation, the model completely ignores the natural data and relies entirely on cues. Note that the data-generating process is easy to intervene on for creating counterfactuals (see Def. 2). Specifically, we focus on interventions that break a cue's correlation with the target label by randomizing it (e.g., uniformly changing location of the box in the CIFAR-10 with box cue dataset). Vice-versa, we also analyze interventions that keep the cue intact, but randomize the underlying image (e.g., putting a cat instead of a dog). This helps us assess how much a model relies on natural attributes that originate in the source image vs. our synthetically embedded cues.

## 3 DEFINING MECHANISTIC SIMILARITY

While mode connectivity via simple paths is a surprising result that indicates the loss landscape of modern neural networks is relatively benign, prior work has not addressed whether models that rely on different mechanisms for making their predictions exhibit mode connectivity as well (see Fig. 1). To analyze this question, we must first define a notion of mechanistic similarity between two models. Our definition is motivated by the following question: *do models that succeed in a similar manner, fail in a similar manner?* We propose to assess failures by measuring a model's response to input transformations: if two models use similar mechanisms, they should respond similarly to transformed inputs. This is analogous to the use of visual illusions (invalid perceptual inferences) for designing models of early visual processing in neuroscience and cognitive science (Tanaka et al., 2019; Marr, 1976). By choosing transformations that embody task-relevant vulnerabilities, we can make this definition operationally well-motivated. For example, by randomizing the synthetic cue in Fig. 2, we can assess whether a model relies on a

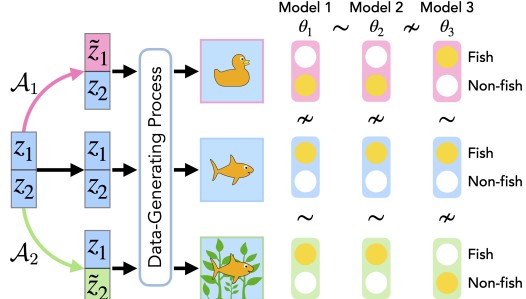

Figure 3: **Mechanistic Similarity:** We define mechanistic similarity of two models based on how they respond to unit interventions on the data-generating process, i.e., interventions on specific dimensions of the latent vector $z$; e.g., $\mathcal{A}_1$ (shape) and $\mathcal{A}_2$ (background) in the figure. Here, yellow circles represent the prediction of a given model (column) on a counterfactual image (row), i.e., $f(\mathcal{E}(x; \mathcal{A}_i); \theta_j)$. Models whose predictions are invariant to the same set of interventions (denoted $\theta_1 \sim \theta_2$) are termed mechanistically similar.

spurious attribute for making its predictions. We next formalize the intuition introduced above.

**Definition 2.** *(Unit Interventions on Data-Generating Process and Counterfactuals.) An isomorphism $\mathcal{A}_i^{\alpha_i} : \mathcal{Z}_i \times \mathcal{Z}_i \rightarrow \mathcal{Z}_i$ defines a unit intervention on the $i^{th}$ dimension of the state $z$ if it alters its value by setting it to a scalar value $\alpha_i \in \mathcal{Z}_i$. The isomorphism $\mathcal{E} : \mathcal{X} \times \mathbb{R}^m \rightarrow \mathcal{X}$ defines a counterfactual if it alters a datapoint $x$ by changing its corresponding latent $z = \mathcal{G}_X^{-1}(x)$ via a set of unit interventions $\widehat{\mathcal{A}} := \{\mathcal{A}_i^{\alpha_i}\}_{i=1}^m$. Specifically, we have $\mathcal{E}(x; \widehat{\mathcal{A}}) = \mathcal{G}_X \circ \mathcal{A}_m^{\alpha_m} \circ \cdots \circ \mathcal{A}_1^{\alpha_1} \circ \mathcal{G}_X^{-1}(x)$.*

Thus, a unit intervention on the data-generating process allows precise manipulation of the latent state $z$, while a counterfactual maps this changed state into the observable data space. Due to independence of latent dimensions, our definition of unit interventions easily composes and can model broader notions of interventions (Schölkopf et al., 2021); combined with counterfactuals, unit interventions are thus sufficient to assess a model's response to general data transformations, as we show next.

**Definition 3.** *(Invariance.) The model $f(.; \theta)$ is termed invariant to unit intervention $\mathcal{A}_i$ if counterfactuals generated by $\mathcal{A}_i$ do not increase its loss, i.e., $\mathcal{L}(f(\mathcal{D}; \theta)) = \mathbb{E}_{\alpha \in \mathcal{Z}_i} \mathcal{L}(f(\mathcal{E}(\mathcal{D}; \mathcal{A}_i^{\alpha}); \theta))$.*

**Proposition 1.** *(Exhaustiveness of Unit Interventions.) If $f(.; \theta)$ is invariant to unit interventions $\mathcal{A}_i$ and $\mathcal{A}_j$, it must be invariant to their composition; conversely, lack of invariance to either $\mathcal{A}_i$ or $\mathcal{A}_j$ precludes invariance to their composition.*

Essentially, the above statement argues studying a model's response to unit interventions is sufficient to characterize which attributes of the data the model is using for prediction: if a model is invariant to a set of unit interventions, it must be invariant to all counterfactuals generated by their composition too; similarly, lack of invariance to a single unit intervention is sufficient to preclude invariance to all counterfactuals produced by the composition of that intervention and a set of invariant interventions. We are now ready to define mechanistic similarity.

**Definition 4.** *(Mechanistic Similarity.) Consider a set of unit interventions $\widehat{\mathcal{A}} := \{\mathcal{A}_i\}$, where $i \in [m]$. For parameters $\theta$, denote the subset of interventions that $f(.; \theta)$ is invariant to as $\mathcal{I}(\theta) \subset \widehat{\mathcal{A}}$. Then, $f(.; \theta_1)$ and $f(.; \theta_2)$ are called mechanistically similar if $I(\theta_1) = I(\theta_2)$.*

Fig. 3 illustrates mechanistic similarity in an intuitive manner. Formally, given a set of independent transformations (instantiated by use of unit interventions), we say two models are mechanistically similar if they exhibit invariance to the same set of interventions. Our definition shares motivation with the idea of *prediction mismatch*, which involves assessing the number of distinct examples two models produce different predictions on, and has been used in prior work to analyze properties such as calibration and catastrophic interference (Hooker et al., 2019; Mania et al., 2019; Maini et al., 2022). In contrast, mechanistic similarity is based on assessment of the number of distinct interventions on the data-generating process to which two models are simultaneously invariant. This makes mechanistic similarity more appropriate for problems involving distribution shifts and robustness, where modeling the data-generating process is of crucial importance (Kaur et al., 2022). We next extend the definition of mode connectivity while accounting for mechanistic similarity.

**Definition 5.** *(Mechanistic Connectivity Along a Path.) Consider two minimizers $\theta_1$ and $\theta_2$ of loss $\mathcal{L}(f(\mathcal{D}; \theta))$ on a dataset $\mathcal{D}$. Let $\mathcal{E}(\mathcal{D}) := \{\mathcal{E}(\mathcal{D}; \mathcal{A}_i^{\alpha_i \sim \mathcal{Z}_i})\}_{i=1}^m$ denote a set of counterfactual datasets designed by applying unit interventions $\mathcal{A}_i$ to all points in dataset $\mathcal{D}$, where intervention assignments $\alpha_i$ are chosen uniformly from the respective range of values $\mathcal{Z}_i$. Then, $\theta_1$ and $\theta_2$ are called mechanistically connected along the path $\gamma_{\theta_1 \to \theta_2}(t)$ if, for all counterfactual datasets, moving along the path does not yield increase in loss; that is, $\theta_1, \theta_2$ exhibit mode connectivity along the same path $\gamma_{\theta_1 \to \theta_2}(t)$ for all counterfactual datasets.*

Essentially, if two minimizers exhibit mechanistic connectivity, then for all pre-defined interventions, there exists a single connectivity path such that moving along it does not yield increase in loss on all counterfactual datasets described by the interventions; that is, all points on the path induce mechanistically similar models. Meanwhile, if two minimizers induce mechanistically dissimilar models, moving along any path between them will necessarily involve a change in the mechanisms used for making predictions. If this change yields increase in loss on a counterfactual dataset, then it is harmful for the distribution shift described by the corresponding intervention. Mechanistic connectivity is designed to succinctly capture this behavior and characterize the connectivities of mechanistically (dis)similar models.

## 4  CONNECTIVITY OF MECHANISTICALLY DISSIMILAR MODELS

Having developed a language to describe the mechanistic similarity of two models, we now characterize connectivity properties exhibited in the landscape by mechanistically dissimilar models. Specifically, we demonstrate that while mechanistically dissimilar models can exhibit connectivity via quadratic paths, they are generally not linearly connected. We start with the following proposition, which follows directly from the results of Nguyen (2019); Simsek et al. (2021), and shows mechanistically dissimilar models can indeed be mode-connected.

**Proposition 2.** *(Mode Connectivity of Mechanistically Dissimilar Models.) Assume $\theta_1, \theta_2$ are minimizers of the loss on a dataset $\mathcal{D}$ and induce mechanistically dissimilar models. Given sufficient overparameterization, there exists a continuous path along which the minimizers are mode connected.*

That is, even if two minimizers of loss on a dataset $\mathcal{D}$ induce models that rely on completely disparate mechanisms, *there necessarily exists a continuous path along which the two minimizers exhibit mode connectivity.* Note, however, the claim does not yet address the simplicity of these connectivity paths, which is empirically observed to be surprisingly high for minimizers retrieved from the same dataset.

To investigate whether this property holds for mechanistically dissimilar models, we train VGG-13 and ResNet-18 models on the synthetic datasets described in Sec. 2.2. We analyze accuracy on counterfactual datasets (see Fig. 2) along quadratic and linear paths (see Eq. 1), including quadratic paths identified using data with/without cue, linear paths, and linear (permuted) paths. Results for ResNet-18 are shown in Fig. 4 and remaining are deferred to App. D. We find that points on the connectivity paths respond differently to counterfactuals, indicating *lack of mechanistic connectivity* (see Def. 5). Simultaneously, we see minimizers that induce mechanistically dissimilar models can be mode connected via relatively simple paths as well: *we can easily identify quadratic, but not linear, mode connectivity paths for two mechanistically dissimilar models.* In fact, we conjecture that lack of linear connectivity between two models is intricately related to their mechanistic similarity.

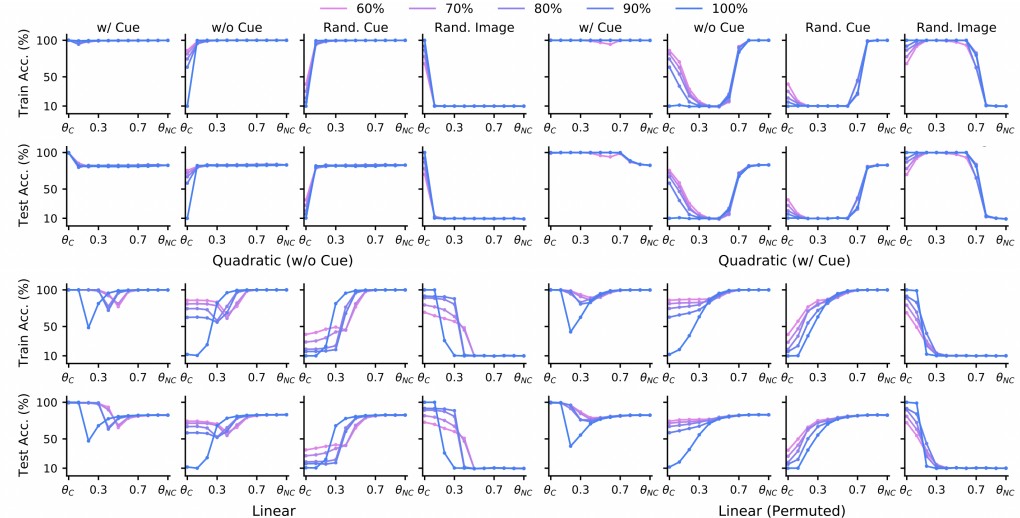

Figure 4: **Are Mechanistically Dissimilar Minimizers Connected?** We train ResNet-18 models on our synthetic CIFAR-10 datasets with and without box-cues (denoted $\theta_C$ and $\theta_{NC}$, respectively). Line colors denote proportion of the training data with synthetic cues. Plot titles denote evaluation data (see Fig. 2), including data where either the cue is present (w/ Cue), absent (w/o Cue), randomized (Rand. Cue), or the underlying image is randomized (Rand. Image). As shown, $\theta_{NC}$ yields the same performance upon randomization of the cue, while the performance of $\theta_C$ decreases substantially; i.e., the two minimizers induce mechanistically dissimilar models. We see: (i) quadratic paths can be easily identified to mode connect mechanistically dissimilar models; (ii) linear paths cannot be identified, even after permutations; and (iii) mechanistic connectivity does not emerge. We see qualitatively similar results for other datasets, models, and loss (see App. D).

**Conjecture 1.** *(Lack of Linear Connectivity implies Mechanistic Dissimilarity.)* *If two minimizers $\theta_1$ and $\theta_2$ of the loss $\mathcal{L}(f(\mathcal{D}; \theta))$ on a dataset $\mathcal{D}$ cannot be linear mode-connected (up to permutations of neurons), their corresponding models $f(.; \theta_1), f(.; \theta_2)$ must be mechanistically dissimilar.*

In App. F, we prove that the above claim holds for a second-order approximation of the landscape and two-layer model on a simplified data-generating process. Here, we show extensive empirical evidence of its validity in more complex settings. Specifically, we follow the experimental protocol of Neyshabur et al. (2020), who demonstrate that a pretrained model exhibits linear mode connectivity on the original pretraining dataset before and after fine-tuning on another target dataset. Specifically, we train VGG-13 and ResNet-18 models on our synthetic datasets with partially predictive cues and then fine-tune them on data without cues. Results on CIFAR-10 with box cues are shown in Fig. 5 and remaining are deferred to App. E. We see that *when linear mode connectivity does not hold, the fine-tuned models behave differently on counterfactuals.* For example, the models before and after fine-tuning using a large learning rate do not exhibit linear mode connectivity; correspondingly, the fine-tuned models exhibit clear invariance to cue attributes, while the pretrained models do not. Similarly, under perfect correlation between labels and cue attributes, with even a small initial learning rate, fine-tuned models are not linear mode connected with their pretrained counterparts; correspondingly, the fine-tuned models learn different mechanisms and show different behavior on counterfactuals. This result can also be analyzed through the lens of gradient starvation (Pezeshki et al., 2021), which argues that the learning dynamics for simple but highly predictive attributes suppress the learning of other, more complex predictive attributes. This implies the model will be rendered invariant to natural attributes in our setup, which are generally more complex to represent (Scimeca et al., 2021; Hermann et al., 2020). Consequently, the model lacks any transferable mechanisms and hence the model's mechanisms have to be altered to fit the new target distribution. To summarize, these results illustrate that altering a model's mechanisms, such as inducing invariance to spurious attributes, can require surmounting loss barriers. *Naïve fine-tuning can thus fail at this task when using small or moderate learning rates.* While large learning rates can overcome this specific problem, they also obviate pretraining: the goal of pretraining is to help improve sample efficiency before transfer to a target distribution; fine-tuning with large learning rates will distort any useful features learned from the pretraining dataset and render the pipeline's sample efficiency similar to training from scratch, as shown by He et al. (2019); Kumar et al. (2022).

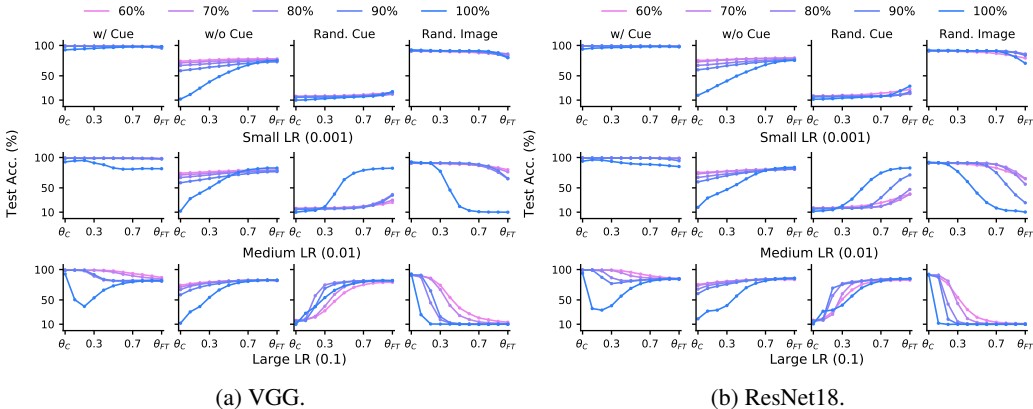

(a) VGG.                                      (b) ResNet18.

Figure 5: **Naïve Fine-Tuning Can Fail to Alter a Model's Mechanisms.** We train VGG-13 and ResNet-18 models on our synthetic CIFAR-10 dataset with box-cues and perform naïve fine-tuning on data without cues for 100 epochs using different initial learning rates (LR) and a step-decay schedule. Corresponding models are denoted $\theta_C$ and $\theta_{FT}$; line colors denote proportion of dataset with synthetic cues; titles denote evaluation datasets, similar to Fig. 4. We plot test accuracy as a function of location on the linear paths (after permutation). Using a large learning rate or enforcing perfect correlation between the cue and label induces loss barriers along the linear path, i.e., linear mode connectivity does not hold. Correspondingly, the models respond differently to counterfactuals, i.e, they are mechanistically dissimilar and not connected. For a small/medium learning rate, $\theta_{FT}$ remains mechanistically similar to $\theta_C$, responding similarly on counterfactuals. Correspondingly, linear mode connectivity holds between the models for data with cues. We see qualitatively similar results for other datasets, models, and loss (see App. E).

## 5    MECHANISTIC FINE-TUNING: ALTERING A MODEL'S MECHANISMS

In this section, we explore the problem of *mechanistic fine-tuning*, where one aims to alter a specific existing mechanism in a pretrained model by fine-tuning the model on a small dataset where the mechanism targeted for alteration performs poorly. This problem setup is also common in recent works on eliminating a model's reliance on spurious attributes (Kirichenko et al., 2022b; Kumar et al., 2022). We now show that our newfound understanding of neural network loss landscapes from the perspective of mechanistic similarity can be used to address this problem.

**Connectivity-Based Fine-Tuning:** Sec. 4 shows that lack of linear mode connectivity between two models is intricately related to their mechanistic dissimilarity. *We thus argue that a valid strategy for altering a model's mechanisms involves moving the model to a region in the landscape that does not exhibit linear mode connectivity to the current parameters.* However, there can be multiple such linearly disconnected regions and we specifically want the ones

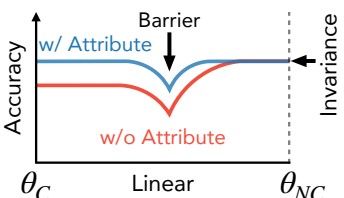

Figure 6: **Geometric clues for altering a model's mechanism .** Given a discriminative input attribute $C$, a linear path connecting a model ($\theta_{NC}$) invariant to the attribute versus a model that relies on the attribute ($\theta_C$) shows (i) an accuracy drop (loss barrier) along the path and (ii) invariant predictions at the endpoint corresponding to $\theta_C$, i.e., $\mathcal{L}(\theta_{NC}, \mathcal{D}_{NC}) = \mathcal{L}(\theta_{NC}, \mathcal{D}_C)$.

that boast our desired invariances. To identify these regions, we propose to use the minimal dataset, denoted $\mathcal{D}_{NC}$, that does not contain the attribute C that the mechanism targeted for alteration in model $f(.; \theta_C)$ tries to identify (see Fig. 6). Specifically, let $\mathcal{L}_{CE}$ denote the cross-entropy loss; $\mathcal{D}^i$ denote the subset of a dataset $\mathcal{D}$ corresponding to samples that belong to the $i^{\text{th}}$ class in a $K$-class classification problem; $\gamma_{\theta \to \theta_C}(t)$ denote the linear path between $\theta$, the parameters we are trying to identify, and $\theta_C$; and $f_r(x; \theta)$ denote the model's representation for an input $x$. Then, Connectivity-Based Fine-Tuning (CBFT) involves running alternating minimization on the following losses:

$$(i) \quad \mathcal{L}_1(\theta) = \underset{\theta}{\operatorname{argmin}} \ \mathbb{E}_{t \sim \text{Truncate}(\mathcal{N}(0.5, 0.5), [0,1])} |\lambda_1 - \mathcal{L}_{CE}(f(\mathcal{D}_C; \gamma_{\theta \to \theta_C}(t)), y)|;$$

$$(ii) \quad \mathcal{L}_2(\theta) = \underset{\theta}{\operatorname{argmin}} \ \mathcal{L}_{CE}(f(\mathcal{D}_{NC}; \theta), y) + \frac{\lambda_2}{K} \sum_{k=1}^{K} \left\| \mathbb{E}_{x \in \mathcal{D}_C^k}(f_r(x; \theta)) - \mathbb{E}_{\tilde{x} \in \mathcal{D}_{NC}^k}(f_r(\tilde{x}; \theta)) \right\|. \tag{3}$$

Table 1: **Evaluating CBFT.** We train ResNet-18 models on our synthetic CIFAR-10, CIFAR-100, and Dominoes dataset with different proportions of samples with cue features and fine-tune them using 2500 "clean" samples from a dataset without any cues. Test accuracies (%) on counterfactual test datasets with No Cue (NC), with Cue (C), Randomized Cue (RC), and Randomized Image (RI) are reported (mean of three seeds). We compare our method, Connectivity-Based Fine-Tuning (CBFT), with several baselines: Fine-tuning with a medium/small learning rate (FT$_{M/S}$), LLR (Kirichenko et al., 2022b), and LPFT (Kumar et al., 2022). $\sim$ denotes invariance is desirable, i.e., accuracy should be similar to that on NC; $\uparrow/\downarrow$ indicate higher/lower accuracy is desirable; best results are in bold. We generally see that all baselines yield large degradations in the absence of cues, and even achieve very high accuracy when the underlying image is randomized. Meanwhile, CBFT is able to break reliance on cues, inducing representations that are often completely invariant to its presence.

| | 60% Cue data | | | | 70% Cue data | | | | 80% Cue data | | | | 90% Cue data | | | |
|---|---|---|---|---|---|---|---|---|---|---|---|---|---|---|---|---|
| C-10 | NC$^\uparrow$ | C$^\sim$ | RC$^\sim$ | RI$^\downarrow$ | NC$^\uparrow$ | C$^\sim$ | RC$^\sim$ | RI$^\downarrow$ | NC$^\uparrow$ | C$^\sim$ | RC$^\sim$ | RI$^\downarrow$ | NC$^\uparrow$ | C$^\sim$ | RC$^\sim$ | RI$^\downarrow$ |
| FT$_M$ | **75.7** | 98.4 | 23.6 | 83.4 | **75.8** | 98.6 | 27.7 | 78.6 | 71.3 | 97.7 | 37.6 | 63.6 | 67.2 | 95.4 | 49.6 | 46.6 |
| FT$_S$ | 75.8 | 98.7 | 17.5 | 90.1 | 74.9 | 98.8 | 16.3 | 91.1 | 69.9 | 98.4 | 15.7 | 90.9 | 64.7 | 97.9 | 15.3 | 90.7 |
| LLR | 71.6 | 95.1 | 36.3 | 57.1 | 70.9 | 95.8 | 29.9 | 65.8 | 65.1 | 81.8 | 27.0 | 53.2 | 59.3 | 70.7 | 24.6 | 40.7 |
| LPFT | 70.6 | 88.1 | 21.0 | 70.7 | 69.6 | 87.3 | 18.7 | 72.5 | 64.4 | 63.8 | 18.8 | 48.0 | 59.7 | 56.6 | 19.8 | 37.8 |
| CBFT | 74.1 | **71.5** | **73.4** | **8.75** | 73.2 | **69.2** | **72.3** | **8.60** | 70.0 | **70.0** | **69.5** | **9.68** | 67.9 | **72.5** | **68.1** | **13.1** |

| | 60% Cue data | | | | 70% Cue data | | | | 80% Cue data | | | | 90% Cue data | | | |
|---|---|---|---|---|---|---|---|---|---|---|---|---|---|---|---|---|
| C-100 | NC$^\uparrow$ | C$^\sim$ | RC$^\sim$ | RI$^\downarrow$ | NC$^\uparrow$ | C$^\sim$ | RC$^\sim$ | RI$^\downarrow$ | NC$^\uparrow$ | C$^\sim$ | RC$^\sim$ | RI$^\downarrow$ | NC$^\uparrow$ | C$^\sim$ | RC$^\sim$ | RI$^\downarrow$ |
| FT$_m$ | **44.4** | 99.2 | 12.8 | 85.3 | **40.3** | 99.6 | 12.3 | 89.8 | 33.6 | 99.0 | 11.4 | 90.5 | 25.2 | 79.2 | 9.79 | 57.9 |
| FT$_s$ | 43.1 | 99.6 | 10.3 | 93.6 | 38.2 | 99.7 | 10.5 | 95.7 | 32.5 | 99.6 | 10.4 | 97.0 | 24.5 | 39.4 | 4.87 | 30.9 |
| LLR | 35.5 | 99.2 | 12.1 | 89.0 | 31.5 | 98.6 | 11.3 | 89.6 | 25.3 | 96.7 | 10.6 | 89.4 | 18.9 | 75.1 | 9.1 | 58.7 |
| LPFT | 35.1 | 93.2 | 10.3 | 82.3 | 31.1 | 90.2 | 9.89 | 78.5 | 25.6 | 89.6 | 9.70 | 80.8 | 18.7 | **28.6** | **4.42** | **19.6** |
| CBFT | 42.7 | **65.0** | **36.4** | **14.6** | 38.5 | **66.7** | **34.7** | **21.2** | **34.6** | **69.3** | **23.0** | **27.9** | **28.5** | 72.9 | 23.2 | 46.0 |

| | 60% Cue data | | | | 70% Cue data | | | | 80% Cue data | | | | 90% Cue data | | | |
|---|---|---|---|---|---|---|---|---|---|---|---|---|---|---|---|---|
| Dom. | NC$^\uparrow$ | C$^\sim$ | RC$^\sim$ | RI$^\downarrow$ | NC$^\uparrow$ | C$^\sim$ | RC$^\sim$ | RI$^\downarrow$ | NC$^\uparrow$ | C$^\sim$ | RC$^\sim$ | RI$^\downarrow$ | NC$^\uparrow$ | C$^\sim$ | RC$^\sim$ | RI$^\downarrow$ |
| FT$_m$ | **77.4** | 96.8 | 43.8 | 56.1 | **76.6** | 96.6 | 42.7 | 58.7 | **74.1** | 95.7 | 41.7 | 61.3 | **68.8** | 95.1 | 40.0 | 57.5 |
| FT$_s$ | 76.4 | 96.9 | 37.5 | 62.4 | 76.8 | 96.6 | 32.5 | 66.5 | 73.2 | 96.4 | 30.8 | 67.7 | 67.3 | 95.2 | 31.2 | 65.6 |
| LLR | 74.6 | 94.4 | 39.8 | 53.0 | 73.9 | 93.2 | 36.3 | 54.7 | 70.8 | 84.8 | 33.1 | 46.6 | 63.3 | 77.0 | 31.2 | 39.0 |
| LPFT | 73.2 | 92.5 | 38.0 | 51.8 | 72.7 | 88.0 | 34.8 | 50.9 | 69.4 | 34.8 | 33.1 | 39.1 | 61.2 | 60.8 | 31.2 | 26.6 |
| CBFT | 72.0 | **64.9** | **67.5** | **9.9** | 71.5 | **70.0** | **59.2** | **12.1** | 70.8 | **69.7** | **65.9** | **11.9** | 67.2 | **68.7** | **61.5** | **14.9** |

The first step above uses a truncated Gaussian distribution constrained to the range $[0, 1]$ to randomly sample a point on the linear path between $\theta$, $\theta_C$ and maximizes the loss at this point up to an upper bound $\lambda_1$. This promotes a loss barrier near the center of the linear path between $\theta$, $\theta_C$. Meanwhile, the second step enforces invariance to the attribute C by reducing the distance between class-average representations on $\mathcal{D}_{NC}$ and $\mathcal{D}_C$, while simultaneously promoting learning of correct labels on the minimal dataset $\mathcal{D}_{NC}$. Detailed ablations are shown in App. C.1 and we see both steps are important for getting the desired results: the barrier loss helps induce a mechanistically dissimilar model, while adding an invariance penalty helps select the exact mechanisms we want the models to differ in.

**Evaluating CBFT:** We empirically validate the effectiveness of CBFT compared to recent baselines designed for reducing a model's reliance on spurious attributes (Kirichenko et al., 2022b; Kumar et al., 2022) (see App. A.2 for training details). Results are reported in Tab. 1. We see that *while the baselines perform well on clean data, they do not yield desired behavior on counterfactual datasets*: e.g., they achieve high accuracy even if we randomize the image. In contrast, *we see that beyond just performing well on clean data, CBFT models show the desired behaviors*: sensitivity to randomization of the image and invariance to spurious attributes. These results suggest CBFT successfully performs mechanistic fine-tuning and provide further corroboration to the claim that lack of linear connectivity implies mechanistic dissimilarity between two models (see Conj. 1).

## 6 RELATED WORK

**Mode connectivity.** Existence of a single, continuous manifold connecting global minimizers was first identified theoretically by Freeman & Bruna (2016); Nguyen (2019) and empirically discovered in concurrent works under the title of "mode connectivity" by Garipov et al. (2018) and Draxler et al. (2018). A geometrical characterization of this manifold was provided by Simsek et al. (2021), who showed the manifold is *primarily* composed of affine subspaces, i.e., linearly connectable solutions. Connectivity properties have been used for designing and analyzing algorithms for several practically relevant applications, such as ensembling (Benton et al., 2021; Izmailov et al., 2018; Wortsman et al., 2021; 2022a), network pruning (Frankle et al., 2020; Entezari et al., 2021), adversarial robustness (Zhao et al., 2020), and multi-task/continual learning (Mirzadeh et al., 2020). During

|  | Linear Mode | Nonlinear Mode | Linear Mechanistic | Nonlinear Mechanistic |
|---|---|---|---|---|
| Mechanistically Similar | ✓ | ✓ | ✓ | ✓ |
| Mechanistically Dissimilar | ✗* | ✓ | ✗ | ✗ |

Table 2: **Summarizing our Findings.** ✓, ✗ respectively indicate whether there always exist paths along which mechanistically (dis)similar models identified using gradient-based optimization can exhibit the type of connectivity specified in the column title.

the course of this work, we became aware of the contemporary empirical paper by Juneja et al. (2022), who investigate whether minimizers connected via linear paths follow similar "decision rules". Their analysis focuses on NLP tasks and does not involve modeling the data-generating process or counterfactual evaluation; their results can be regarded as use of an alternative strategy to further verify our claims on a different modality.

**Fine-tuning.** Fine-tuning is a well-established practice in deep learning. The most basic fine-tuning method is to treat the pretrained model as an initialization, and continue training with new data. A variant is to train only a subset of parameters, such as the final classification layer (Kirichenko et al., 2022b;a), possibly fine-tuning the entire model after that (Kumar et al., 2022; Rosenfeld et al., 2022). While Kirichenko et al. (2022b) argue that "last layer re-training is sufficient for robustness to spurious correlations", our more in-depth evaluation shows that this method actually does not eliminate sensitivity to spurious features. Our findings are more congruent with those of Neyshabur et al. (2020), who find that fine-tuned models tend to remain linearly connected to the pretrained model, suggesting that fine-tuning may fail to change the prediction mechanisms used by a model.

**Model editing.** Model editing refers to fine-tuning approaches that aim to make a targeted change to a particular aspect of model's behavior without incidentally affecting other aspects. For instance, Sinitsin et al. (2020) give the example of correcting a model's prediction error on a particular example without changing its predictions on other examples. Prior work on model editing aims to make changes that are "local" in input space, e.g., only affecting the model's "understanding" of who the current prime minister of the UK is (Mitchell et al., 2021; Santurkar et al., 2021). Mechanistic fine-tuning shares this motivation of "targeted" alteration of a model; however, the problem's overarching goal is to make changes to the causal mechanisms the model implements to make its predictions. Specifically, we aim to make a model invariant to features that is was not already invariant to (or vice versa), without changing any of its other learned invariances. Such a change can influence many of a model's predictions, making model editing approaches inappropriate for our setup.

## 7 CONCLUSION

In this work, we characterized how minimizers that induce models reliant on different prediction mechanisms are connected in the loss landscape (see Tab. 2 for a summary). Such models can perform equally well on a specific distribution, yet differ vastly in their performance on other distributions. We proposed a notion of *mechanistic similarity* to describe such models, instantiating the idea with shared invariances, and extending the prior notion of mode-connectivity to account for mechanistic similarity, which we call *mechanistic connectivity*. Our goal is *mechanistic fine-tuning*, in which one alters certain specific prediction mechanisms used by a model. Our analysis reveals several surprising findings: (i) mechanistically dissimilar minimizers can be mode-connected via relatively simple, but non-linear, paths; (ii) linear mode connectivity of two minimizers is intricately related to the mechanistic similarity of their induced models; (iii) naïve fine-tuning can fail to eliminate spurious attributes learned during pretraining, arguably due to loss barriers between mechanistically dissimilar minimizers; and (iv) finding linearly disconnected regions in the landscape enables sample-efficient alteration of a model's mechanisms. While the datasets we use differ only via our introduction of synthetic attributes, it should be straightforward to generalize our work to any other setting involving two models (with the same parameter space) trained on different distributions. For instance, it can be interesting to compare models trained on completely different tasks or data modalities. It can also be interesting, in the light of our results, to analyze and understand benefits/ limitations of recent methods that exploit linear interpolability of fine-tuned models to create compute-efficient ensembles (Wortsman et al., 2022b;a). We leave these directions to future work.

---

*There are exceptional, but primarily theoretical, cases where the connectivity defn. can hold (see App. F).

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

# A  TRAINING DETAILS AND DATASETS

When training from scratch (e.g., in Fig. 4), we train models using SGD for 100 epochs with a batch-size of 256. Learning rate starts at 0.1 and is dropped by a factor of 10 at the 40th and 80th epochs. No data augmentations are used. When fine-tuning to assess linear connectivity in Fig. 5, we train models for a further 100 epochs on data without cues using different initial learning rates, but the same step-decay schedule (decay factor of 0.1 at decay epochs 40 and 80).

## A.1  DATASET VISUALIZATIONS AND TRAINING CURVES

When using synthetic datasets, if a proportion $c$ of samples is to be assigned the cue feature, we use the first $c\%$ samples of all classes to assign them the respective cues. We do not store the samples beforehand; instead, we use manually designed PyTorch data-loaders that allow for easy manipulation of samples in an online manner, enabling straightforward counterfactual evaluations. While the dataset construction was discussed in Sec. 2.2, we provide several visualizations of randomly sampled datapoints from different classes and their counterfactuals in Fig. 7 (CIFAR-10 with box cue), Fig. 8 (CIFAR-100 with box/color cue), and Fig. 9 (Dominoes: CIFAR-10 with concatenated FashionMNIST image cue). Learning curves with train/test accuracies for VGG / ResNet-18 models trained on different proportions of samples with cue features for these datasets are reported in Figs. 10a and 11a (CIFAR-10 with box cue), Fig. 10b, 11b (CIFAR-100 with box/color cue), and Fig. 10c, 11c (Dominoes).

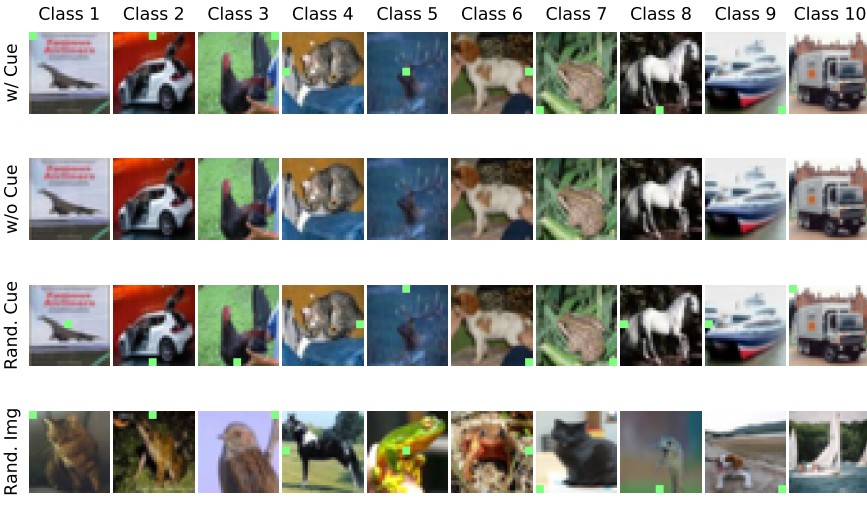

Figure 7: CIFAR-10 with Box cue.

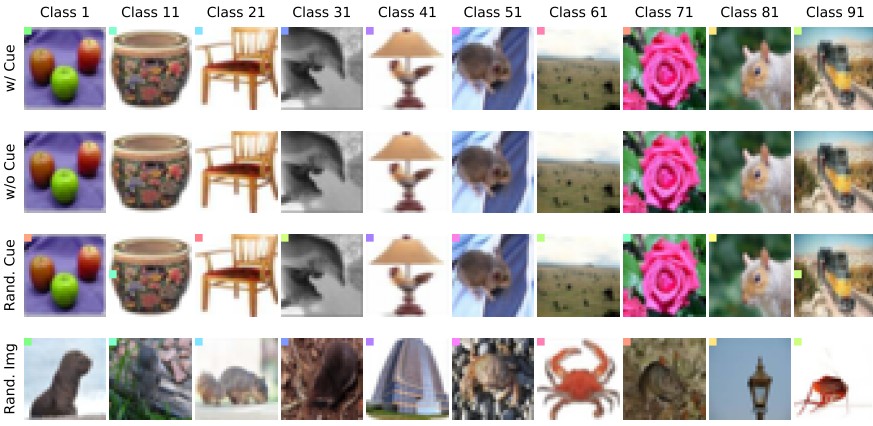

Figure 8: CIFAR-100 with Box/Color cue.

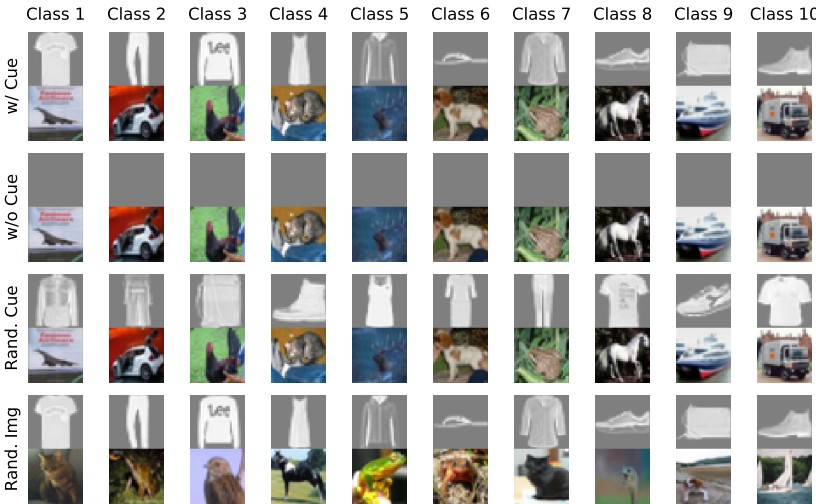

Figure 9: Dominoes: CIFAR-10 with their corresponding ID image from Fashion-MNIST as the cue.

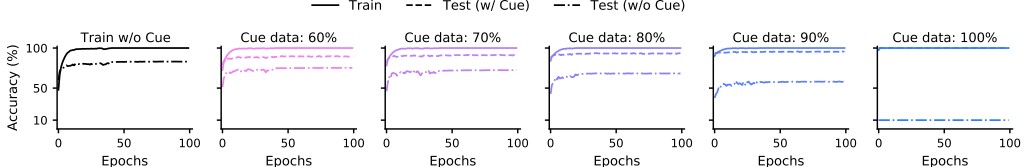

(a) CIFAR-10 with box cues, wherein the box's location is a function of the target label.

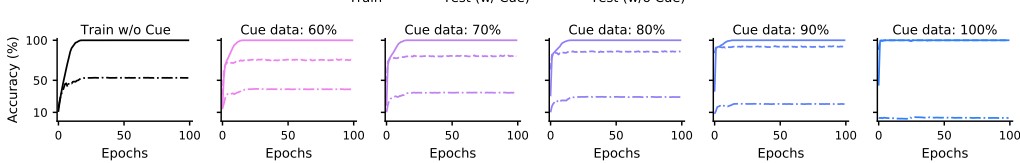

(b) CIFAR-10 with box cues, wherein the box's location is a function of the target label.

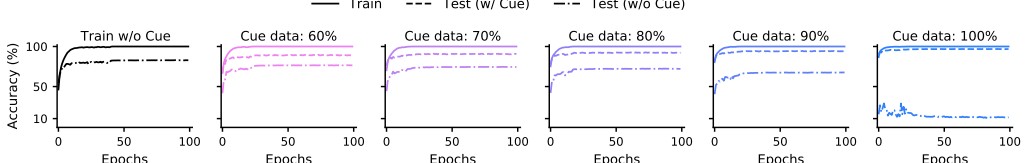

(c) Dominoes, wherein Fashion-MNIST images are appended to CIFAR-10 images and act as the spurious cues.

Figure 10: Learning curves for VGG-13 models.

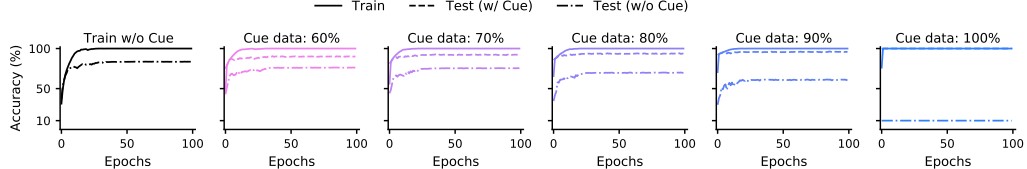

(a) CIFAR-10 with box cues, wherein the box's location is a function of the target label.

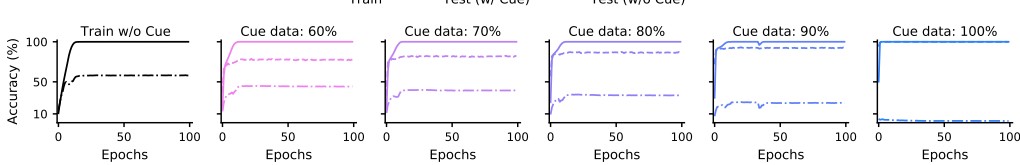

(b) CIFAR-10 with box cues, wherein the box's location is a function of the target label.

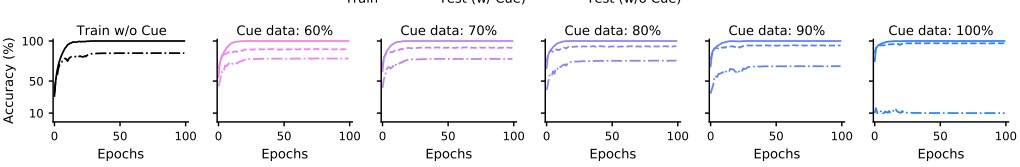

(c) Dominoes, wherein Fashion-MNIST images are appended to CIFAR-10 images and act as the spurious cues.

Figure 11: Learning curves for ResNet-18 models.

## A.2 TRAINING DETAILS FOR TAB. 1

We train models using SGD on the synthetic data with cue features, reserving 2,500 training samples as "clean" data for further fine-tuning to remove reliance on the spurious cue. Depending on the method, the fine-tuning setup involves different hyperparameters. For consistency, we follow Kirichenko et al. (2022b) and Kumar et al. (2022) in using a cosine schedule for fine-tuning on clean data.

**Naïve Fine-Tuning**. We use different initial learning rates, including medium (0.01) and small (0.001). For a large learning rate, we note that while fine-tuning on a minimal set induces good invariance properties, the performance on the original, without cue data (called NC in tables) is often rather poor. Hence, we omit those results.

**LLRT** (Kirichenko et al., 2022b). We freeze the model parameters at their current state, remove the final linear layer, and replace it with a randomly initialized one. The layer is fine-tuned on clean data for 100 epochs with a cosine decay schedule that starts at a LR of 30.

**LPFT** (Kumar et al., 2022). First, we follow the protocol above for LLRT to produce a new linear layer. Thereafter, the entire model is fine-tuned on clean data for 20 epochs with initial learning rates of 0.01, 0.001, and 0.0001. The best retrieved results on validation data are reported.

**CBFT**. We run CBFT for 20 epochs, using an initial learning rate of 0.01 with a cosine decay schedule (similar to the baselines). The method turns out to be fairly robust to the exact values of $\lambda_1$ and $\lambda_2$; hence we fix both to 1. We also note that since training will yield gradients for the model that has parameters $\gamma_{\theta \to \theta_C}(t)$, we need to explicitly compute the gradients for $\theta$ by using the following relationship for some general objective function $\mathcal{L}$:

$$\nabla_\theta \mathcal{L}(\gamma_{\theta \to \theta_C}(t)) = \left(\nabla_\theta \gamma_{\theta \to \theta_C}(t)\right)^T \nabla_{\gamma_{\theta \to \theta_C}(t)} \mathcal{L}(\gamma_{\theta \to \theta_C}(t)) = (1-t)\nabla_{\gamma_{\theta \to \theta_C}(t)} \mathcal{L}(\gamma_{\theta \to \theta_C}(t)).$$

Thus, one need only compute gradient of an objective with respect to $\gamma_{\theta \to \theta_C}(t)$ and multiply that by a factor of $1-t$ to retrieve the gradient of the objective with respect to $\theta$. This step has to be carried out explicitly and hence we have to break the optimization process of CBFT into two steps (see Eq. 3), executing alternating minimization for the barrier and invariance losses.

# B IDENTIFYING CONNECTIVITY PATHS

## B.1 QUADRATIC PATHS

The qudaratic path is defined as follows.

$$\gamma_{\theta_1 \to \theta_2}(t) = t^2 \theta_1 + 2t(1-t)\theta_{12} + (1-t)^2 \theta_2. \tag{4}$$

The set of parameters $\theta_{12}$ can be thought of as the vertex of a parabola that helps anchor the curve. To identify this set of parameters, we follow Garipov et al. (2018) and train points uniformly sampled from the quadratic path to achieve zero loss on a given dataset $\mathcal{D}$, i.e.,

$$\theta_{12} = \underset{\theta}{\operatorname{argmin}} \, \mathbb{E}_{x \in \mathcal{D}, t \in [0,1]}(\mathcal{L}(f(x; \gamma_{\theta_1 \to \theta_2)(t)}))). \tag{5}$$

Consequently, note that a quadratic path necessarily depends on the dataset used for its identification and it is not mandatory that it generalize across datasets/distributions. This is precisely what we see in our results in Fig. 4, where we are able to identify quadratic mode connectivty between two sets of parameters on a given dataset, but those paths do not generalize to counterfactual datasets.

## B.2 FINDING PERMUTATIONS FOR LINEAR CONNECTIVITY

Given two minimizers $\theta_1, \theta_2$, identifying the linear path between them involves merely interpolating the parameters. Entezari et al. (2021); Ainsworth et al. (2022); Singh & Jaggi (2020) hypothesize that minimizers discovered using SGD can always be linearly mode connected up to permutations of neurons that align the two models in their activations or weights. That is, there generally exists a permutation $\pi$ that connects $\pi(\theta_1)$ with $\theta_2$ in the sense of Def. 1. To empirically analyze this claim in our work, we identify $\pi$ by maximizing the similarity of activations produced by model with parameters $\theta_1$ and $\theta_2$:

$$\pi^* = \underset{\pi}{\operatorname{argmin}} ||f(x; \pi(\theta_1)) - f(x; \theta_2)||. \tag{6}$$

Given that solving the problem above is NP-Hard (Entezari et al., 2021; Ainsworth et al., 2022; Singh & Jaggi, 2020), we propose to solve it greedily by computing representations at each layer of the two models, finding a permutation that matches the representations maximally, and then repeating the process for the next layer. To this end, we use inputs with a batch-size of 512 and run the matching process over the entire original datasets (i.e., ones without cues). We note that we did conduct minimal experiments on finding permutations using data with cues, instead of without, but never found any noticeable differences in the results. Hence, we decided to use the original data without cues throughout our experiments for finding linear paths. Intuitively, we suspect the exact choice of dataset does not matter for our experimental setup because we analyze pairs of models which include one model that is invariant to the cue and one that is not. Since the invariant models produce the same representations on data with / without cues, the target for permutation matching remains the same.

For finding the permutation, we use SciPy's linear assignment solver (SciPy, 2016). Specifically, if $\pi_{<l}$ denotes permutations before layer $l$, we use the following algorithm.

> **Initialize:** $\theta_1$, $\theta_2$, dataset $\mathcal{D}$, model $f$, $\pi_l{}_1^L = I$, where I is identity permutation.
> $l \leftarrow 1$
> $L \leftarrow$ # of Layers
> **for** $x \in \mathcal{D}$ **do**
> $\quad l \leftarrow 1$
> $\quad$ **for** $l \leq L$ **do**
> $\quad\quad \pi_l = \underset{\pi}{\operatorname{argmin}} ||f_l(x; \pi_{<l}(\theta_1)) - f_l(x; \theta_2)|| \qquad \triangleright f_l$ denotes representation at layer $l$
> $\quad\quad l \leftarrow l + 1$
> $\quad$ **end for**
> **end for**

Table 3: **Ablating CBFT.** We train ResNet-18 models on our synthetic CIFAR-10, CIFAR-100, and Dominoes dataset with different proportions of samples with cue features and fine-tune them using 2500 "clean" samples from a dataset without any cues. Test accuracies (%) on counterfactual test datasets with No Cue (NC), with Cue (C), Randomized Cue (RC), and Randomized Image (RI) are reported. We compare Connectivity-Based Fine-Tuning (CBFT) with two of its ablations (see App. C.1): (i) $-\mathcal{L}_{\text{barrier}}$, for which the barrier inducing loss is removed from the training process and (ii) $-\mathcal{L}_{\text{Inv.}}$, for which the invariance loss is removed. $\sim$ denotes invariance is desirable, i.e., accuracy should be similar to that on NC; $\uparrow/\downarrow$ indicate higher/lower accuracy is desirable; best results are in bold. For discussion, please see App. C.1.

| | 60% Cue data | | | | 70% Cue data | | | | 80% Cue data | | | | 90% Cue data | | | |
|---|---|---|---|---|---|---|---|---|---|---|---|---|---|---|---|---|
| C-10 | NC$^\uparrow$ | C$^\sim$ | RC$^\sim$ | RI$^\downarrow$ | NC$^\uparrow$ | C$^\sim$ | RC$^\sim$ | RI$^\downarrow$ | NC$^\uparrow$ | C$^\sim$ | RC$^\sim$ | RI$^\downarrow$ | NC$^\uparrow$ | C$^\sim$ | RC$^\sim$ | RI$^\downarrow$ |
| CBFT | 74.1 | **71.5** | **73.4** | **8.75** | 73.2 | **69.2** | **72.3** | 8.60 | 70.0 | **70.0** | 69.5 | 9.68 | **67.9** | 72.5 | 68.1 | 13.1 |
| $-\mathcal{L}_{\text{barrier}}$ | **75.8** | 93 | 69.3 | 24.4 | **75.9** | 90 | 72.1 | 18.6 | **71.6** | 89.9 | 66.3 | 23.5 | 67.8 | 89.6 | 65.1 | 20.5 |
| $-\mathcal{L}_{\text{Inv.}}$ | 73.4 | 69.4 | 68.8 | 14.2 | 72.9 | 65.2 | 71.3 | 8.26 | 69.3 | 64.8 | 68.1 | 9.72 | 65.8 | **64.8** | 65 | **10.3** |
| C-100 | NC$^\uparrow$ | C$^\sim$ | RC$^\sim$ | RI$^\downarrow$ | NC$^\uparrow$ | C$^\sim$ | RC$^\sim$ | RI$^\downarrow$ | NC$^\uparrow$ | C$^\sim$ | RC$^\sim$ | RI$^\downarrow$ | NC$^\uparrow$ | C$^\sim$ | RC$^\sim$ | RI$^\downarrow$ |
| CBFT | 42.7 | 65.0 | **36.4** | 14.6 | 38.5 | **66.7** | **34.7** | 21.2 | **34.6** | 69.3 | 23.0 | 27.9 | 28.5 | 72.9 | **23.2** | 46.0 |
| $-\mathcal{L}_{\text{barrier}}$ | **44.7** | 99.8 | 17.5 | 81.6 | **40.2** | 99.9 | 13.7 | 88.9 | 34.6 | 99.9 | 11.3 | 95.1 | 26.5 | 99.1 | 13.5 | 82.2 |
| $-\mathcal{L}_{\text{Inv.}}$ | 43.2 | **59.4** | 36.5 | **12.5** | 35.7 | 64.2 | 26 | 25.5 | 34.1 | 70.2 | **23.5** | 36.7 | 24.7 | 69.2 | 15.9 | **45.6** |
| Dom. | NC$^\uparrow$ | C$^\sim$ | RC$^\sim$ | RI$^\downarrow$ | NC$^\uparrow$ | C$^\sim$ | RC$^\sim$ | RI$^\downarrow$ | NC$^\uparrow$ | C$^\sim$ | RC$^\sim$ | RI$^\downarrow$ | NC$^\uparrow$ | C$^\sim$ | RC$^\sim$ | RI$^\downarrow$ |
| CBFT | 72.0 | **64.9** | **67.5** | 9.9 | 71.5 | **70.0** | **59.2** | 12.1 | 70.8 | **69.7** | **65.9** | 11.9 | 67.2 | **68.7** | **61.5** | 14.9 |
| $-\mathcal{L}_{\text{barrier}}$ | **77.1** | 94.9 | 63.2 | 32.7 | **77.4** | 94.2 | 65.8 | 29.2 | **74.5** | 93.3 | 63.5 | 30.1 | 67.1 | 91.9 | 55.5 | 32.9 |
| $-\mathcal{L}_{\text{Inv.}}$ | 74.2 | 40.4 | 41.8 | **6.93** | 74.6 | 28.2 | 24.9 | **10.6** | 71.3 | 20.1 | 22.2 | **6.92** | 66 | 21.2 | 20.9 | **6.26** |

# C  ADDITIONAL RESULTS ON CBFT

## C.1  ABLATIONS

To analyze the role played by the two loss functions involved in the alternating minimization steps of Connectivity-Based Fine-Tuning (CBFT) (see Sec. 5, Eq. 3), we present an ablation study as follows. We analyze two variants of CBFT: (i) $-\mathcal{L}_{\text{barrier}}$, for which the barrier inducing loss $\mathbb{E}_{t\sim\text{Truncate}(\mathcal{N}(0.5,0.5),[0,1])}|\lambda_1 - \mathcal{L}_{\text{CE}}(f(\mathcal{D}_{\text{C}}; \gamma_{\theta\to\theta_{\text{C}}}(t)), y)|$ has been removed from the training process, and (ii) $-\mathcal{L}_{\text{Inv.}}$, for which the invariance loss $\left(\sum_{k=1}^K \left\|\mathbb{E}_{x\in\mathcal{D}_{\text{C}}^k}(f_r(x;\theta)) - \mathbb{E}_{\tilde{x}\in\mathcal{D}_{\text{NC}}^k}(f_r(\tilde{x};\theta))\right\|\right)$ has been removed. Results are shown in Tab. 3. We find that without the barrier loss, the trained model is unable to break its reliance on spurious cues, even though it generally achieves the best performance on data without cues (NC in table). Meanwhile, without the invariance loss, the trained model indeed loses sensitivity to spurious cues and shows poor performance when the underlying image is randomized, as we desire. However, in few instances the model can become anti-correlated with the spurious cue (e.g., see results on Dominoes). This is expected since the barrier loss's goal is to move the model to a region in the landscape that follows different mechanisms (with respect to the pre-trained model) by inducing a loss barrier; without the invariance loss, the model can learn to induce this barrier by merely becoming anti-correlated with the spurious cue. The invariance loss helps prevent this pitfall, selecting a mechanistically dissimilar region in the landscape that is uncorrelated, instead of anti-correlated with the spurious cue. These results confirm the validity of our claims in Sec. 5: *preventing linear connectivity helps induce mechanistic dissimilarity and an invariance penalty helps select the exact mechanisms we want the models to differ in.* Overall, this ablation study help us infer that while the two losses involved in CBFT have their individual benefits, it is only when they are combined that they give the best results.

## C.2  COMPARISON WITH TRAINING FROM SCRATCH

We compare CBFT against training from scratch on the minimal clean dataset that we assume access to during the training process for all baselines and CBFT in Tab. 1. Specifically, we train ResNet-18 models for 100 epochs using an initial learning rate of 0.1 and a cosine decay schedule. Results are provided in Tab. 4 and we see training from scratch significantly underperforms all baselines and CBFT (results reported in Tab. 1). This is expected since our setup assumes access to only a *minimal* clean dataset for inducing invariance to spurious attributes. Since training from scratch is not a sample efficient strategy, it cannot perform well in this setting. We also highlight that using as initialization a model pretrained on an unclean dataset, i.e., one that contains spurious attributes, will

Table 4: **Training from scratch on minimal clean data.** We train ResNet-18 models on the 2500 "clean" samples used in Tab. 1 from the original CIFAR-10, CIFAR-100, and Dominoes datasets. Test accuracies (%) on counterfactual test datasets with No Cue (NC), with Cue (C), Randomized Cue (RC), and Randomized Image (RI) are reported. $\sim$ denotes invariance is desirable, i.e., accuracy should be similar to that on NC; $\uparrow/\downarrow$ indicate higher/lower accuracy is desirable. We clearly see training from scratch severely underperforms all other baselines and CBFT (see Tab. 1). This is expected since our setup assumes access to only a *minimal* clean dataset for inducing invariance to spurious attributes. Since training from scratch is not a sample efficient strategy, it cannot perform well in this setting. Note that models trained on Dominoes show reduced performance in the presence of cues because we Fashion-MNIST images as cues in that dataset. The shift in distribution is rather non-trivial and seems to sufficiently confuse the model to reduce its performance.

|       | $NC^{\uparrow}$ | $C^{\sim}$ | $RC^{\sim}$ | $RI^{\downarrow}$ |
|-------|------|------|------|------|
| C-10  | 47.5 | 47.4 | 47.5 | 9.69 |
| C-100 | 16.5 | 16.4 | 16.4 | 1.19 |
| Dom.  | 48.5 | 31   | 31   | 10.8 |

make this overall process equal to naïve fine-tuning on the clean dataset; we already provide results for naïve fine-tuning in Tab. 1.

# D    FURTHER RESULTS SHOWING NON-LINEAR MODE CONNECTIVITY OF MECHANISTICALLY DISSIMILAR MINIMIZERS

We train VGG-13 and ResNet-18 models on our synthetic CIFAR-10 / CIFAR-100 / Dominoes datasets with cues (see Figs. 7, 8, and 9) and the original datasets themselves. Parameters of the corresponding models are denoted $\theta_C$ and $\theta_{NC}$. We identify connectivity paths along pairs of parameters, specifically evaluating quadratic paths identified using data without cues (denoted Quadratic w/o Cues), quadratic path identified using data with cue (denoted Quadratic w/ Cue), linear path (denoted Linear), and linear path after permuting $\theta_C$ to maximally match $\theta_{NC}$'s activations (denoted Linear Permuted). In the following, plot titles denote evaluation dataset, including datasets where either the cue is present (denoted w/ Cue), absent (denoted w/o Cue), randomized (denoted Rand. Cue), or the underlying image is randomized but the cue remains the same (denoted Rand. Image). Line colors denote the proportion of dataset that has synthetic cues.

Across all our results, we see the set of parameters $\theta_{NC}$ yields the same performance upon randomization of the cue, while the performance of $\theta_C$ degrades substantially–i.e., the two modes are mechanistically dissimilar due to lack of shared invariances (see Def. 4). Nonetheless, we can identify quadratic (but not linear) paths that mode-connect these mechanistically dissimilar minimizers, hence corroborating Prop. 2 across several datasets and model architectures, showing *mechanistically dissimilar modes can also be mode connected via relatively simple paths as well*. However, different points on the connectivity paths respond differently to counterfactuals, indicating *lack of mechanistic connectivity*.

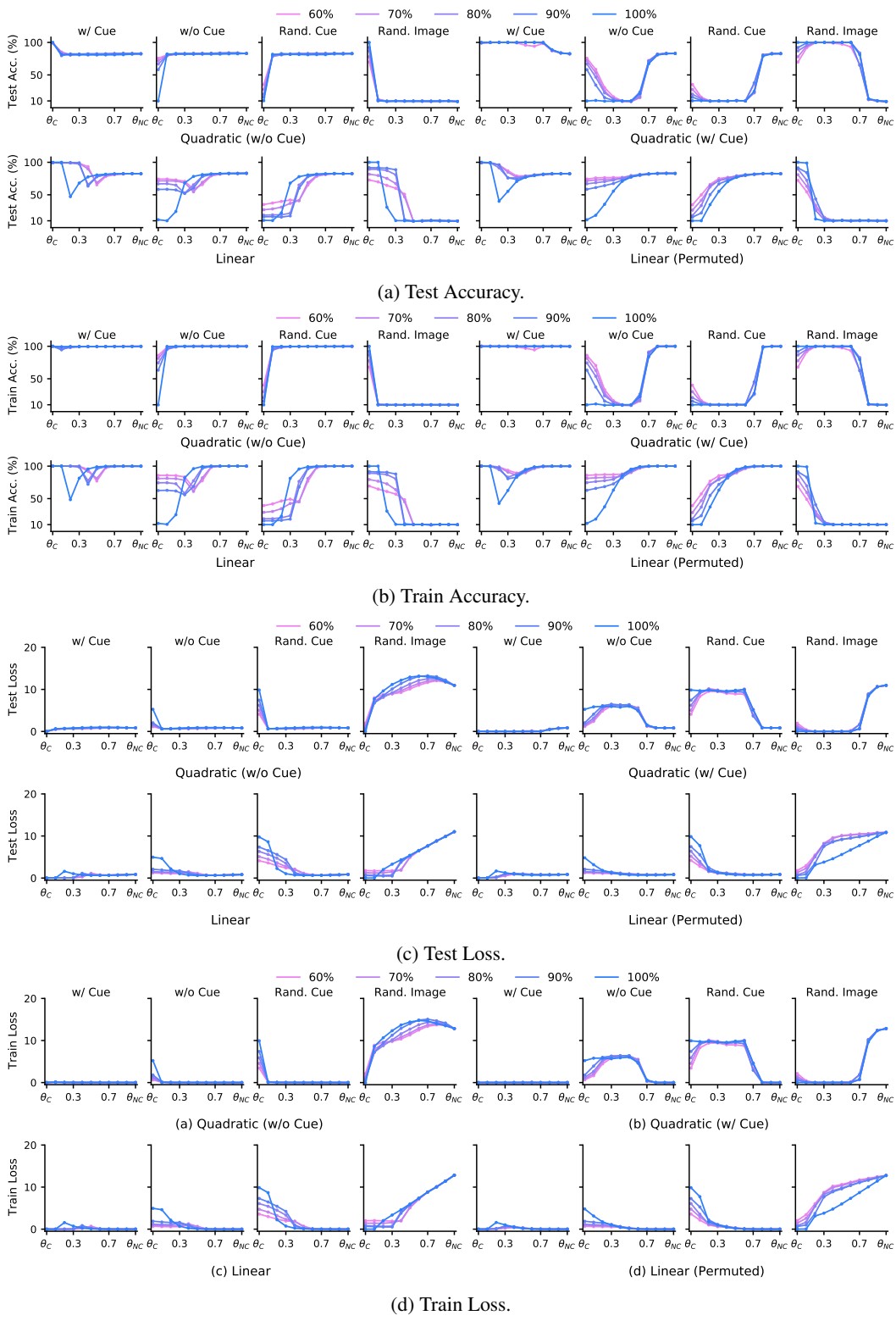

Figure 12: **VGG-13 on CIFAR-10 with Box Cue**. We plot test/train accuracy/loss curves along different connectivity paths and see thorough corroboration of our claims in the main text: Mechanistically dissimilar minimizers can be connected via nonlinear paths on a given dataset, but behave different on counterfactuals, indicating lack of mechanistic connectivity.

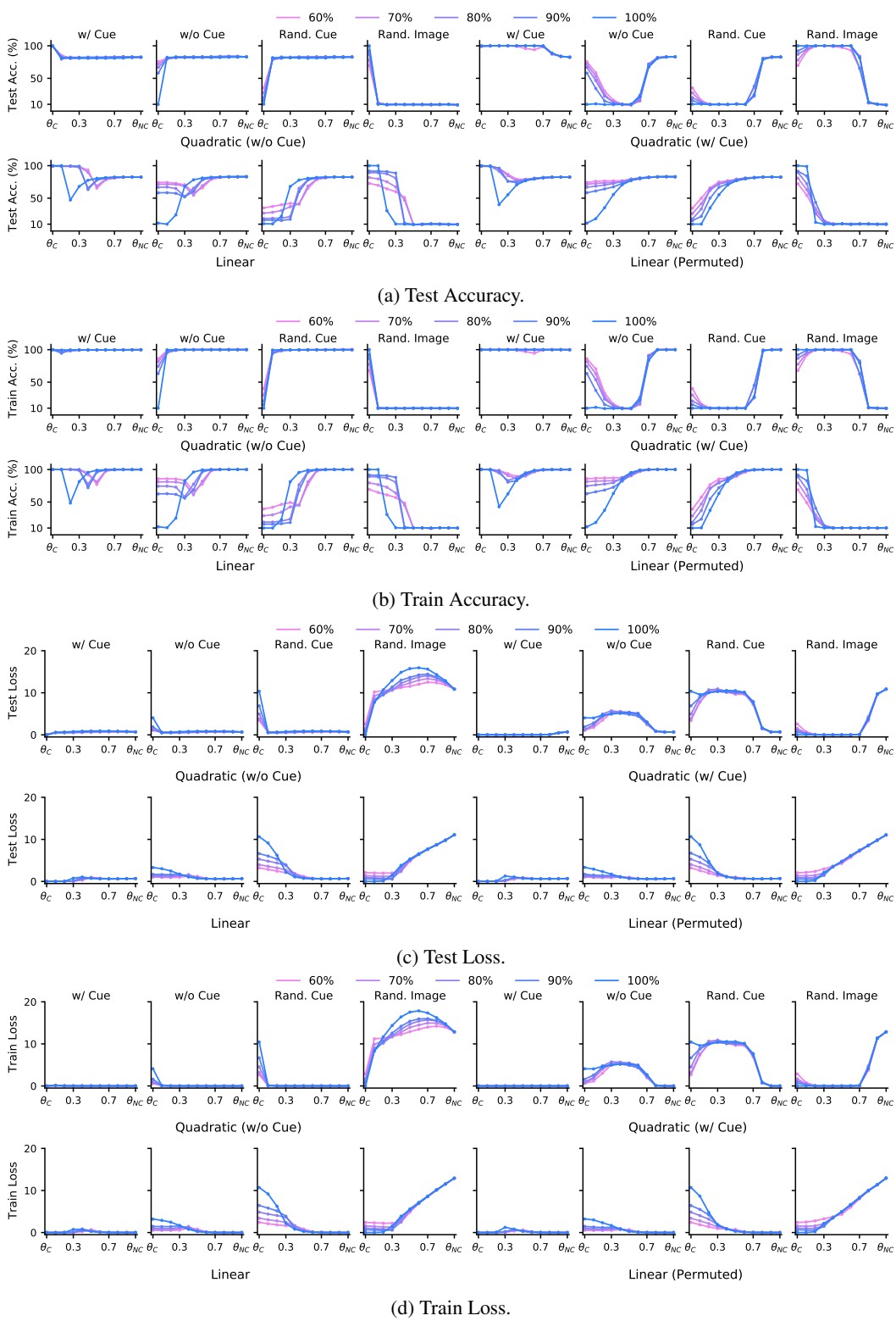

Figure 13: **ResNet-18 on CIFAR-10 with Box Cue**. We plot test/train accuracy/loss curves along different connectivity paths and see thorough corroboration of our claims in the main text: Mechanistically dissimilar minimizers can be connected via nonlinear paths on a given dataset, but behave different on counterfactuals, indicating lack of mechanistic connectivity.

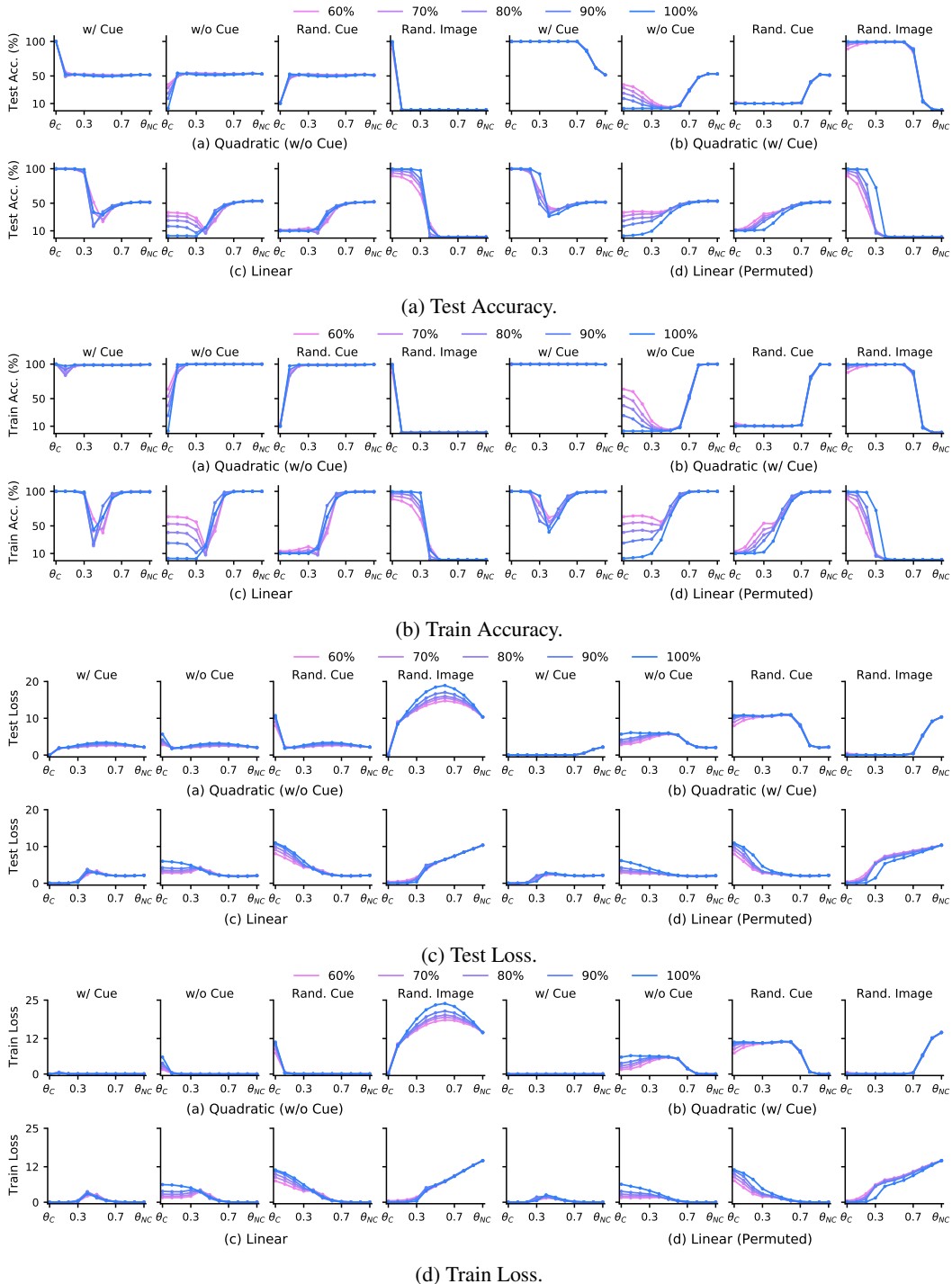

Figure 14: **VGG-13 on CIFAR-100 with Box/Color Cue**. We plot test/train accuracy/loss curves along different connectivity paths and see thorough corroboration of our claims in the main text: Mechanistically dissimilar minimizers can be connected via nonlinear paths on a given dataset, but behave different on counterfactuals, indicating lack of mechanistic connectivity.

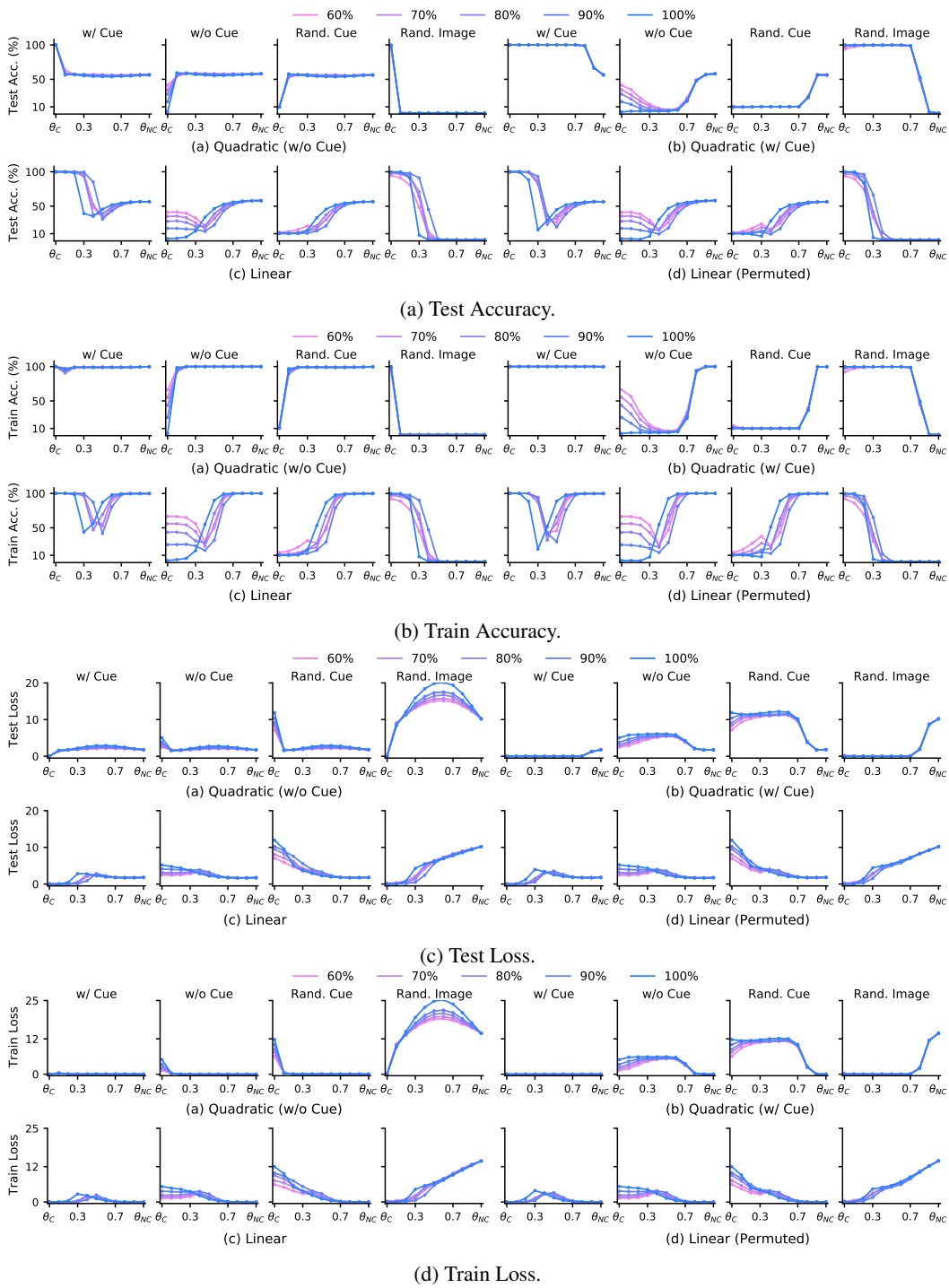

Figure 15: **ResNet-18 on CIFAR-100 with Box/Color Cue**. We plot test/train accuracy/loss curves along different connectivity paths and see thorough corroboration of our claims in the main text: Mechanistically dissimilar minimizers can be connected via nonlinear paths on a given dataset, but behave different on counterfactuals, indicating lack of mechanistic connectivity.

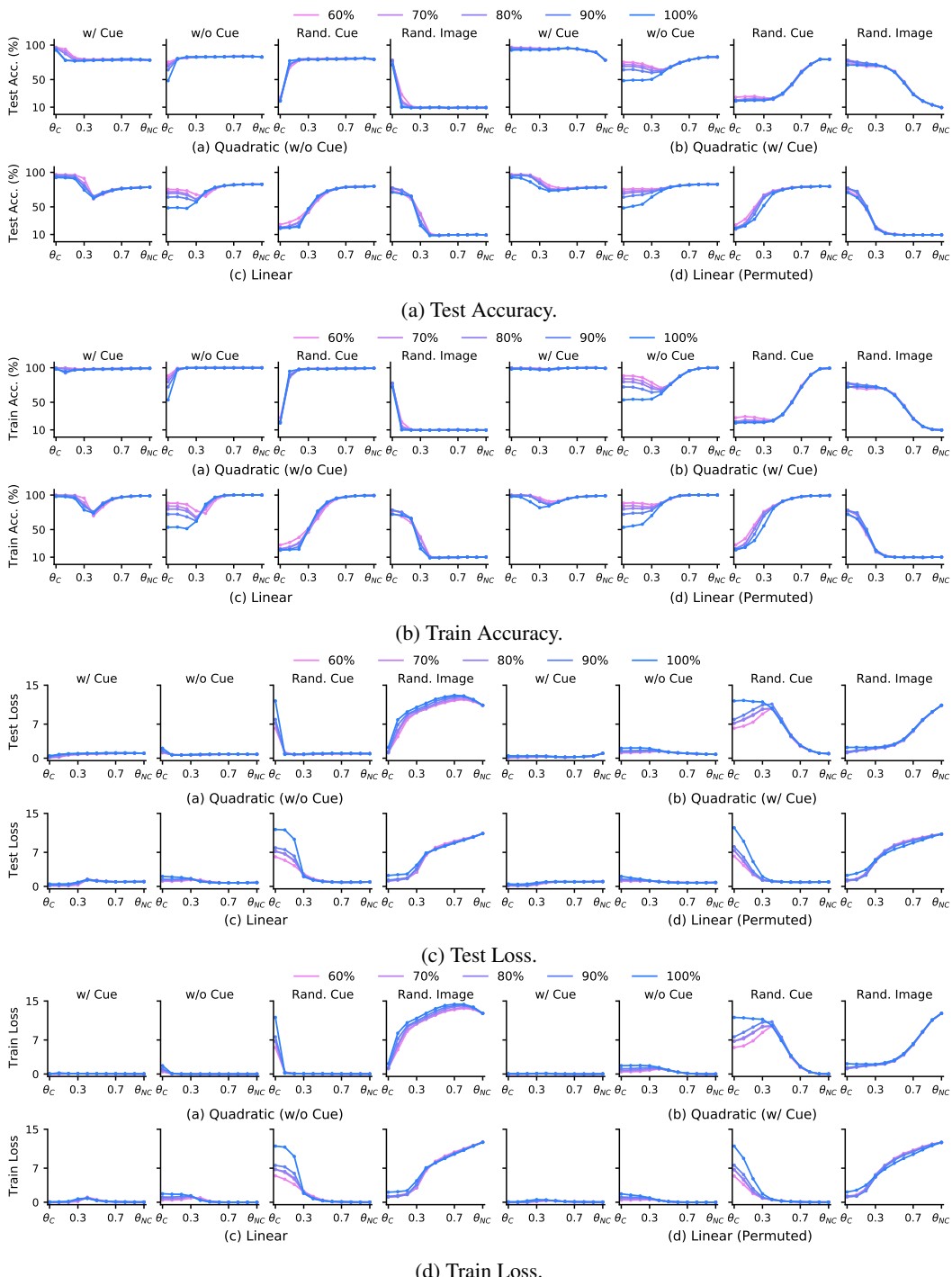

Figure 16: **VGG-13 on Dominoes**. We plot test/train accuracy/loss curves along different connectivity paths and see thorough corroboration of our claims in the main text: Mechanistically dissimilar minimizers can be connected via nonlinear paths on a given dataset, but behave different on counterfactuals, indicating lack of mechanistic connectivity.

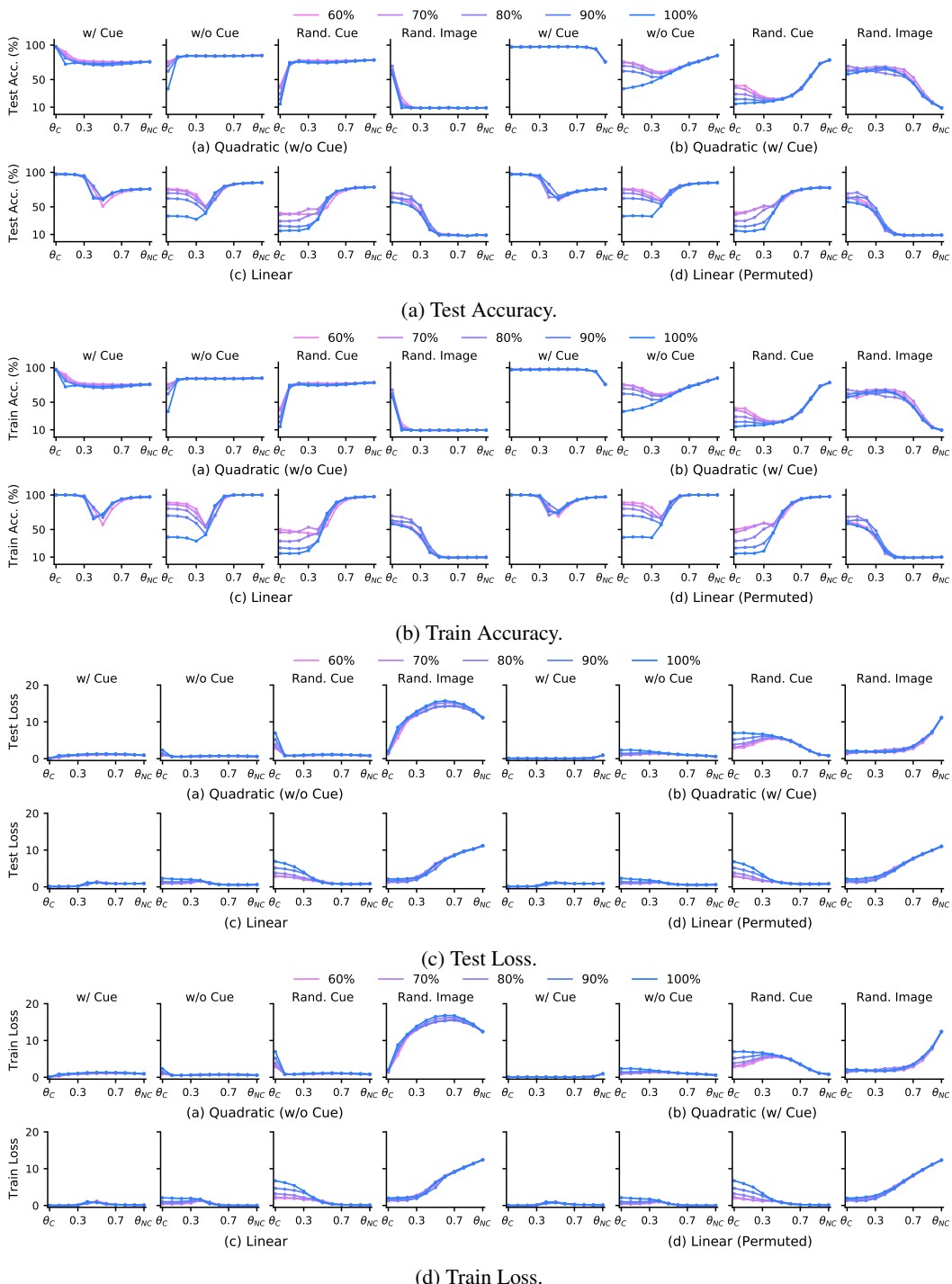

Figure 17: **ResNet-18 on Dominoes**. We plot test/train accuracy/loss curves along different connectivity paths and see thorough corroboration of our claims in the main text: Mechanistically dissimilar minimizers can be connected via nonlinear paths on a given dataset, but behave different on counterfactuals, indicating lack of mechanistic connectivity.

# E   FURTHER RESULTS SHOWING LACK OF LINEAR CONNECTIVITY IMPLIES MECHANISTIC DISSIMILARITY

We train VGG-13 and ResNet-18 models on our synthetic CIFAR-10 / CIFAR-100 / Dominoes datasets with cues (see Figs. 7, 8, and 9). Corresponding models are denoted $\theta_C$. These models are then fine-tuned on the original CIFAR-10 / CIFAR-100 datasets that do not have any cue features. Specifically, we use different learning rates (LR) and train for 100 epochs with a step-decay schedule (decay at epoch 40 and 80 by a factor of 0.1). Corresponding models are denoted $\theta_{FT}$. In the following, plot titles denote evaluation dataset, including datasets where either the cue is present (denoted w/ Cue), absent (denoted w/o Cue), randomized (denoted Rand. Cue), or the underlying image is randomized but the cue remains the same (denoted Rand. Image). Line colors denote the proportion of dataset that has contains our synthetically embedded cues.

Across all our results, we see that using a large enough learning rate or enforcing perfect correlation between the cue and label induces loss barriers along the linear path, i.e., linear mode connectivity does not hold. Correspondingly, we see the models respond differently to counterfactuals, i.e, they are mechanistically dissimilar and not connected. For a small enough learning rate, $\theta_{FT}$ remains mechanistically similar to $\theta_C$, responding similarly on counterfactuals. Correspondingly, we see linear mode connectivity holds between the models on data with cues.

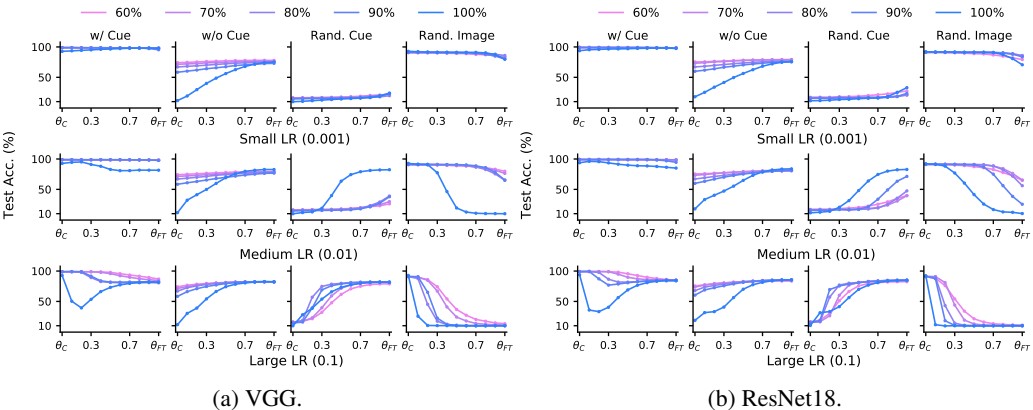

Figure 18: **Fine-tuning of models trained on CIFAR-10 with Box Cue**. We plot test accuracy curves along the linear path between $\theta_C$ and $\theta_{FT}$ and see thorough corroboration of our claims in the main text: Linearly connected minimizers exhibit mechanistic similarity, behaving identically on counterfactual datasets, indicating mechanistic connectivity.

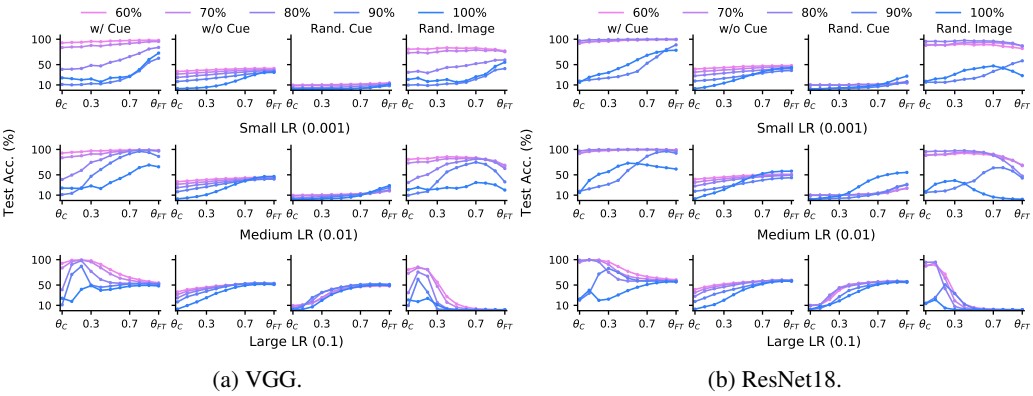

Figure 19: **Fine-tuning of models trained on CIFAR-100 with Box/Color Cue**. We plot test accuracy along the linear path between $\theta_C$ and $\theta_{FT}$ and see thorough corroboration of our claims in the main text: Linearly connected minimizers exhibit mechanistic similarity, behaving identically on counterfactual datasets, indicating mechanistic connectivity.

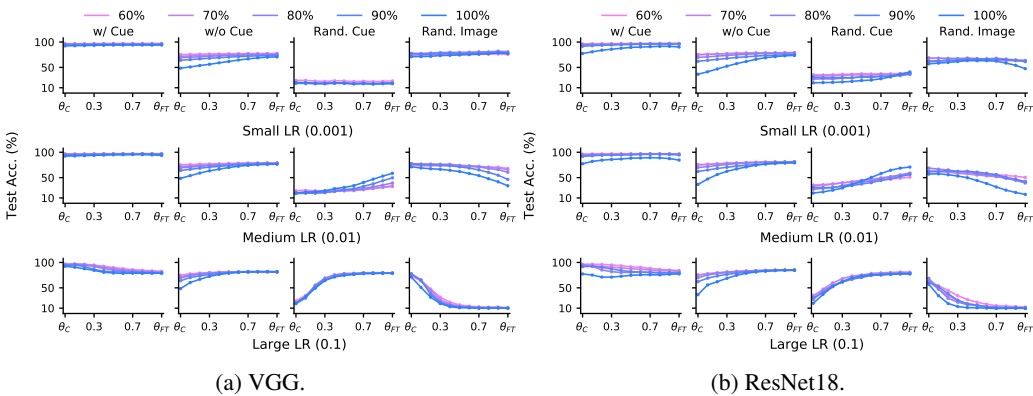

Figure 20: **Fine-tuning of models trained on Dominoes**. We plot test accuracy along the linear path between $\theta_C$ and $\theta_{FT}$ and see thorough corroboration of our claims in the main text: Linearly connected minimizers exhibit mechanistic similarity, behaving identically on counterfactual datasets, indicating mechanistic connectivity.

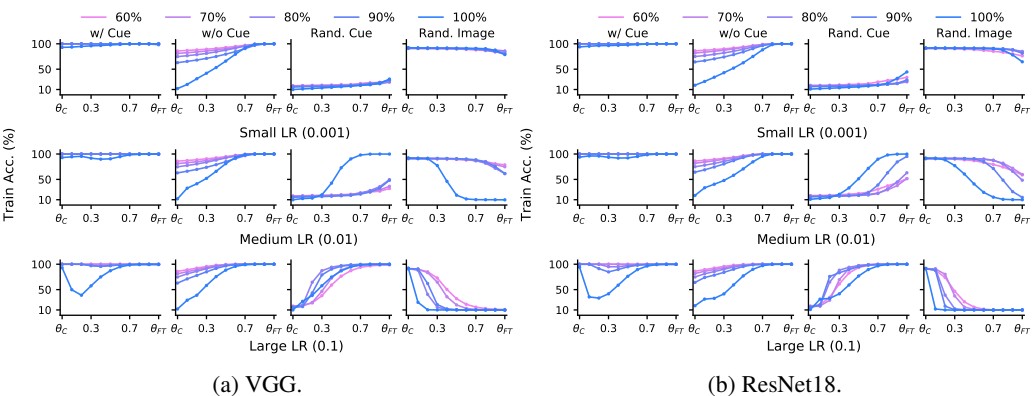

Figure 21: **Fine-tuning of models trained on CIFAR-10 with Box Cue**. We plot train accuracy curves along the linear path between $\theta_C$ and $\theta_{FT}$ and see thorough corroboration of our claims in the main text: Linearly connected minimizers exhibit mechanistic similarity, behaving identically on counterfactual datasets, indicating mechanistic connectivity.

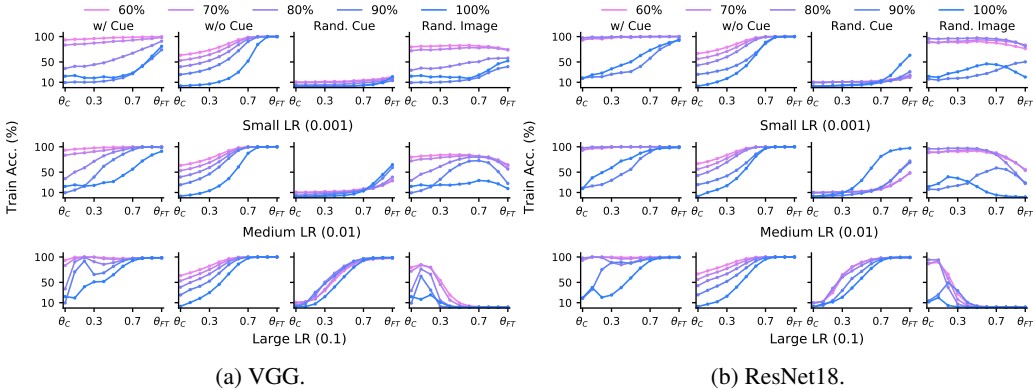

Figure 22: **Fine-tuning of models trained on CIFAR-100 with Box/Color Cue**. We plot train accuracy curves along the linear path between $\theta_C$ and $\theta_{FT}$ and see thorough corroboration of our claims in the main text: Linearly connected minimizers exhibit mechanistic similarity, behaving identically on counterfactual datasets, indicating mechanistic connectivity.

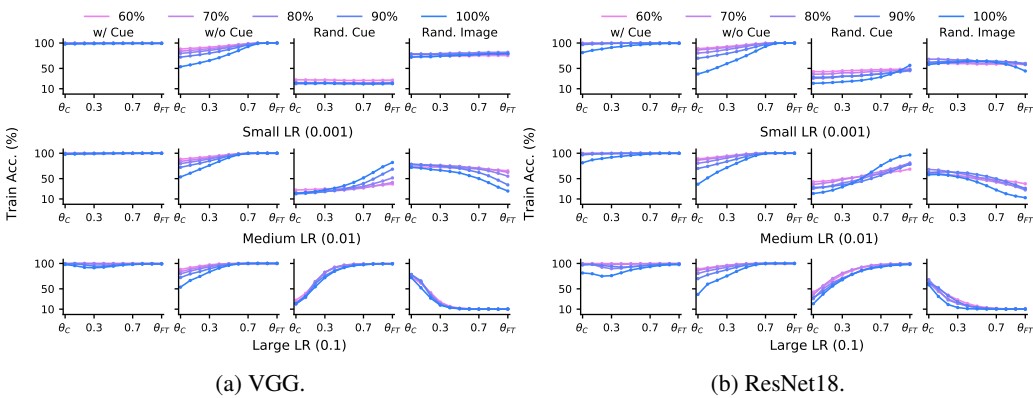

Figure 23: **Fine-tuning of models trained on Dominoes**. We plot test accuracy curves along the linear path between $\theta_C$ and $\theta_{FT}$ and see thorough corroboration of our claims in the main text: Linearly connected minimizers exhibit mechanistic similarity, behaving identically on counterfactual datasets, indicating mechanistic connectivity.

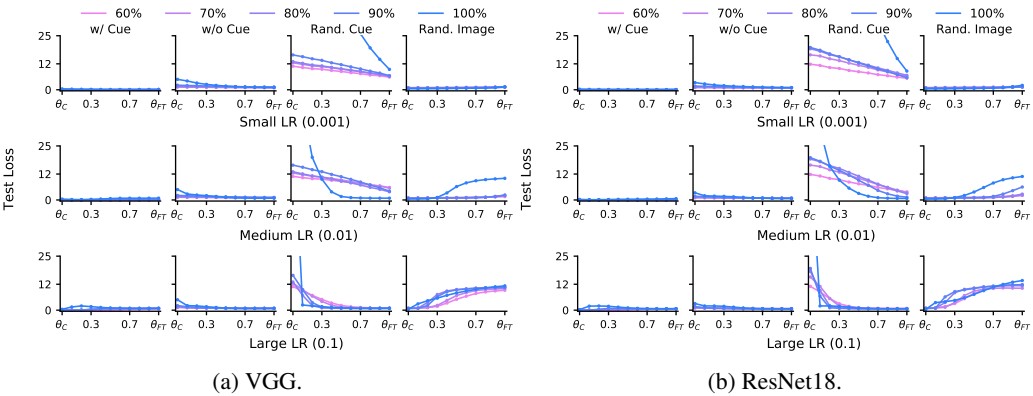

Figure 24: **Fine-tuning of models trained on CIFAR-10 with Box Cue**. We plot test loss curves along the linear path between $\theta_C$ and $\theta_{FT}$ and see thorough corroboration of our claims in the main text: Linearly connected minimizers exhibit mechanistic similarity, behaving identically on counterfactual datasets, indicating mechanistic connectivity.

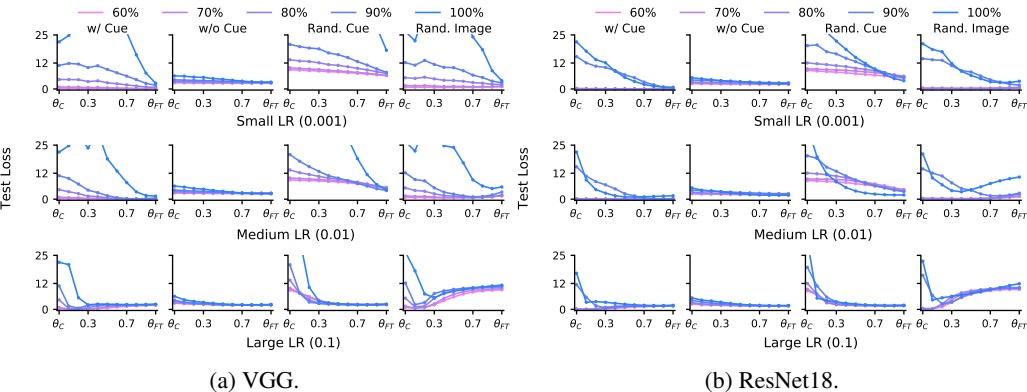

Figure 25: **Fine-tuning of models trained on CIFAR-100 with Box / Color Cue**. We plot test loss curves along the linear path between $\theta_C$ and $\theta_{FT}$ and see thorough corroboration of our claims in the main text: Linearly connected minimizers exhibit mechanistic similarity, behaving identically on counterfactual datasets, indicating mechanistic connectivity.

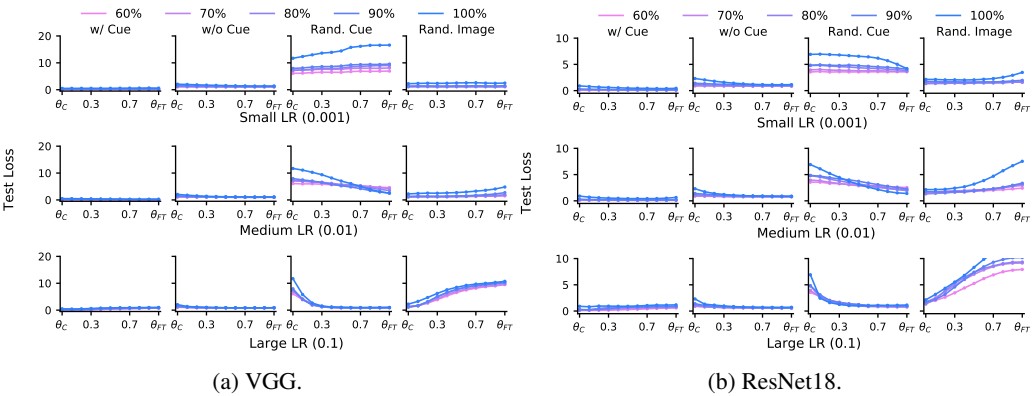

Figure 26: **Fine-tuning of models trained on Dominoes**. We plot test loss curves along the linear path between $\theta_C$ and $\theta_{FT}$ and see thorough corroboration of our claims in the main text: Linearly connected minimizers exhibit mechanistic similarity, behaving identically on counterfactual datasets, indicating mechanistic connectivity.

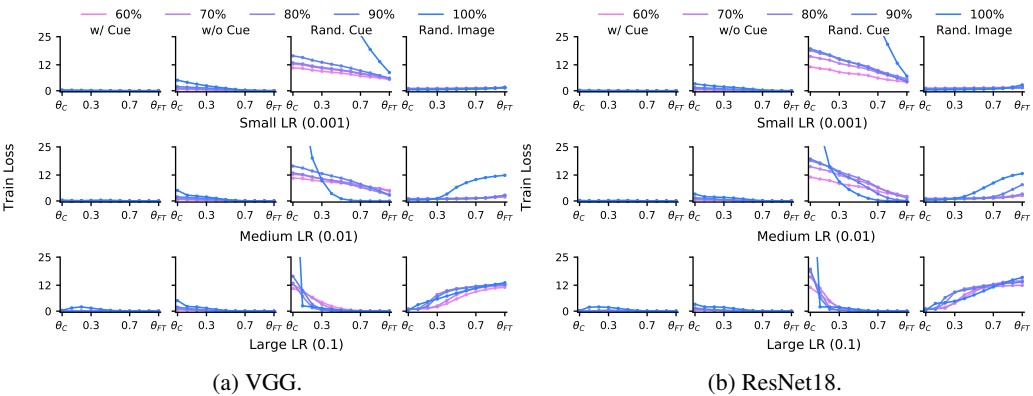

Figure 27: **Fine-tuning of models trained on CIFAR-10 with Box Cue**. We plot train loss curves along the linear path between $\theta_C$ and $\theta_{FT}$ and see thorough corroboration of our claims in the main text: Linearly connected minimizers exhibit mechanistic similarity, behaving identically on counterfactual datasets, indicating mechanistic connectivity.

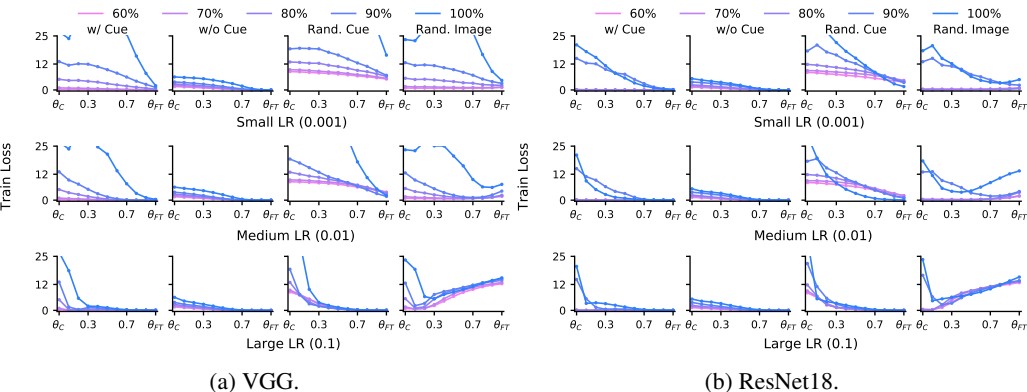

Figure 28: **Fine-tuning of models trained on CIFAR-100 with Box / Color Cue**. We plot train loss curves along the linear path between $\theta_C$ and $\theta_{FT}$ and see thorough corroboration of our claims in the main text: Linearly connected minimizers exhibit mechanistic similarity, behaving identically on counterfactual datasets, indicating mechanistic connectivity.

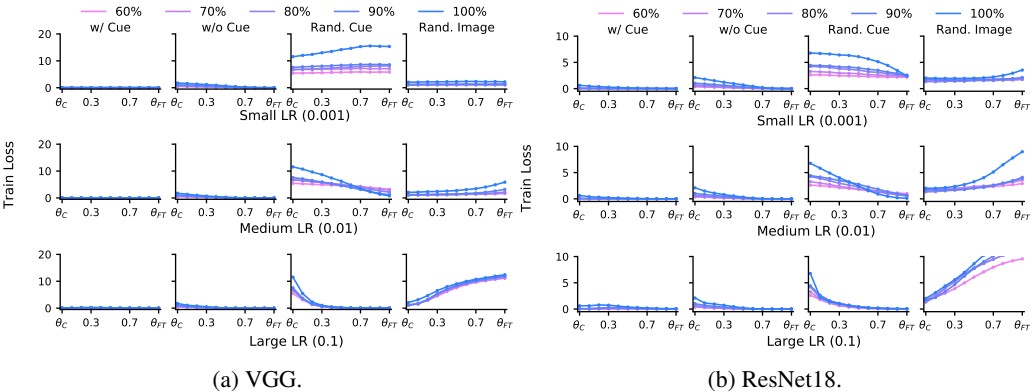

(a) VGG.

(b) ResNet18.

Figure 29: **Fine-tuning of models trained on Dominoes**. We plot train loss curves along the linear path between $\theta_C$ and $\theta_{FT}$ and see thorough corroboration of our claims in the main text: Linearly connected minimizers exhibit mechanistic similarity, behaving identically on counterfactual datasets, indicating mechanistic connectivity.

## F    DEFERRED PROOFS

### F.1    EXHAUSTIVENESS OF UNIT INTERVENTIONS

**Proposition 1.** *(Exhaustiveness of Unit Interventions.) If $f(.;\theta)$ is invariant to unit interventions $\mathcal{A}_i$ and $\mathcal{A}_j$, it must be invariant to their composition; conversely, lack of invariance to either $\mathcal{A}_i$ or $\mathcal{A}_j$ precludes invariance to their composition.*

*Proof.* Assume the set of parameters $\theta$ induces a model that exhibits invariance to the intervention $\mathcal{A}_i$. Independently, consider another intervention $\mathcal{A}_j$. Then, $f(\mathcal{E}(x;\{\mathcal{A}_i, \mathcal{A}_j\});\theta) = f(\mathcal{G}_X \circ \mathcal{A}_i \circ \mathcal{A}_j \circ \mathcal{G}_X^{-1}(x);\theta) = f(\mathcal{G}_X \circ \mathcal{A}_i \circ \mathcal{G}_X^{-1}(\mathcal{E}(x;\mathcal{A}_j));\theta) = f(\mathcal{E}(\mathcal{E}(x;\mathcal{A}_j);\mathcal{A}_i);\theta) = f(\mathcal{E}(x;\mathcal{A}_j);\theta)$, where the last equality happens due to the assumed invariance of $\mathcal{A}_i$. Now, if $\theta$ exhibits invariance to $\mathcal{A}_j$ as well, we have $f(\mathcal{E}(x;\{\mathcal{A}_i, \mathcal{A}_j\});\theta) = f(\mathcal{E}(x;\mathcal{A}_j);\theta) = f(x;\theta)$, i.e., the model induced by $\theta$ is invariant to the simultaneous operation (i.e., composition) of $\mathcal{A}_i$ and $\mathcal{A}_j$. Meanwhile, if $\theta$ is invariant $\mathcal{A}_i$ but not to $\mathcal{A}_j$, we have $f(\mathcal{E}(x;\{\mathcal{A}_i, \mathcal{A}_j\});\theta) = f(\mathcal{E}(x;\mathcal{A}_j);\theta) \neq f(x;\theta)$, i.e., $\theta$ induces a model that lack invariance to the simultaneous operation (i.e., composition) of $\mathcal{A}_i$ and $\mathcal{A}_j$.

Note that the derivation above did not rely on the fact that the interventions are "unit" in the sense that they act on independent dimensions. However, if one considers general interventions that can act on multiple dimensions of the latent space simultaneously, then a given intervention can undo the effects of another. For example, assume a model is not invariant to unit interventions on a dimension that rotates an object, but are invariant to unit interventions on all other latent dimensions. Then, if two general interventions involve operation on this latent dimension, they can make an object rotate by equal and opposite angles, while changing some other dimensions of the latent state that the model is invariant to. In this case, the interventions end up undoing their effect, and the overall state change does not yield any influence on the model output. By assuming unit interventions that enforce transformations on specific dimensions, we circumvent this failure mode.                            $\square$

### F.2    MODE CONNECTIVITY OF MECHANISTICALLY DISSIMILAR MODELS

We first repeat the following result from prior work (paraphrased per our notations and setup).

**Lemma 1.** *(Simsek et al., 2021). Consider an $L$-layer network $f(.;\theta)$, whose activation function $\phi$ satisfies $\phi(0) \neq 0$, $\phi^{(n)} \neq 0$ for infinitely many odd and even values of $n$, where $\phi^{(n)}$ denotes the $n^{th}$ derivative of $\phi$. Let $r_1^*, r_2^*, \ldots, r_L^*$ be the minimum number of neurons needed in layers 1 to $L$ for achieving zero error (cross-entropy or mean-square error) on a dataset $\mathcal{D}$ and call a network overparameterized if for all layers $l$, it contains number of neurons $r_l > r_l^*$. Then, under overparameterization, there always exists a continuous, zero-loss path that connects two minimizers.*

The result above involves showing permutation symmetry of neural networks yields a single continuous manifold of zero loss, and then proving all parameters that yield zero-loss lie on this manifold. We highlight the amount of overparameterization needed for the claim's validity is rather mild, i.e., just one additional neuron per layer. Also note that while the proof makes assumptions on the analyticity of the activation function used, this constraint is only mandatory for ease of theoretical analysis. Moreover, continuous approximations to ReLU exist which satisfy these assumptions. For example, $\phi(x) = \phi_{\text{softplus}}(x) + \phi_{sigmoid}(4x)$, where $\phi_{\text{softplus}}(x) = \ln(1 + \exp(x))$ and $\phi_{\text{sigmoid}}(x) = 1/1+\exp(-x)$. Similar result was also shown by Nguyen (2019), who demonstrates networks with a pyramidal structure, i.e., networks for which the width of any given layer is less than or equal to its preceding layers, satisfy Lemma 1.

Our claim on mode connectivity of mechanistically dissimilar models now follows as a corollary.

**Proposition 2.** *(Mode Connectivity of Mechanistically Dissimilar Models.)* *Assume $\theta_1, \theta_2$ are minimizers of the loss on a dataset $\mathcal{D}$ and induce mechanistically dissimilar models. Given sufficient overparameterization, there exists a continuous path along which the minimizers are mode connected.*

*Proof.* By definition, $\mathcal{L}(f(\mathcal{D}; \theta)) = 0$ for $\theta \in \theta_1, \theta_2$. Since the distribution of data plays no role in the proof of Lemma 1, the result must hold for two minimizers that rely on entirely disparate mechanisms (e.g., background vs. shape) for achieving zero-loss on a dataset $\mathcal{D}$. The claim then directly follows as a corollary of Lemma 1, assuming the model is overparameterized in the sense defined there and the loss is either cross-entropy or mean-square error. □

### F.3 LACK OF LINEAR CONNECTIVITY IMPLIES MECHANISTIC DISSIMILARITY

**Conjecture 1.** *(Lack of Linear Connectivity implies Mechanistic Dissimilarity.)* *If two minimizers $\theta_1$ and $\theta_2$ of the loss $\mathcal{L}(f(\mathcal{D}; \theta))$ on a dataset $\mathcal{D}$ cannot be linear mode-connected (up to permutations of neurons), their corresponding models $f(.; \theta_1), f(.; \theta_2)$ must be mechanistically dissimilar.*

Neural networks boast the well-known permutation symmetry phenomenon: permuting neurons, while accounting for the fan-in and fan-out weights, yields a model that is functionally the same (Hecht-Nielsen, 1990). That is, after permutation, the model encodes the exact same function as the original model. To avoid this degeneracy, we will assume that we are analyzing two minimizers $\theta_1, \theta_2$ that necessarily are *not* permutations of each other. In practice, one can run recent methodologies on "neural alignment" to ensure this assumption is valid (Ainsworth et al., 2022; Singh & Jaggi, 2020).

*Proof.* As per Prop. 1, we need only establish invariance to unit interventions for characterizing the mechanisms underlying a model's decision rules and, correspondingly, ascertain mechanistic similarity between two model parameterizations. To that end, we consider a unit intervention $\mathcal{A}_i$ that we assume the minimizer $\theta_1$ is invariant to. We will analyze the loss of the model parameterized with linear interpolation of $\theta_1, \theta_2$ on a counterfactual sample $\mathcal{E}(x; \mathcal{A}_i)$ generated using intervention $\mathcal{A}_i$. For brevity, we denote the latent state of $z$ as $z = \mathcal{G}_X^{-1}(x)$; correspondingly, we denote the intervened latent state as $\mathcal{A}_i^{\alpha_i}(z) = z + \Delta z$, where $\Delta z$ is 0 in all but the $i^{\text{th}}$ dimension, where it is equal to $\Delta z_i = \alpha_i$. We can thus write: $\mathcal{E}(x; \mathcal{A}_i^{\alpha_i}) = \mathcal{G}_X \circ \mathcal{A}_i^{\alpha_i} \circ \mathcal{G}_X^{-1}(x) = \mathcal{G}_X(\tilde{z}) = \mathcal{G}_X(z + \Delta z)$.

We now consider the parameterization along a general path $\gamma_{\theta_1 \to \theta_2}(t)$ such that $\gamma_{\theta_1 \to \theta_2}(0) = \theta_1$ and $\gamma_{\theta_1 \to \theta_2}(1) = \theta_2$. We assess its loss on the counterfactual data via a second-order expansion along the data-generating process:

$$
\begin{aligned}
&L\left(f\left(\mathcal{E}\left(x; \mathcal{A}_i^{\alpha_i}\right); \gamma_{\theta_1 \to \theta_2}(t)\right)\right) \\
&= L\left(f\left(\mathcal{G}_X\left(z + \Delta z\right); \gamma_{\theta_1 \to \theta_2}(t)\right)\right), \\
&= L\left(f\left(\mathcal{G}_X\left(z\right); \gamma_{\theta_1 \to \theta_2}(t)\right)\right) + \left(\Delta z\right)^T \nabla_z L\left(f\left(\mathcal{G}_X\left(z\right); \gamma_{\theta_1 \to \theta_2}(t)\right)\right) \\
&\qquad\qquad\qquad + \frac{1}{2}\left(\Delta z\right)^T \nabla_z^2 L\left(f\left(\mathcal{G}_X\left(z\right); \gamma_{\theta_1 \to \theta_2}(t)\right)\right)\left(\Delta z\right) + \mathcal{O}(\alpha_i^3), \quad (7) \\
&\approx L\left(f\left(\mathcal{G}_X\left(z\right); \gamma_{\theta_1 \to \theta_2}(t)\right)\right) + \alpha_i \frac{\partial}{\partial z_i} L\left(f\left(\mathcal{G}_X\left(z\right); \gamma_{\theta_1 \to \theta_2}(t)\right)\right) \\
&\qquad\qquad\qquad + \frac{1}{2}\left(\alpha_i\right)^2 \frac{\partial^2}{\partial z_i^2} L\left(f\left(\mathcal{G}_X\left(z\right); \gamma_{\theta_1 \to \theta_2}(t)\right)\right).
\end{aligned}
$$

The parameterization along a general path connecting the two minimizers $\theta_1, \theta_2$ can be written in the following form: $\gamma_{\theta_1 \to \theta_2}(t) = \theta_1 + \Delta\theta(t, 1)$, where $\Delta\theta(t, 1) = \gamma_{\theta_1 \to \theta_2}(t) - \theta_1$. Then, expanding the loss achieved by the model with this parameterization on the original data up to second-order along the change in parameters, we get the following.

$$
\begin{aligned}
L\left(f\left(\mathcal{G}_X(z); \gamma_{\theta_1 \to \theta_2}(t)\right)\right) &= L\left(f\left(\mathcal{G}_X(z); \theta_1 + \Delta\theta(t, 1)\right)\right) \\
&= L\left(f\left(\mathcal{G}_X(z); \theta_1\right)\right) + \left(\Delta\theta(t, 1)\right)^T \nabla_\theta L\left(f\left(\mathcal{G}_X(z); \theta_1\right)\right) \\
&\quad + \frac{1}{2}\left(\Delta\theta(t, 1)\right)^T \nabla_\theta^2 L\left(f\left(\mathcal{G}_X(z); \theta_1\right)\right)\left(\Delta\theta(t, 1)\right) + \mathcal{O}(\|\Delta\theta(t, 1)\|^3), \\
&\approx \frac{1}{2}\left(\Delta\theta(t, 1)\right)^T \nabla_\theta^2 L\left(f\left(\mathcal{G}_X(z); \theta_1\right)\right)\left(\Delta\theta(t, 1)\right),
\end{aligned}
\tag{8}
$$

where the loss and the gradient term can be ignored because $\theta_1$ is a minimizer of the loss on dataset $\mathcal{D}$. Now, substituting Equation 8 into Equation 7, we get the following.

$$
\begin{aligned}
&L\left(f\left(\mathcal{E}\left(x; \mathcal{A}_i^{\alpha_i}\right); \gamma_{\theta_1 \to \theta_2}(t)\right)\right) \\
&= L\left(f\left(\mathcal{G}_X(z); \gamma_{\theta_1 \to \theta_2}(t)\right)\right) + \alpha_i \frac{\partial}{\partial z_i} L\left(f\left(\mathcal{G}_X(z); \gamma_{\theta_1 \to \theta_2}(t)\right)\right) + \frac{1}{2}\left(\alpha_i\right)^2 \frac{\partial^2}{\partial z_i^2} L\left(f\left(\mathcal{G}_X(z); \gamma_{\theta_1 \to \theta_2}(t)\right)\right), \\
&= \frac{1}{2}\Delta\theta(t, 1)^T \nabla_\theta^2 L\left(f\left(\mathcal{G}_X(z); \theta_1\right)\right)\Delta\theta(t, 1) \\
&\quad + \alpha_i \frac{\partial}{\partial z_i}\left(\frac{1}{2}\Delta\theta(t, 1)^T \nabla_\theta^2 L\left(f\left(\mathcal{G}_X(z); \theta_1\right)\right)\Delta\theta(t, 1)\right) \\
&\quad + \frac{1}{2}\left(\alpha_i\right)^2 \frac{\partial^2}{\partial z_i^2}\left(\frac{1}{2}\Delta\theta(t, 1)^T \nabla_\theta^2 L\left(f\left(\mathcal{G}_X(z); \theta_1\right)\right)\Delta\theta(t, 1)\right), \\
&= \frac{1}{2}\Delta\theta(t, 1)^T \nabla_\theta^2 \left[L\left(f\left(\mathcal{G}_X(z); \theta_1\right)\right) + \alpha_i \frac{\partial}{\partial z_i} L\left(f\left(\mathcal{G}_X(z); \theta_1\right)\right) + \frac{1}{2}\left(\alpha_i\right)^2 \frac{\partial^2}{\partial z_i^2} L\left(f\left(\mathcal{G}_X(z); \theta_1\right)\right)\right]\Delta\theta(t, 1), \\
&= \frac{1}{2}\Delta\theta(t, 1)^T \nabla_\theta^2 \left[L\left(f\left(\mathcal{G}_X(z); \theta_1\right)\right) + \left(\Delta z\right)^T \nabla_z L\left(f\left(\mathcal{G}_X(z); \theta_1\right)\right) + \frac{1}{2}\left(\Delta z\right)^T \nabla_z^2 L\left(f\left(\mathcal{G}_X(z); \theta_1\right)\right)\left(\Delta z\right)\right]\Delta\theta(t, 1), \\
&\approx \frac{1}{2}\Delta\theta(t, 1)^T \nabla_\theta^2 \left[L\left(f\left(\mathcal{G}_X(z + \Delta z); \theta_1\right)\right)\right]\Delta\theta(t, 1), \\
&= \frac{1}{2}\Delta\theta(t, 1)^T \nabla_\theta^2 \left[L\left(f\left(\mathcal{E}\left(x; \mathcal{A}_i^{\alpha_i}\right); \theta_1\right)\right)\right]\Delta\theta(t, 1), \\
&= \frac{1}{2}\Delta\theta(t, 1)^T \nabla_\theta^2 \left[L\left(f\left(x; \theta_1\right)\right)\right]\Delta\theta(t, 1),
\end{aligned}
\tag{9}
$$

where the last equality follows because of the assumed invariance of $\theta_1$ to the intervention $\mathcal{A}_i^{\alpha_i}$. Now, if the connectivity path $\gamma_{\theta_1 \to \theta_2}(t)$ is not linear, then there exists an interpolation along the linear path connecting minimizers $\theta_1, \theta_2$ that has a loss higher than the two minimizers. That is, the displacement vector $\Delta\theta(t, 1)$ does not lie in the null-space of the Hessian and $\Delta\theta(t, 1)^T \nabla_\theta^2 \left[L\left(f\left(x; \theta_1\right)\right)\right]\Delta\theta(t, 1) \neq 0$. Substituting this relation into Equation 9, we get $L\left(f\left(\mathcal{E}\left(x; \mathcal{A}_i^{\alpha_i}\right); \gamma_{\theta_1 \to \theta_2}(t)\right)\right) \neq 0$.

$\square$

### F.4 PROVING CONJECTURE 1 IN A SIMPLIFIED SETTING

**Conjecture 1.** *(Lack of Linear Connectivity implies Mechanistic Dissimilarity.) If two minimizers $\theta_1$ and $\theta_2$ of the loss $\mathcal{L}(f(\mathcal{D}; \theta))$ on a dataset $\mathcal{D}$ cannot be linear mode-connected (up to permutations of neurons), their corresponding models $f(.; \theta_1), f(.; \theta_2)$ must be mechanistically dissimilar.*

As we show next, the conjecture above can be proven in a simplified setting.

**Model Setup:** We consider a binary classification task on a dataset $\mathcal{D} = \{x_i, y_i\}_{i=1}^M$, where $x_i \in \mathbb{R}^D$, $y \in \mathcal{Y} = \{0, 1\}$, and $M$ is the number of samples. The model is a two-layer, fully connected network defined as follows: $f(x; W) = \frac{1}{N}\mathbf{1}^T \phi(W^T x)$. Here, $W \in \mathbb{R}^{D \times N}$ denotes the hidden layer with

$N$ neurons, $\mathbf{1} \in \mathbb{R}^N$ is an all ones vector, and $\phi(.)$ is the ReLU activation function. The model is trained to minimize a loss $\mathcal{L}(f(\mathcal{D}; W)) = \frac{1}{M} \sum_{i=1}^M l(y_i, f(x_i; W))$, where $l(.,.)$ denotes the sample-wise loss whose global minimizer yields $y_i = f(x_i; W)$ for all $x_i \in \mathcal{D}$. This property is satisfied by several loss functions, e.g., mean-square error, L-1 loss, hinge loss, etc. We assume all minimizers are global and interpolating, i.e., they achieve zero loss, a property known to be true for overparameterized neural networks (Kawaguchi, 2016; Kawaguchi & Kaelbling, 2020; Nguyen et al., 2018; Nguyen & Mondelli, 2020; Arora et al., 2019). This implies if $W_*$ is a minimizer, $\forall i \in [M], y_i = f(x_i; W_*) = \frac{1}{N} \mathbf{1}^T \phi(W_*^T x_i)$.

We next describe the data-generating process that we will focus on in the following discussion.

**Data-Generating Process:** We consider a data-generating process with multiple predictive attributes of different complexity, inspired by the one proposed by Shah et al. (2020).

Consider a non-negative even integer $K$. Define the sets $S_0(K)$ and $S_1(K)$ that respectively include odd and even integers between $[-\frac{K}{2}, \frac{K}{2}]$. We use sign$(.)$ to denote the sign function, which outputs 1 if $x > 0$, 0 if $x = 0$, and $-1$ if $x < 0$. Unif$(S)$ denotes a uniform distribution over the set $S$. We define a randomized process $s$, such that $s_K(0) \sim \text{Unif}(S_0(\frac{K}{2}))$, $s_K(1) \sim \text{Unif}(S_1(\frac{K}{2}))$. Correspondingly, given a margin parameter $\delta \in [0, 0.5]$, we define the randomized function $T_K(z) : \{0, 1\} \to \mathbb{R}$ as follows.

$$T_K(z) := \begin{cases} \frac{\sqrt{3}}{\sqrt{D}}(z - \epsilon \operatorname{sign}(z)), & \text{where } \epsilon \sim \text{Unif}([0, 2\delta]), \text{ if } K = 0, \\ \frac{2\sqrt{3}}{K\sqrt{D}}(s_K(z) + \epsilon), & \text{where } \epsilon \sim \text{Unif}([-\delta, \delta]), \text{ if } K \geq 1, |s_K(z)| \neq \frac{K}{2}, \\ \frac{2\sqrt{3}}{K\sqrt{D}}(s_K(z) - \epsilon \operatorname{sign}(z)), & \text{where } \epsilon \sim \text{Unif}([0, \delta]), \text{ if } K \geq 1, |s_K(z)| = \frac{K}{2}. \end{cases}$$
(10)

Note that $T_K(z)$ produces a zero-mean output with variance $1/D$. The margin $\delta$ allows us to draw infinite samples from the function. More importantly, $T_K(z)$ is left-invertible, i.e., given its output, we can infer $z$. Correspondingly, if $z$ defines the target label $y$, inverting the attribute $T_K(z)$ will allow us to solve a classification task defined on this attribute. However, this inversion process requires inference of $K$ piece-wise linear splines to model the optimal decision boundaries (see Fig. 30). The scalar $K$ can thus can be considered a measure of the complexity of the attribute, inline with prior work on simplicity bias in neural networks (Nakkiran et al., 2019; Shah et al., 2020; Valle-Perez et al., 2018; Scimeca et al., 2021; Hu et al., 2020). For example, if $K = 0$, the attribute is linearly separable and of least complexity. This notion of complexity is particularly natural for studying neural networks with ReLU activations because each neuron in such a model represents a spline function and several such neurons can approximate complex decision boundaries by representing them with such piece-wise spline functions (Balestriero et al., 2018; Balestriero & Baraniuk, 2018; Balestriero, 2017; Wang et al., 2018).

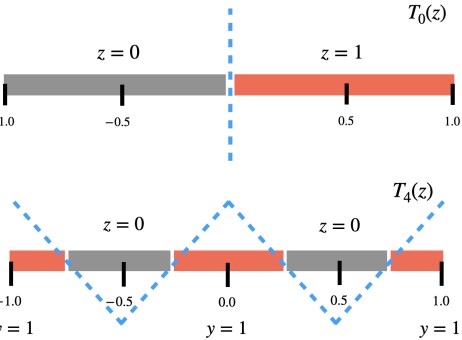

Figure 30: **K-Complex Outputs.** We illustrate the output range of randomized function $T_K(z)$ (Eq. 10) for $K = 0$ (top) and $K = 4$ (bottom). Note the input $z \in \{0, 1\}$ (shown grey, red respectively) deterministically controls the output values. Thus, if $y = z$, the output $T_K(z)$ is perfectly predictive of the target label. However, inverting this function requires inference of $K$ piece-wise linear splines to model the optimal decision boundaries (shown blue, dotted lines). Increasing the value of $K$ makes the task consequently harder.

The overall data-generating process $\mathcal{G}(Z)$ transforms the $n$-dimensional random variable $Z$ from the latent space $z_1 \times z_2 \times \cdots \times z_n \in \{0, 1\}^n$ to produce samples with $n$ attributes of different complexities $T_{K_1}(z_1), T_{K_2}(z_2), \ldots, T_{K_n}(z_n)$ and appends $D - n$ noisy attributes, sampled from a symmetric, zero-mean distribution $\mathcal{V}$ with variance $\frac{1}{d}$; e.g., the Gaussian distribution $\mathcal{N}(0, 1/\sqrt{D})$ or uniform distribution $\mathcal{U}\left(-\sqrt{3/D}, \sqrt{3/D}\right)$. Correspondingly, generation of a sample $(x, y)$ can be represented as follows:

$$(x, y) := \mathcal{G}(Z) = [T_{K_1}(z_1), T_{K_2}(z_2), \ldots, T_{K_n}(z_n), \nu_1, \nu_2, \ldots, \nu_{d-n}]^T,$$
(11)

where $\nu_i \sim \mathcal{V}$ for all $i \in \{1, 2, \ldots, D-n\}$. In the above, the target label $y$ is assumed to be generated by the function $\mathcal{G}_y(.) = T_{K_i}^{-1}(.)$, which inverts the attribute $T_{K_i}(z_i)$ that is assumed to define the label. Note again another attribute, say $T_{K_j}(z_j)$, will also be predictive of the label if the corresponding latent, $z_j$, is correlated with $z_i$. This is similar to putting a correlated *cue* attribute in the data, as was done in our experiments in the main paper. Thus, while the process above is relatively simplified, it is a valid abstraction of our empirical setup from the paper. In the following, we will consider perfectly predictive attributes, i.e., we do not assume partial correlation.

We next define the notion of an *activation pattern* and *matching of two models in their activation patterns*. In the following, $\Sigma_N$ denotes the permutation group of order $N$ over the set $\{1, 2, \ldots, N\}$ (Bronstein et al., 2021).

**Definition 6. (Activation Pattern.)** *The activation pattern of a model $f(.; W)$ on input $x$ is defined as the vector $\phi'(W^T x)$ whose elements are indicator variables denoting whether the $j^{th}$ hidden neuron is activated for input $x$, i.e., $\phi'(W^T x)[j] = 1\left(W_j^T x > 0\right)$.*

Note that $\phi(W^T x) = \phi'(W^T x) \odot (W^T x)$, where $\odot$ denotes the element-wise product, if $\phi(.)$ is the ReLU function.

**Definition 7. (Matching in Activation Patterns.)** *Consider a dataset $\mathcal{D}$ and two models $f(.; W_1)$ and $f(.; W_2)$. We call the models matching in activation patterns on dataset $\mathcal{D}$ if there exists a permutation $\pi \in \Sigma_N$ that rearranges neurons of $f(.; W_2)$ to match $f(.; W_1)$'s activation patterns, i.e., $\phi'(W_1^T x) = \phi'(\pi(W_2)^T x)$ for all $x \in \mathcal{D}$.*

Next, we establish the relationship between two models' activation patterns and linear mode connectivity.

**Lemma 2. (Alignment Constraint for Linear Mode Connectivity.)** *If minimizers $W_\alpha$ and $W_\beta$ exhibit linear mode connectivity on dataset $\mathcal{D}$, then the models $f(.; W_\alpha), f(.; W_\beta)$ are matching in activation patterns on the dataset. That is, for all $x \in \mathcal{D}$, we have $\phi'(W_\alpha^T x) = \phi'(W_\beta^T x)$.*

*Proof.* Note that linear mode connectivity is a translation invariance property of the loss in the parameter space. Since we assume interpolating minimizers, this invariance extends to model predictions. Consequently, the derivative of the model prediction along the linear path $\gamma_{W_\alpha \to W_\beta}(t) = W(t) = W_\alpha + t(W_\beta - W_\alpha)$ is zero; that is, $\frac{\partial}{\partial t} f(x; W(t)) = 0$. This implies,

$$\frac{\partial}{\partial t} \mathbf{1}^T \phi(W(t)^T x) = \phi'(W(t)^T x)^T (W_\beta - W_\alpha)^T x = 0. \tag{12}$$

Substituting $t = 1$ in Eq. 12, we get,

$$\phi'(W_\beta^T x)^T (W_\alpha^T x) = \phi'(W_\beta^T x)^T (W_\beta^T x) = \mathbf{1}^T (\phi'(W_\beta^T x) \odot \phi(W_\beta^T x)) = \mathbf{1}^T \phi(W_\alpha^T x). \tag{13}$$

This implies,

$$(\phi'(W_\alpha^T x) - \phi'(W_\beta^T x))^T (W_\alpha^T x) = 0. \tag{14}$$

Next, we define the following vector.

$$\mathbf{1}_{(\alpha_+ \beta_-)} = \begin{cases} 1, & \text{if } W_\alpha^T x > 0 \text{ and } W_\beta^T x \le 0; \\ 0, & \text{otherwise.} \end{cases} \tag{15}$$

Define the vector $\mathbf{1}_{(\alpha_- \beta_+)}$ in a similar manner. Then, it is easy to see that

$$\phi'(W_\alpha^T x) - \phi'(W_\beta^T x) = \mathbf{1}_{(\alpha_+ \beta_-)} - \mathbf{1}_{(\alpha_- \beta_+)}. \tag{16}$$

Substituting the above relationship in Eq. 14 gives

$$\mathbf{1}_{(\alpha_+ \beta_-)}^T (W_\alpha^T x) = \mathbf{1}_{(\alpha_- \beta_+)}^T (W_\alpha^T x). \tag{17}$$

Note that in the above equation, the left-hand side is a sum of positive reals, while the right-hand side is a sum of negative reals. That is, the equality cannot hold unless both are equal to zero for all $x \in \mathcal{D}$. This implies $\mathbf{1}_{(\alpha_+ \beta_-)} = \mathbf{1}_{(\alpha_- \beta_+)} = \mathbf{0}$. That is, there is no neuron in model $f(.; W_\alpha)$ that is active while the corresponding index neuron in $f(.; W_\beta)$ is inactive. Consequently, for linear mode connectivity to hold, the neurons at the same index in the two models should activate/inactivate together for any given sample, hence producing the same set of activation patterns. This completes the proof. $\square$

Table 5: **Illustrating Simplicity Bias.** We train models on a dataset with predictive attributes of complexities 0 and 4. Column titles indicate which attributes were allowed to remain predictive during training, i.e., were not randomized via interventions: e.g., $K_1 = 0$ implies only the linearly separable attribute is predictive in the training data. Rows report difference in loss on a test dataset $\mathcal{D}_{K_1}$ which contains attributes of complexity $K_1$ and another test dataset $\mathcal{D}_{K_2}$ which contains attributes of complexity $K_2$. Results are computed up to 4 digits of precision and averaged over 3 seeds. We see models trained on data with both predictive attributes behave similarly to models trained on $K = 0$ attribute only; that is, they are invariant to the more complex attribute for which $K = 4$.

| Complexity of Train Attribute | $K_1 = 0$ | | $K_1 = 4$ | | $K_1 = 0, 4$ | |
|---|---|---|---|---|---|---|
| Complexity of Test Attribute | $K_2 = 0$ | $K_2 = 4$ | $K_2 = 0$ | $K_2 = 4$ | $K_2 = 0$ | $K_2 = 4$ |
| $\lvert \mathcal{L}(f(\mathcal{D}_{K_1}; W)) - \mathcal{L}(f(\mathcal{D}_{K_2}; W)) \rvert$ | 0.0 | 22.79 | 26.31 | 0.0 | 0.0 | 18.84 |

**Remark 1.** *(Lemma 2 highlights why neurons must be permuted for linear mode connectivity.) The lemma above shows that if two models produce the same activation patterns, the models are "effectively linear" with respect to each other. This enables linear interpolation of the two models without increasing error. We also highlight that if two models produce activation patterns that are a permutation of each other (e.g., this can happen if their initializations were permutations of each other), then un-permuting them will make the models linear mode connected. Thus, Lemma 2 is inherently an alignment constraint and can also be regarded as a precise condition under which the conjecture by Entezari et al. (2021) holds: models trained independently on a given dataset using gradient descent are linear mode connected. Even though the result above was shown for a two-layer model, it is easy to see that a more general statement is true: if two minimizers induce models that produce the same activation patterns, then there exists a permutation of neurons under which the two models can be linear mode connected.*

**Remark 2.** *(Wasserstein-1 distance between activation patterns serves as proxy for when linear mode connectivity will hold.) The activation pattern of a model for a given sample is a vector of binary variables. Thus, the difference between two activation patterns $\phi'(W_\alpha^T x)$ and $\phi'(W_\beta^T x)$ can be computed by simply comparing their means $\frac{1}{N}\lvert \mathbf{1}^T \phi'(W_\alpha^T x) - \mathbf{1}^T \phi'(W_\beta^T x) \rvert$, which is in fact the Wasserstein-1 distance between two Bernoulli distribution for which $p = \frac{1}{N}\lvert \mathbf{1}^T \phi'(W^T x) \rvert$. This value $p$ can be regarded as the probability a neuron in the model is activated. Correspondingly, when the Wasserstein-1 distance between two activation patterns is low, we can expect that there exists a permutation of neurons that allows the two models to be linear mode connected. The W-1 distance can thus be regarded as a proxy for assessing whether two models can be linear mode connected. Further, we highlight that even though this result is derived for a specific model architecture, it is actually quite general: any two models with zero W-1 distance must be linear mode connectable (up to permutations) because their activation patterns will necessarily be the same.*

Next, we rephrase the result on simplicity bias of neural networks by Shah et al. (2020); Valle-Perez et al. (2018); Nakkiran et al. (2019); Scimeca et al. (2021) using the notations defined in this paper.

**Lemma 3.** *(Simplicity Bias.) Assume a data-generating process $\mathcal{G}$ produces $n$ perfectly predictive attributes with respective complexities $[K] = \{K_1, K_2, \ldots, K_n\}$. Let $m$ be the index of the latent corresponding to the simplest attribute, i.e., $m := \arg\min [K]$. If $W$ is a minimizer identified using gradient descent on a dataset that contains IID samples retrieved from $\mathcal{G}$, then the corresponding model $f(.; W)$ will be invariant to unit interventions on all but the latents of the simplest predictive attribute, i.e., $\mathcal{I}(W) = \{\mathcal{A}_i : i \neq m\}$.*

Thus, even if a dataset contains multiple predictive attributes, minimizers identified using gradient descent induce models that only utilize the simplest attributes for making their predictions.

We provide empirical demonstration of this claim in Tab. 5. Specifically, we train models using SGD for our assumed $f(.; W)$ architecture, using 512 neurons in the hidden layer. We sample 50000, 128-dimensional inputs from the data-generating process discussed in Eq. 11 with $K = 0$ and $K = 4$ complex predictive attributes present in the dataset. We analyze three training scenarios: (i) when only $K = 0$ attribute is allowed to be predictive and the $K = 4$ attribute is randomized via interventions; (ii) when only $K = 4$ attribute is allowed to be predictive and the $K = 0$ attribute is randomized via interventions; and (iii) when both attributes are allowed to be predictive. Evaluation involves assessing invariance of the trained model to the two predictive attributes by computing loss on a test

dataset that contains both predictive attributes and a counterfactual variant of the dataset for which either (a) $K = 0$ or (b) $K = 4$ complexity attributes have been randomized via interventions. If loss remains the same, the model is invariant to interventions on the attribute of that complexity, thus implying the model has not learned a mechanism to identify that attribute. Results are shown in Tab. 5. We see intervening on the $K = 0$ attribute yields an increase in loss in scenarios (i), (iii), indicating those models have learned a mechanism to identify that attribute. Meanwhile, intervening on the $K = 4$ attribute yields an increase in loss only in scenario (ii), indicating the models trained from the other two scenarios are invariant to the $K = 4$ complex attribute. While this is expected for scenario (i), the fact that this behavior emerges for the scenario (iii), where both predictive attributes can be used for training, is a consequence of simplicity bias of SGD.

Now consider a setting where two models make their predictions using different simplest predictive attributes from a dataset containing multiple predictive attributes. Then, if two such models rely on attributes of different complexities, we can be certain they produce different activation patterns.

**Lemma 4.** *(Disparate Complexity of Mechanisms Disallows Matching in Activations). Consider an IID sampled dataset $\mathcal{D}_{\alpha,\beta}$ from a data-generating process that produces predictive attributes $T_{K_\alpha}(.), T_{K_\beta}(.)$, where, without loss of generality, $K_\alpha > K_\beta$. Let $W_\alpha$ denote a minimizer of the loss $\mathcal{L}(f(\mathcal{D}_{\alpha,\beta}; W))$ and assume its induced model relies on $T_{K_\alpha}(.)$ for making its predictions; similarly define $W_\beta$. Then, there exists no permutation $\pi \in \Sigma_N$ such that $f(.; W_\alpha)$ and $f(.; \pi(W_\beta))$ are matching in activation patterns on $\mathcal{D}_{\alpha,\beta}$.*

*Proof.* The claim follows via contradiction. Assume a permutation $\pi$ exists such that the two models are matching in activation patterns on $\mathcal{D}_{\alpha,\beta}$. Denote the weights of the $i^{\text{th}}$ neuron in $W_\alpha$ via $W_\alpha^i$. Then $W_\alpha^i, \pi(W_\beta)^i$ are the weights of the neurons matched via $\pi$. Since using the attribute $T_{K_\alpha}(.)$ for predicting the label corresponds to inference of $2K_\alpha$ piece-wise spline functions, the probability the $i^{\text{th}}$ neuron with weights $W_\alpha^i$ will be activated for an IID sampled input $x$ from the data-generating process is $\frac{1}{2K_\alpha}$. However, since $K_\alpha \neq K_\beta$, the neuron with weights $\pi(W_\beta)^i$ does not activate with the same probability. That is, there exist samples for which $W_\alpha^i$ is activated, but $\pi(W_\beta)^i$ is not. This contradicts our assumption that there exists a permutation that allows matching in activation patterns for the two models. $\square$

Combining the results above, we have the following theorem.

**Theorem 1.** *(Disparity in Simplest Attributes Precludes Matching). Consider a dataset $\mathcal{D}$ contains multiple predictive attributes. Assume two minimizers of the loss $\mathcal{L}(f(\mathcal{D}; W))$ induce mechanistically dissimilar models that identify attributes of different complexity to make their predictions. Then, their exists no permutations of neurons for which the models exhibit linear mode connectivity.*

*Proof.* The result follows directly from the application of Lemmas 2, 3, 4. Specifically, Lemma 2 shows matching in activation patterns is required for two models to exhibit linear mode connectivity (up to permutations). Lemma 3 shows one need only analyze mechanistic dissimilarity with respect to the simplest attributes to compare the activation patterns between two models. Lemma 4 shows if two models use attributes of different complexity to make their predictions, they cannot match in activation patterns. $\square$

**Empirical Verification:** See Fig. 31.

Let us now revisit Conjecture 1 for our simplified setup. If two minimizers are not linear mode connected, their induced models must produce different activation patterns as per Lemma 2. In Theorem 1, we have shown if these models are mechanistically dissimilar such that they learn mechanisms to identify attributes of different complexity, then they will produce different activation patterns and not exhibit linear mode connectivity, partially verifying the conjecture.

However, it remains possible that two mechanistically dissimilar models learns mechanisms to identify attributes that are *different*, but have the *same complexity*. Can such minimizers possibly produce the same activation patterns on a dataset, hence exhibiting linear mode connectivity? *In theory*, it is easy to see the answer is yes; e.g., any two solutions of the linear system of equations $y = Wx$ can be interpolated regardless of their prediction mechanisms (hence the ✗* in Tab. 2). However, this constraint that the mechanisms must rely on equal complexity attributes is relatively theoretical only. For example, if $\mathcal{D}_1$ contains multiple attributes of similar complexity that allow minimizers

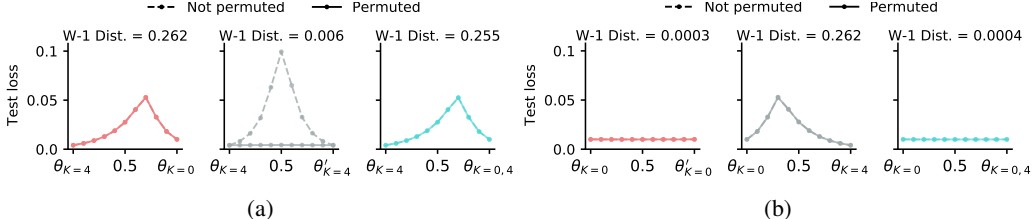

Figure 31: **Linear Connectivity and Mechanistic Similarity.** We train models on 50000 samples drawn from the data-generating process discussed in Eq. 10, allowing two predictive attributes of complexity $K = 0$ and $K = 4$. Models are also trained on counterfactual variants of this dataset, where one of the predictive attributes has been randomized via interventions. For example, $\theta_{K=0,4}$ denotes a minimizer trained on data with both attributes, while $\theta_{K=0}$ denotes a minimizer identifed via training on the counterfactual dataset that contains only the $K = 0$ predictive attribute; note $\theta_0'$ denotes use of a different initialization seed. Subsequently, we assess linear mode connectivity (before and after permutation) of models by using an evaluation dataset of 10000 samples similar to the base training dataset, i.e., both $K = 0$ and $K = 4$ predictive attributes are allowed. Plot titles denote Wasserstein-1 distance between the two models whose linear mode connectivity is being assessed. We see that models which have learned mechanisms to identify attributes of different complexity have a large Wasserstein-1 distance between their activation patterns; consequently, they cannot be linear mode connected, even after permutation of neurons. Meanwhile, models reliant on the same mechanisms have a small Wasserstein distance and can indeed be linear mode connected. For example, $\theta_{K=0}$ and $\theta_{K=0,4}$ learn the same mechanisms due to simplicity bias and can be linearly connected (see Tab. 5), but they do not exhibit linear connectivity with $\theta_{K=4}$; meanwhile, $\theta_{K=4}$ and $\theta_{K=4}'$ can be linearly connected. Note however the latter case of more complex, i.e., $K = 4$ attribute required permutations to match the neurons for linear connectivity, while the former case of linearly separable attribute did not. This behavior emerges due to the fact that all neurons learn to be *always active* for the $K = 0$ predictive attribute–see Soudry et al. (2018); Shah et al. (2020) for proof.

retrieved from $\mathcal{D}_2$ to perform well on the dataset, then given SGD (and related algorithms) force neural networks to converge to max-margin solutions, we see that minimizers retrieved via training on $\mathcal{D}_1$ will have already learned mechanisms to identify *all attributes of same complexity* (Soudry et al., 2018; Lyu & Li, 2019; Gunasekar et al., 2018; Nacson et al., 2019). In that case, use of $\mathcal{D}_2$ to create a minimizer that learns a different mechanism is practically moot, since we can already learn the relevant mechanism from $\mathcal{D}_1$ itself. Nonetheless, this hints at the idea that naïve fine-tuning can successfully alter a model's mechanisms to perform well on a target distribution when the desired mechanism is of similar complexity; otherwise, a loss barrier must be surmounted for successful learning on the target distribution. Therefore, if a model relies on spurious attributes, which are often linearly separable (Shah et al., 2020), fine-tuning will only help identify another linearly separable attribute in the target distribution. This concept can be used to analyze benefits and limitations of recent averaging-based ensemble methods (Wortsman et al., 2022b;a; Rame et al., 2022), though we leave this direction to future work.

