# OpenReview forum: "Mechanistic Mode Connectivity"
_ICLR.cc/2023/Conference — Submitted to ICLR 2023_

### Official Review · Reviewer_Y4FF · 2022-10-13

**Confidence:** 3
**Correctness:** 3
**Technical Novelty And Significance:** 3
**Empirical Novelty And Significance:** 3
**Recommendation:** 6

**Clarity, Quality, Novelty And Reproducibility:**

In terms of clarity I thoroughly enjoyed Table 1 and Figure 6 which clearly communicate the overall findings in an easy to understand manner. The work is mostly original, and I recommend releasing code so that it is reproducible.

**Strength And Weaknesses:**

Strengths: Thorough scientific investigation of mode connectivity/mechanistic mode connectivity. Definitions, propositions, and proposed methods are well motivated, and the proposed CBFT method demonstrates good empirical performance.

Weaknesses: I very much enjoyed the paper, and the following concern may reflect a bias perspective, however I thought it was still worth raising:  My main concern is with the experimental set-up. While interesting from a scientific perspective, I'm not sure that the issues addressed in this paper are interesting practically. As such, I'd recommend making this more clear in the paper. As one concrete instantiation of this concern, the paper claims: "We have demonstrated a significant limitation of naive fine-tuning: it can fail to eliminate spurious patterns learned during pre-training." However, to me the point of pre-training is to pre-train on all the data you can. We see this in LM pre-training on trillions of tokens, or recently in vision models with billion scale datasets such as LAION. Accordingly, the setting studied in this paper seems like the experimental set-up in this paper may be a bit contrived. For instance, my guess is that CBFT would actually make the model less robust if applied to fine-tuning CLIP (e.g., this may be of interest: https://arxiv.org/abs/2109.01903). In short, to me the fact that pre-training biases the fine-tuning solution seems to be a feature, not a bug, of pre-training.

**Summary Of The Paper:**

This paper studies the optimization landscape of models which have learned different mechanisms. Beyond the standard definition of linear mode connectivity, they introduce "mechanistic mode connectivity" which tests mode-connectivity under changes in the data distribution. Based on these findings they propose a fine-tuning mechanism to exit the basin associated with a spurious pre-training solution.

**Summary Of The Review:**

This paper is a thorough and interesting scientific investigation of the optimization landscape of mechanistically similar/dissimilar models. My concern is that the experimental set-up is detached from real world use cases of pre-training/fine-tuning.

---

> ### Author Response · Authors · 2022-11-13
> **Response to Reviewer Y4FF**
>
> Thank you for your positive feedback! As per your suggestion, we have the updated supplementary material to include an extensive codebase that allows creating synthetic datasets, their counterfactuals, training models, finding connectivity paths, and running CBFT. We address remaining comments below.
>
> -------
>
> ### [Main Comments / Questions]
>
> > **I very much enjoyed the paper, and the following concern may reflect a bias perspective, however I thought it was still worth raising: My main concern is with the experimental set-up. While interesting from a scientific perspective, I'm not sure that the issues addressed in this paper are interesting practically. As such, I'd recommend making this more clear in the paper. As one concrete instantiation of this concern, the paper claims: "We have demonstrated a significant limitation of naive fine-tuning: it can fail to eliminate spurious patterns learned during pre-training." However, to me the point of pre-training is to pre-train on all the data you can. We see this in LM pre-training on trillions of tokens, or recently in vision models with billion scale datasets such as LAION. Accordingly, the setting studied in this paper seems like the experimental set-up in this paper may be a bit contrived. For instance, my guess is that CBFT would actually make the model less robust if applied to fine-tuning CLIP (e.g., this may be of interest: https://arxiv.org/abs/2109.01903). In short, to me the fact that pre-training biases the fine-tuning solution seems to be a feature, not a bug, of pre-training.**
>
> We agree that our focus in this paper, i.e., to formalize the idea of mechanistic fine-tuning and analyze it via connectivity properties of mechanistically dissimilar models, was primarily scientific. We have updated the text to reflect this further.
>
> Evaluating CBFT in large-scale setting is certainly a useful suggestion and direction, but large-scale pre-training experiments would exceed the resources currently available to us. While we cannot make conclusive statements without conducting thorough experiments, we would like to provide a few reasons why CBFT might still be relevant in this setting.  First, large models may still suffer from issues of bias (e.g. gender or racial bias).  For CLIP in particular, the authors conduct evaluations illustrating such bias in Section 7 of Radford et al. 2021 [1]; see Table 6 as an example--the model is much more likely to classify people with a certain skin color as belonging to a non-human category. Second, from a theoretical point of view, arbitrarily large and diverse datasets are insufficient to guarantee that a model learns the correct causal mechanisms underlying the data generating process--the precise *distribution* of data still matters; see Remark in Krueger et al. 2021 [2].
>
>
> [1] Radford et al. 2021 (https://arxiv.org/abs/2103.00020)
>
> [2] Krueger et al. 2021 (https://arxiv.org/abs/2003.00688)
>
> ------
>
> ### [Shorter Comments / Questions]
>
> > **I recommend releasing code so that it is reproducible**
>
> Please see the updated supplementary material. The code for creating our used synthetic datasets, their counterfactuals, training models, finding connectivity paths, and CBFT has now been uploaded.
>
> -------

---

> > ### Comment · Reviewer_Y4FF · 2022-11-15
> > **Thanks.**
> >
> > Thanks for the response, I will keep my score as is.

---

### Official Review · Reviewer_iiL3 · 2022-10-25

**Confidence:** 4
**Correctness:** 3
**Technical Novelty And Significance:** 3
**Empirical Novelty And Significance:** 3
**Recommendation:** 6

**Clarity, Quality, Novelty And Reproducibility:**

The paper is clearly written, with the theoretical statements and the experimental findings being easy to understand. The authors propose an interesting novel conjecture, along with experiments on well-known models in synthetic datasets, which will be of immense benefit to the scientific community.

**Strength And Weaknesses:**

The paper introduces a novel connection between mode connectivity and a model's sensitivity to spurious features. The conjecture is in itself very interesting and will be useful for the scientific community. Moreover, the authors present experiments in synthetic datasets to support their conjecture and show a simple Taylor expansion-based proof for local correctness. To round it off, they propose novel connectivity based fine-tuning, based on the conjecture, to remove a model's dependence on spurious cues.


I have the following questions:

(a) In Figures 4 and 5, shouldn't train/evaluation loss be plotted, as opposed to the train/test accuracy because definitions 4, 5, proposition 1, and conjecture 1 include loss function? Do we still see barriers in linear paths between mechanistically dissimilar models in figures 4 and 5, with loss function?

(b) The CBFT algorithm (in 3) assumes that we have access to both a clean dataset and a dataset with statistical cues. Can the authors comment on whether one can identify examples from both groups in an unsupervised manner? If not, why should one use CBFT and not directly train the model (from scratch) on the clean examples in the dataset?

(c) It's not clear how the authors run CBFT introduced in eq. (3) on real-world models.

(i)  How do the authors optimize step (i)? Do they explicitly search over the neighborhood around $\theta_C$ for one $\theta$ that maximizes the loss function on the line connecting $\theta$ and $\theta_C$? Or do they conduct an optimization of the form $\min_{\theta, 0 \le t \le 1} | \lambda_1 - \mathcal{L}_{CE} ( \hat{y}  ( \mathcal{D}_C ;  (1-t) \theta + t \theta_C ) | $?

(ii) Furthermore, do the authors follow an alternative minimization for steps (i) and (ii)?

(iii) Do the authors include permutation in step (i)?

(d)  All statistical cues introduced in the paper are very simple (linearly separable) cues.

Will we observe the same invariances to stronger forms of statistical cues e.g. label specific translations and rotations of the images? An example is a classification setting comparing images of cats and dogs, with the images of cats rotated by 45 degrees and the images of dogs rotated by 135 degrees.

(e) The authors point out a very important characteristic of "quadratic paths" in the appendix: the evaluation performance of the model on the path will depend on the examples used to find the path, and hence may not show generalization across datasets/distributions.

However, isn't this also true for the permutation learned for linear paths, since it involves finding the best permutation $\pi$ with a greedy algorithm on a given set of examples?





**Summary Of The Paper:**

The authors aim to understand a model's sensitivity to spurious features through the lens of mode connectivity. First, they come up with a definition of the mechanistic similarity of different modes, on the basis of invariance to different interventions.  Then, they show a stronger form of connectivity between different modes, on the basis of mechanistic similarity. Finally, they conjecture that all modes with mechanistic similarity must be connected with a linear path up to permutations. They corroborate their conjecture with multiple experiments on CIFAR-10. Finally, they propose a method of connectivity-based fine-tuning to eliminate a model's sensitivity to spurious features.

**Summary Of The Review:**

Overall, my scores are slightly on the positive side. The paper aims to advance our understanding of mode connectivity and its connection to the decisions of a model. However, the proposed algorithm CBFT assumes that we must have access to both clean examples and examples with statistical cues, which begs the question of the relevance of CBFT compared to the direct training on clean examples. Moreover, experiments with stronger statistical cues will help strengthen the conjecture of the paper.

---

> ### Author Response · Authors · 2022-11-13
> **Response to Reviewer iiL3 (Part 1)**
>
> Thank you for your comments! As per your suggestions, we have made several updates to the paper, including addition of pointers to loss curves, addition of details on CBFT's execution, new results on CBFT, and a theoretical analysis that is rooted in complexity of discriminative attributes. We address your questions below.
>
> ------
> ### [Main Comments / Questions]
>
> > **(a) In Figures 4 and 5, shouldn't train/evaluation loss be plotted, as opposed to the train/test accuracy because definitions 4, 5, proposition 1, and conjecture 1 include loss function? Do we still see barriers in linear paths between mechanistically dissimilar models in figures 4 and 5, with loss function?**
>
> Please note we do provide loss curves in Appendix D, E and see qualitatively the same behaviors for both the measures--accuracy and loss. We have also added notes to all figures in the main text, referring readers to these sections.
>
> Our decision to use accuracy curves in the main paper was motivated by a visual presentation challenge: when studying connectivity on counterfactual datasets where the spurious cue or the underlying image is randomized, it was common for a few points on the connectivity path to have very large losses. Using a range containing these extreme values makes it difficult to discern (the smaller) performance differences among points on a connectivity path, and also make comparisons between evaluations on different counterfactual datasets challenging (since y-axes cannot use the same scale). In contrast, Train / Test accuracy curves are easier to visualize because they are constrained to a uniform range of 0-100%, with all large loss models yielding random (1 / # of classes) accuracy. Thus, while their qualitative patterns are the same, using accuracy plots better illustrated the salient concepts.
>
> As an aside, we also note that accuracy curves and loss curves are equally popular in mode connectivity papers: e.g., Frankle et al. [1], who had one of the first papers popularizing the idea of linear mode connectivity, only show accuracy curves in their work. Nonetheless, for thoroughness, we include both accuracy and loss curves in this paper.
>
>
> [1] Frankle et al. (https://arxiv.org/abs/1912.05671)
>
> ------
>
> > **(b) The CBFT algorithm (in 3) assumes that we have access to both a clean dataset and a dataset with statistical cues. Can the authors comment on whether one can identify examples from both groups in an unsupervised manner? If not, why should one use CBFT and not directly train the model (from scratch) on the clean examples in the dataset?**
>
> Please note that we assume access to only a **minimal** clean dataset, exactly the same as recent works that design methods to mitigate a model's sensitivity to spurious attributes [1, 2]. For example, in our evaluation, we reserve *only 2,500* samples out of the total 50,000 samples for the different datasets analyzed in our paper (for both CBFT and the baseline techniques). Given this sample efficiency constraint, *training from scratch is not expected to be a viable strategy* for developing a model that performs well on the given task and also boasts the desired invariance properties. To demonstrate this empirically, we now include additional results on training from scratch on the minimal clean datasets in Appendix C.2. *The results clearly demonstrate training from scratch severely lags behind performance of CBFT and other baselines.* For example, on our CIFAR-10 with box cues dataset, we see CBFT outperforms training from scratch by 21--28% on both data with and without spurious attributes; the results depend on the proportion of spurious data in the pretraining step.
>
> [1] Kirichenko et al. (https://arxiv.org/abs/2204.02937)
>
> [2] Kumar et al. (https://arxiv.org/abs/2202.10054)
>
> ------

---

> > ### Author Response · Authors · 2022-11-13
> > **Response to Reviewer iiL3 (Part 2)**
> >
> > > **(c) It's not clear how the authors run CBFT introduced in eq. (3) on real-world models.**
> > >
> > >  **(i). How do the authors optimize step (i)? Do they explicitly search over the neighborhood around $\theta\_C$ for one $\theta$ that maximizes the loss function on the line connecting $\theta$ and $\theta\_C$? Or do they conduct an optimization of the form $\min |\lambda\_1 - \mathcal{L}\_{\text{CE}}(f(\mathcal{D}\_{\text{C}}; \gamma\_{\theta \to \theta\_{\text{C}}}(t)), y)|$**
> > >
> > > **(ii). Furthermore, do the authors follow an alternative minimization for steps (i) and (ii)?**
> > >
> > > **(iii). Do the authors include permutation in step (i)?**
> >
> > We apologize that this was not clear in the original submission. We have now *significantly expanded the discussion* in Section 5 and Appendix A.2 to include details on how CBFT and other baselines are executed. We have also added ablation studies on CBFT in Appendix C.1 to further measure which part of the algorithm helps CBFT achieve high performance. We answer your other questions below.
> >
> >
> > **Answer to (i).** We conduct an optimization of the form $\min\_{\theta} |\lambda\_1 - \mathcal{L}\_{\text{CE}}(f(\mathcal{D}\_{\text{C}}; \gamma\_{\theta \to \theta\_{\text{C}}}(t)), y)|$ in step (i) of CBFT. Specifically, we first randomly sample points on the linear path $\gamma\_{\theta \to \theta\_{\text{C}}}(t))$ using a truncated Gaussian distribution that is constrained to the range $[0, 1]$ and has a mean and standard deviation of 0.5 and then maximize the loss at these points up to an upper bound $\lambda\_1$. This ensures the loss barrier has a higher likelihood of occuring at the center of the linear path.
> >
> >
> > **Answer to (ii).** We indeed conduct an alternating minimization for the two steps. This is necessary because one cannot differentiate through the operation that computes parameters on the linear path $\gamma\_{\theta \to \theta\_C}(t)$ and attaches them to a model for optimization. To circumvent this problem, we run autodiff to compute gradients for the model that has parameters $\gamma\_{\theta \to \theta\_C}(t)$ and explicitly compute gradients with respect to $\theta$ by using the following relationship: $\nabla\_{\theta} \mathcal{L}(\gamma\_{\theta \to \theta\_{C}}(t)) = \left(\nabla\_{\theta} \gamma\_{\theta \to \theta\_{C}}(t)\right)^{T} \, \nabla\_{\gamma\_{\theta \to \theta\_{C}}(t)} \mathcal{L}(\gamma\_{\theta \to \theta\_{C}}(t)) = (1-t) \nabla\_{\gamma\_{\theta \to \theta\_{C}}(t)} \mathcal{L}(\gamma\_{\theta \to \theta\_{C}}(t)).$ That is, we first compute the gradient of the objective with respect to $\gamma\_{\theta \to \theta\_{C}}(t)$ and multiply that by a factor of $1-t$ to retrieve the gradient of the objective with respect to $\theta$. Since this step must be carried out explicitly, we need to use an alternating minimization process for optimizing CBFT's barrier and invariance losses (i.e., steps (i) and (ii) in Eq. 3).
> >
> >
> > **Answer to (iii).**  We note it is unnecessary to account for permutations in step (i) of CBFT. Specifically, as shown by Simsek et al. [1], two minimizers in the landscape that are permutations of each other are separated by a large loss, permutation-induced saddle along the linear path between them. While CBFT's step (i) also increases loss near the center of the linear path between two minimizers, we do not allow it to increase beyond a reasonable upper bound (controlled by the hyperparameter $\lambda\_1$). This constraint helps prevent the model from moving to another basin that is a permutation of the current one.
> >
> > [1] Simsek et al. (https://arxiv.org/abs/2105.12221)
> >
> > -------

---

> > > ### Author Response · Authors · 2022-11-13
> > > **Response to Reviewer iiL3 (Part 3)**
> > >
> > > > **(d) All statistical cues introduced in the paper are very simple (linearly separable) cues.**
> > >
> > > Thank you for this comment! We will first clarifying details of our experimental setup and then discuss a new interesting theoretical analysis motivated by your question.
> > >
> > >
> > > **Clarification on Experimental Setup:** We first note that we use statistical cues that are *easily separable* because, as shown by prior work, naturally occuring spurious attributes in real data are generally easily separable [1, 2, 3, 4, 5, 6, 7, 8], prompting neural networks to rely on them due to their simplicity bias [1]. Thus, making the cues in our analysis easily separable helps use them as stand-ins for naturally occuring spurious attributes and makes our analysis practically relevant. Second, we highlight that the cues used in our experiments are not necessarily *linearly separable*. In fact, only our "located box cue" variant of CIFAR-10 (see Fig. 2) contains linearly separable cues; the CIFAR-100 with "located / colored box cue" and the Dominoes datasets *do not* contain linearly separable cues (see Fig. 2). For example, in Dominoes, we use images from Fashion-MNIST as cues, which are not linearly separable, but have a lower complexity than CIFAR-10 images that we use as the primary dataset for defining "natural" images. Similarly, the located / colored box in CIFAR-100 uses a non-linear cue because all classes that share the first digit share the same box location, while all classes that share the same second digit share the same box color. Thus, the model has to learn at least a depth-2 decision tree to solve the task by using the cue, implying the cue is not linearly separable.
> > >
> > >
> > >
> > > **Theoretical Analysis:** Prompted by your comment, we realized we can prove Conjecture 1 for a two-layer fully connected network with ReLU activations by developing a data-generating process that creates inputs with multiple predictive attributes of different complexities (see Appendix F.4; pages 35--39). This process is an extension of that proposed by Shah et al. [1] to analyze the simplicity bias of neural networks and defines the complexity of an attribute to be the number of piece-wise linear splines needed to achieve zero loss on a discriminative task by using that attribute. In this sense, a linearly separable attribute has the lowest complexity, since it requires only one piece-wise linear spline (i.e., a line). In fact, *the analysis includes several results that we believe can be of independent interest to the ICLR community,* e.g., in Lemma 2, we demonstrate why one needs to permute two models to observe linear mode connectivity, and in Remark 2 we propose a measure that can be used to predict whether there exists a permutation that allows for two models to be linear-mode connected!
> > >
> > >
> > > [1] Shah et al. (https://arxiv.org/abs/2006.07710)
> > >
> > > [2] Geirhos et al. (https://arxiv.org/abs/2004.07780)
> > >
> > > [3] Geirhos et al. (https://arxiv.org/abs/1811.12231)
> > >
> > > [4] Hermann et al. (https://arxiv.org/abs/1911.09071)
> > >
> > > [5] Beery et al. (https://arxiv.org/abs/1807.04975)
> > >
> > > [6] Jacobsen et al. (https://arxiv.org/abs/1811.00401)
> > >
> > > [7] Teney et al. (https://arxiv.org/abs/2207.02598)
> > >
> > > [8] D'Amour et al. (https://arxiv.org/abs/2011.03395)
> > >
> > > --------
> > >
> > > > **The authors point out a very important characteristic of "quadratic paths" in the appendix: the evaluation performance of the model on the path will depend on the examples used to find the path, and hence may not show generalization across datasets/distributions. However, isn't this also true for the permutation learned for linear paths, since it involves finding the best permutation $\pi$ with a greedy algorithm on a given set of examples?**
> > >
> > > Thank you for this query! We considered using datasets with and without cues in our preliminary experiments to find the optimal permutations that match two models in activations. However, we found that the paths identified for the two datasets showed exactly the same behavior. Since our experiments are relatively expensive to conduct, we decided to include only paths identified using the data without cues: accounting for multiple datasets, models, training pipelines, we performed 10,560 evaluations for just linear (permuted) paths. We have added a note in the appendix to make this clear.
> > >
> > > ------
> > > ------
> > >
> > > ***Overall Summary:*** Thank you again for carefully going through our results and providing insightful feedback that has significantly improved the paper! We hope that our replies have clarified your concerns and please let us know if you have any further questions. In light of these clarifications, we would appreciate if you can consider increasing your rating score.

---

> ### Author Response · Authors · 2022-12-04
> **Gentle Reminder**
>
> Dear Reviewer iiL3,
>
> We hope that we have justifiably answered your questions and properly accommodated the feedback in your review. Since the discussion period ends soon, please let us know if you have any further questions. Thank you!

---

### Official Review · Reviewer_bN6r · 2022-10-27

**Confidence:** 2
**Clarity, Quality, Novelty And Reproducibility:** 1. The author doesn't provide code.
2…
**Correctness:** 3
**Technical Novelty And Significance:** 3
**Empirical Novelty And Significance:** 3
**Recommendation:** 6

**Strength And Weaknesses:**

Pros:
1. The paper is well-written and easy to follow, although minor grammar error also exist (e.g. 2nd paragraph line 3 "to to").
2. The introduced problem is quite interesting and well-motivated.
3. The paper has conducted extensive experiment results to verify the effectiveness of the proposed hypothesis.

Cons:
1. For figure 2, I'm confused by the last two rows of right figure. What is the difference between "Rand. cue" and "Rand. Image w/ cue"?
2. For the augmented data, are they used along with original data for training? Or Models are only trained on the augmented data?

**Summary Of The Paper:**

The paper studies a new concept mechanistic mode-connectivity, which focusing on the mode-connectivity of models trained on different data distribution. Meanwhile, the paper also proposes connectivity based fine-tuning (CBFT), aiming at overriding existing mechanisms of a pretrained model by fine tuning it on a small out-of-distribution dataset.

**Summary Of The Review:**

The overall quality of this paper is good, where the analysis is comprehensive. Hence I recommend to accept this paper.

---

> ### Author Response · Authors · 2022-11-13
> **Response to Reviewer bN6r**
>
> Thank you for your feedback! We have updated the paper to address your questions and uploaded an extensive codebase with the supplementary material. We summarize the updates and answer remaining questions below.
>
> -----
>
> ### [Main Comments / Questions]
>
> > **For figure 2, I'm confused by the last two rows of right figure. What is the difference between "Rand. cue" and "Rand. Image w/ cue"?**
>
> We apologize for the lack of clarity in the original submission. We have expanded the discussion in Figure 2 and Section 2.2 of our revision to clarify the definitions of these datasets. We have also changed the title "Rand. Image w/ Cue" to "Rand. Image" for brevity and clarity. We further provide a summary description of these datasets below.
>
> Given an image with spurious cues, "Rand. Cue" and "Rand. Image" denote counterfactual variants of the image in which, respectively, synthetic and natural discriminative attributes of a sample are randomized to break their correlation with the target label.
>
> *(i) Rand. Cue:* This denotes a counterfactual in which the discriminative property of the spurious cue is randomized, but the underlying image remains the same. These counterfactuals allow us to assess how dependent a model is on the spurious cue during prediction. For example, consider our variant of the CIFAR-10 dataset in which we embed a *box cue* in an image, such that the *location* of the box depends on the target label--that is, the location of the box is the discriminative property of the spurious cue. The "Rand. Cue" counterfactual then involves randomizing the location of this box by moving it to a different location in the image that is predefined to correspond to another target label. If a model's loss does not increase when we randomize the location in this manner, then we can conclude that the model is invariant to the box cue--a desirable property.
>
> *(ii) Rand. Image:* This denotes a counterfactual in which the natural discriminative attributes are randomized, but the spurious cue remains unchanged. These counterfactuals allow us to assess how much a model depends on the natural discriminative information present in an image to make its predictions. For example, consider again the CIFAR-10 dataset with *box cues* discussed above. In this dataset, images of airplanes have a box located at the top left corner. The "Rand. Image" counterfactual for these images then involves keeping the box intact at the top left corner, but replacing the underlying airplane image with another image that belongs to a different class (e.g., a dog image). If a model's loss does not increase when we replace the image in this manner, then we can conclude that the model is invariant to the image itself--an undesirable property.
>
> ------
>
> > **For the augmented data, are they used along with original data for training? Or Models are only trained on the augmented data?**
>
> Depending on the experiment, we train models on varying proportions of data augmented with or without spurious cues. Specifically, the proportion of samples augmented with spurious cues in the training data changes according to the experiment. For example, in Figures 4, 5, and Table 1, we train models with six different proportions of data with spurious cues: 0%, 60%, 70%, 80%, 90%, or 100%. This construction allows us to create mechanistically dissimilar models by enforcing unshared invariances, as mentioned in Section 2.2 and shown in Figs. 10, 11 in the Appendix.
>
> ------
>
> ###  [Shorter Comments / Questions]
>
> > **The author doesn't provide code.**
>
> Please see the updated supplementary material. The code for creating our used synthetic datasets, their counterfactuals, training models, finding connectivity paths, and CBFT has now been uploaded.
>
> --------
> --------
>
> ***Overall Summary:*** Thank you again for providing useful feedback that has helped us improve the paper's clarity! We hope that our replies have clarified your concerns and please let us know if you have any further questions. In light of these clarifications, we would appreciate if you can consider increasing your rating and confidence score.

---

> ### Author Response · Authors · 2022-12-04
> **Gentle Reminder**
>
> Dear Reviewer bN6r,
>
> We hope that we have justifiably answered your questions and properly accommodated the feedback in your review. Since the discussion period ends soon, please let us know if you have any further questions. Thank you!

---

### Official Review · Reviewer_fq5i · 2022-10-29

**Confidence:** 3
**Correctness:** 3
**Technical Novelty And Significance:** 3
**Empirical Novelty And Significance:** 3
**Recommendation:** 6

**Clarity, Quality, Novelty And Reproducibility:**

Overall, the paper is well-written. The figures could use some clarification:

Figures with grids of plots were a bit dense to parse easily (e.g., Figure 4 and Figure 5).
- Figure 3 diagrams are hard to understand, especially the grid of similarity descriptions. Unclear what the rows and columns of the grid of rectangles are illustrating.
- Figure 4 is difficult to parse because column labels like "Linear Permuted" are not defined in the main text.

Reproducibility is unclear, but the authors mention they will release it during the rebuttal phase.

**Strength And Weaknesses:**

Strengths:
- The hypothesis this paper proposes about the optimization landscape and, specifically, the functional relationship with the energy barrier is interesting and an important area of better understanding the utility of mode connectivity.
- Thorough appendix with corresponding loss curves. (This reviewer's opinion: loss curves are easier to interpret for mode connectivity than accuracy curves)

Weaknesses:
- Unclear on which of the two steps in CBFT contributes to the improvements. An ablation analysis of each step would help disentangle the contribution.
- The definition of mode connectivity shown in Figure 6 seems to make mode connectivity and mechanistic similarity expected and potentially uninteresting. The monotonicity of the evaluation curve going from \theta_c to \theta_{nc} should be expected since the transition is going from a model trained on that dataset to one trained on a different dataset and vice versa. Could the authors clarify this issue? Perhaps this reviewer has made an error in understanding Figure 6. A more interesting result would be a "u" shaped accuracy barrier. But on most plots in Figure 4, this does not appear, and most curves are the expected monotonic trend.

**Summary Of The Paper:**

This work associates the notion of linear mode connectivity of two models with the functional similarity (mechanistic similarity) of the two models. They argue that the linear mode connectivity of the two models indicates they both have inherited potentially spurious/undesirable representations. Furthermore, they argue that naive fine-tuning cannot remove these spurious representations. Based on this connection, the authors propose a new algorithm referred to as connectivity-based fine tuning (CBFT), which encourages two properties for optimizing the parameters:

- Removal of the linear path between to spurious model
- Encourage functional similarity of the current model across spurious and non-spurious datasets.

**Summary Of The Review:**

There needs to be more work in understanding mode connectivity's functional utility, which this paper meaningfully explores. As it stands, mode connectivity (linear or non-linear) is an empirical phenomenon that is not well understood. This work explores possible avenues for improving model generalization based on this phenomenon. The main caveat to this work is the clarity is made somewhat opaque by the presentation of grids of experiments. The idea would be communicated much better if those results were condensed down.

---

> ### Author Response · Authors · 2022-11-13
> **Response to Reviewer fq5i (part 1)**
>
> Thank you so much for taking the time to write a very insightful review! We have thoroughly reflected your valuable feedbacks in the updated manuscript by including suggested ablation experiments on CBFT, releasing an extensive codebase, expanding figure captions, and implementing recommended edits throughout. We summarize the updates and answer remaining questions below.
>
> -----
>
> ### [Main Comments / Questions]
>
> > **Unclear on which of the two steps in CBFT contributes to the improvements. An ablation analysis of each step would help disentangle the contribution.**
>
> Thank you for this suggestion! We have added a detailed ablation experiments in Appendix C to study the impact of the two steps involved in CBFT, i.e., inducing a loss barrier between minimizers and enforcing invariance to spurious cues. The results corroborate our claims very well. Specifically, models trained *without steps to induce a loss barrier* continue to rely on the spurious cues, yielding mechanistically similar models and hence *undesirable* results: we see low accuracy when the spurious cue is randomized and high accuracy when the underlying natural image is randomized. Meanwhile, models trained *without the invariance loss* yield mechanistically dissimilar models which often have desirable properties: high accuracy when the spurious cue is randomized and low accuracy when the underlying natural image is randomized. However, these models can in certain cases noticeably underperform CBFT-trained models when the spurious cue is present (e.g., see results on Dominoes). This behavior is expected since one can induce loss barriers by learning to be anti-correlated with the spurious cue, i.e., learning to intentionally produce incorrect predictions when the cue is present. Adding the invariance loss rectifies this pitfall because the model learns to produce the same features regardless of the presence/absence of the cue, yielding high accuracy in both scenarios. *Consequenly, we see CBFT consistently outperforms the two ablated variants.*
>
>
> *Summary*: Inducing a barrier loss helps prevent linear connectivity and induce mechanistically dissimilar models, while adding an invariance penalty helps select the exact mechanisms we want the models to differ in. Overall, both steps in CBFT are required for achieving desired results, i.e., invariance to spurious cues and reduced performance under randomization of natural attributes.
>
> ---------
>
> > **The definition of mode connectivity shown in Figure 6 seems to make mode connectivity and mechanistic similarity expected and potentially uninteresting. The monotonicity of the evaluation curve going from \theta_c to \theta_{nc} should be expected since the transition is going from a model trained on that dataset to one trained on a different dataset and vice versa. Could the authors clarify this issue? Perhaps this reviewer has made an error in understanding Figure 6. A more interesting result would be a "u" shaped accuracy barrier. But on most plots in Figure 4, this does not appear, and most curves are the expected monotonic trend.**
>
> Thank you for this very careful feedback and we apologize for the confusion! We mistakenly plotted the slope in this conceptual figure (i.e., Figure 6) incorrectly, leading to a drawing that did not convey the message we desired.
>
> We have now updated Figure 6 to clarify our intended meaning. While the main geometric feature of the "u" shaped accuracy barrier stays intact, you are right that the overall slope of evaluation curve going from $\theta_c$ to $\theta_{nc}$ should have been flat. This qualitative pattern is in fact what we empirically observe in our results in Figure 4 (specifically, the "Linear, w/ Cue" and "Linear w/o Cue" panels).
>
> ---------
>
> ### [Shorter Comments / Questions]
>
> > **Figure 3 diagrams are hard to understand, especially the grid of similarity descriptions. Unclear what the rows and columns of the grid of rectangles are illustrating.**
>
> Thank you for this feedback! We have updated both Figure 3 and its caption to clarify how each entry of the grid represents an output given a pair of a counterfactual image (row) and a model (column).
>
> ---------
>
> > **Figure 4 is difficult to parse because column labels like "Linear Permuted" are not defined in the main text.**
>
> Please see Sec. 2.1: We now have expanded the discussion of linear mode connectivity under permutations in that section and also introduce the phrase "Linear (permuted)" there itself.
>
> We have also added the following line to the main text when discussing our experimental setup: "We analyze accuracy on ... linear (permuted) paths, i.e., linear paths after permuting model neurons to match in activations."
>
> ---------

---

> > ### Author Response · Authors · 2022-11-13
> > **Response to Reviewer fq5i (part 2)**
> >
> > > **Thorough appendix with corresponding loss curves. (This reviewer's opinion: loss curves are easier to interpret for mode connectivity than accuracy curves.)**
> >
> > As noted by the reviewer, we do provide loss curves in Appendix D, E and see qualitatively the same behaviors for both the measures--accuracy and loss. We have also added notes to all figures in the main text, referring readers to these sections.
> >
> > Our decision to use accuracy curves in the main paper was motivated by a visual presentation challenge: when studying connectivity on counterfactual datasets where the spurious cue or the underlying image is randomized, it was common for a few points on the connectivity path to have very large losses. Using a range containing these extreme values makes it difficult to discern (the smaller) performance differences among points on a connectivity path, and also make comparisons between evaluations on different counterfactual datasets challenging (since y-axes cannot use the same scale). In contrast, Train / Test accuracy curves are easier to visualize because they are constrained to a uniform range of 0-100%, with all large loss models yielding random (1 / # of classes) accuracy. Thus, while their qualitative patterns are the same, using accuracy plots better illustrated the salient concepts.
> >
> > As an aside, we also note that accuracy curves and loss curves are equally popular in mode connectivity papers: e.g., Frankle et al. [1], who had one of the first papers popularizing the idea of linear mode connectivity, only show accuracy curves in their work. Nonetheless, for thoroughness, we include both accuracy and loss curves in this paper.
> >
> >
> > [1] Frankle et al. (https://arxiv.org/abs/1912.05671)
> >
> > ---------
> > ---------
> >
> > ***Overall Summary:*** Thank you again for carefully going through our results and providing insightful feedback that have significantly improved the paper! We hope that our replies have clarified your concerns and please let us know if you have any further questions. In light of these clarifications, we would appreciate if you can consider increasing your rating and confidence score.

---

> ### Author Response · Authors · 2022-12-04
> **Gentle Reminder**
>
> Dear Reviewer fq5i,
>
> We hope that we have justifiably answered your questions and properly accommodated the feedback in your initial review. Since the discussion period ends soon, please let us know if you have any further questions. Thank you!

---

### Official Review · Reviewer_X6v6 · 2022-11-24

**Confidence:** 3
**Correctness:** 2
**Technical Novelty And Significance:** 3
**Empirical Novelty And Significance:** 3
**Recommendation:** 5

**Clarity, Quality, Novelty And Reproducibility:**

Overall, this is a well-written paper. In contrast to the contribution of each section, the connection between the sections is insufficient.

* Clarity: The Sections, except for Section 5, are clear, but their connections are lacking.
* Quality: Mechanistic similarity (Section 3) and Mechanistic mode connectivity (Section 4) are great, but Connectivity-Based Fine-Tuning lacks motivation. (See the second issue of Weakness.)
* Novelty: The novelty and contribution points of this paper are clear.
* Reproducibility: Although I have not run it, I believe the submitted code will ensure reproducibility.

**Strength And Weaknesses:**

* Strength

  * From a theoretical point of view, their contributions are largely twofold. First, authors **applied the concept of invariance** to explain the underlying mechanism of the models. (Section 3) Second, a **fascinating conjecture** (Conjecture 1) was presented by the authors about how mode connectivity, an abstract concept in loss-landscape, will affect real-world applications. (Section 4)
  * On the application side, the authors proposed Connectivity-Based Fine-Tuning as a way to **intentionally editing** the pre-trained model's underlying mechanism. (Section 5)
  * These contribution points are novel and interesting for scientists studying the loss-landscape of NNs.

* Weakness

  However, readers who wish to apply these findings to their problems may encounter some difficulties for the following reasons. Considering this is discussion phase 2, it is impossible to revise the paper, but I believe the first and third issues should be addressed.

  1. It is unclear how to get rid of spurious attributes in Sections 3 and 4. For example, the authors only explain how the proposed notion of mechanistically similar confirms the functional equivalence of the two models in Section 3. **There is no discussion of what mechanstically similarity corresponds to spurious attributes**. This issue also applies to Section 4 (i.e., There is no explicit connection between spurious attributes and Connectivity-Baed Fine-Tuning).  As a result, readers may feel as if there is no natural connection between sections. **Since the content of each section is interesting enough, I believe the connectivity between sections is the greater concern.**

  2. Although the code was submitted during the discussion period, Section 5's Connectivity-Based Fine-Tuning remains vague in their main paper. For example, how do you pick a minimal dataset $\mathcal{D}_{NC}$ that does not contain attribute C? Also, why does the average representation matching term used in (ii) of equation (2) make an invariant prediction? For example, what happens if the term is the average of multiple augmentations of a single image rather than the average of multiple images of a single class? **Since Connectivity-Based Fine-Tuning is not the only (or optimal) algorithm that achieves the authors' purpose, these questions are quite natural, and the authors should have explained them appropriately.**

  3. There are some mathematical expressions that seem to be abused.

     * Although an inverse mapping is required for isomorphism, $\mathcal{A}^{\alpha_i}_i$ has no inverse mapping because it fixes the value of the $i$-th component of the latent variable to $\alpha_i$.

     * While authors denote the domain of $\mathcal{E}$ as $\mathcal{X} \times \mathbb{R}^{m}$, $\mathbb{R}^{m}$ is incorrect because $\hat{\mathcal{A}}$ is a concatenation of interventions.

     * Authors used square brackets both to represent lists (e.g., $[K]$ in Section 2) and to represent closed intervals ($[0,1]$ in Section 5).

     * Furthermore, some mathematical expressions are used without any definition. For example, ㅅthey did not define exactly what $[K]$ means. This can only be inferred from context. This issue also applies to the Truncate function in (ii) of equation (3). The exact mathematical expression of $f_\gamma$ in Section 5 is not specified. Similarly, $\gamma_{\theta \rightarrow \theta_C}(t)$ is not a path, but rather a point on a path.

       **Combined with the first weakness, these misuses of mathematical expressions will present a huge barrier for readers unfamiliar with the field.**

**Summary Of The Paper:**

This paper is primarily concerned with **identifying spurious attributes** learned by NNs when learning a specific dataset and **correcting them** during fine-tuning. To this end, they propose mechanistic similarity, a concept that checks whether two pre-trained models with low loss are functionally similar. Based on this concept, they develop connectivity-based fine-tuning, a fine-tuning technique that can modify the mechanism of a pre-trained model.

**Summary Of The Review:**

This paper proposes a novel solution (Connectivity-Based Fine-Tuning) to an interesting problem (removing spurious attributes). While the concepts of Mechanistic Similarity and Mechanistic Mode Connectivity are exciting and inspiring, no discussion has been given on how to remove the spurious attribute. The paper will be helpful to readers in many fields with revisions for natural connections between sections.

---

> ### Author Response · Authors · 2022-11-25
> **Response to Reviewer X6v6 (Part 1)**
>
> We thank the reviewer for their highly positive feedback on the contributions of our paper! We are glad they found our analysis "novel", "fascinating", "exciting and inspiring" and "interesting". In the following, we respond to specific comments / questions by the reviewer.
>
> ### [On Paper Summary]
>
> We believe there is some misunderstanding in the reviewer's drawn conclusions about our paper's motivations and objectives. We clarify this below.
>
> > **Paper summary: This paper is primarily concerned with *identifying spurious attributes* learned by NNs when learning a specific dataset and *correcting them* during fine-tuning.**
>
> **We emphasize our paper's practical goal is mitigation, not identification, of spurious attributes**. While there are already several papers that focus on identifying spurious attributes (e.g., see [1]), *our focus is removing a model's reliance on such attributes*. We highlight this setting is *exactly the same* as several prior works on neural network robustness [2, 3]: such works also assume the targeted spurious attribute is already known and that a minimal "clean" dataset is available where the spurious attribute is not predictive of the target label.
>
>
> [1] Singla et al. (http://salient-imagenet.cs.umd.edu)
>
> [2] Kirichenko et al. (https://arxiv.org/abs/2204.02937)
>
> [3] Kumar et al. (https://arxiv.org/abs/2202.10054)
>
>
> > **Paper summary (Continued): ... To this end, they propose mechanistic similarity, a concept that checks whether two pre-trained models with low loss are functionally similar. Based on this concept, they develop connectivity-based fine-tuning, a fine-tuning technique that can modify the mechanism of a pre-trained model.**
>
> **Please note we never claim mechanistic similarity is defined for identification of spurious attributes.** *Our motivation for defining mechanistic similarity was to formally assess whether two models that make predictions using entirely different input attributes (spurious or not) are connected in the loss landscape via relatively simple paths*. We exploit this analysis to design a technique (called CBFT) for altering a model's mechanisms in a sample-efficient manner, and use it for the practically relevant objective of removing a model's reliance on spurious attributes.
>
> ---------
>
> ### [Major Comments / Questions]
>
> > **Readers may feel as if there is no natural connection between sections. *Since the content of each section is interesting enough, I believe the connectivity between sections is the greater concern.***
>
> **We emphasize all sections are indeed tightly connected.** As we thoroughly discuss in the introduction (Section 1), our goal is altering a model's mechanisms by analyzing how mechanistically dissimilar models are connected in the loss landscape. To this end, we define a notion of Mechanistic Similarity (Section 3) and formally / empirically analyze connectivity properties of mechanistically (dis)similar models (Section 4), demonstrating an intricate relationship between lack of linear connectivity and mechanistic dissimilarity. Based on this analysis, we introduce CBFT, a method designed for altering a model's mechanisms and remove its reliance on spurious attributes (see Section 5). *Given the tight relationship and motivation connecting different sections, we disagree with the reviewer that the sections are not well connected.*
>
> We also highlight specific points in the paper where we already clarify the relationship between different sections: in second paragraph of Section 1, we thoroughly discuss our practical goal is altering a model's mechanisms and removing its reliance on spurious attributes. In paragraph 3 of Section 1, we discuss how we will achieve this goal by analyzing mode connectivity of mechanistically dissimilar models and designing an algorithm (CBFT) that helps remove a model's reliance on spurious attributes; Figure 1 was also dedicated to this purpose. Moreover, the second paragraph of Section 5 explicitly states that CBFT is designed on the basis of our connectivity analysis of mechanistically dissimilar models; Figure 6 also intuitively clarifies this point. *Overall, we stress the organization and relationship between different sections has been highlighted throughout the paper.*
>
> ---------

---

> > ### Author Response · Authors · 2022-11-25
> > **Response to Reviewer X6v6 (Part 2)**
> >
> > > **It is unclear how to get rid of spurious attributes in Sections 3 and 4. For example, the authors only explain how the proposed notion of mechanistically similar confirms the functional equivalence of the two models in Section 3. *There is no discussion of what mechanstically similarity corresponds to spurious attributes.* This issue also applies to Section 4 (i.e., There is no explicit connection between spurious attributes and Connectivity-Baed Fine-Tuning).**
> >
> > We emphasize again that the *the goal of Sections 3, 4 was not to discuss identification/removal of spurious attributes, but rather to establish the scientific grounds that lead to our proposed method for removing a model's reliance on such attributes, i.e., CBFT (in Section 5).* **That is, formalizing the definition of mechanistic similarity (Section 3) and analyzing mode connectivity of mechanistically (dis)similar models (Section 4) are essential steps for designing CBFT in Section 5.** We also stress that *we indeed discuss the relationship between our proposed definitions for mechanistic similarity / connectivity and distribution shifts* (see paragraphs after defintions of mechanistic connectivity and mechanistic similarity on Page 5).
> >
> >
> > ---------
> >
> > > **Although the code was submitted during the discussion period, Section 5's Connectivity-Based Fine-Tuning remains vague in their main paper. For example, how do you pick a minimal dataset $D_{NC}$ that does not contain attribute $C$? Also, why does the average representation matching term used in (ii) of equation (2) make an invariant prediction? For example, what happens if the term is the average of multiple augmentations of a single image rather than the average of multiple images of a single class? *Since Connectivity-Based Fine-Tuning is not the only (or optimal) algorithm that achieves the authors' purpose, these questions are quite natural, and the authors should have explained them appropriately.***
> >
> > We emphasize that we already provide a highly detailed codebase, thorough implementation details (spread over Section 5 and Appendix A.2), extensive empirical results, and several ablations to analyze CBFT. Thus, we do believe the technique has been thoroughly explained. However, we respond to your more specific comments below.
> >
> > - **On picking $\mathcal{D}_{NC}$.** Please note we already state in the paper (see Paragraph 1 in Section 5 and Appendix A.2) that our experimental setup follows that of prior work on mitigating a model's reliance on spurious attributes [1, 2]: that is, we assume the minimal clean dataset is randomly sampled from a data generating process that differs from the pretraining one in regards to the absence of the spurious attribute. Importantly, therefore, these samples are not mere augmentations of the existing samples. For example, similar to the setup of Kirichenko et al. [1], we also use the Dominoes dataset in our evaluation and reserve 2500 clean samples of this dataset for fine-tuning (remaining samples may or may not contain the spurious attribute, depending on the proportions listed in Table 1). While we believe the setup was clear in section on CBFT (Section 5) and elaborated on in the Appendix (Section A.2), we do agree with the reviewer that expanding the discussion can help improve legibility for readers unfamiliar with literature on removing a model's reliance on spurious attributes. We promise to make this update in the final version of the paper.
> >
> > - **On invariance loss.** Please note an average representation / centroid based invariance loss is commonly used in domain adaptation / domain generalization literature (e.g., see [3], which uses invariance loss on class-average representations like us, and [4], which uses a penalty on the covariance matrix, whose diagonal is the class-average representation). Essentially, this loss can be considered an approximate invariance penalty. While a per-sample invariance loss would be ideal and can lead to perfect invariance, such a loss would require access to exactly the same samples in the presence and absence of spurious cues (these samples can be considered augmentations of each other). This can be an unrealistic demand from a practical standpoint. We are happy to expand the discussion and clarify this point further in the final revision of the paper with references to [3] and [4]. Currently, we note that the success of CBFT in inducing invariant representations (see Table 1) already demonstrates the approximate and more realistic penalty of average representation invariance works well.
> >
> >
> > [1] Kirichenko et al. (https://arxiv.org/abs/2204.02937)
> >
> > [2] Kumar et al. (https://arxiv.org/abs/2202.10054)
> >
> > [3] Li et al. (https://arxiv.org/abs/1807.08479)
> >
> > [4] Sun and Saenko (https://arxiv.org/abs/1607.01719)
> >
> >
> > -------

---

> > > ### Author Response · Authors · 2022-11-25
> > > **Response to Reviewer X6v6 (Part 3)**
> > >
> > > ### [Minor Comments / Questions]
> > >
> > > > **There are some mathematical expressions that seem to be abused.**
> > >
> > > Thank you for this comment on minor notational confusions. We promise these will be fixed and reflected in the final version (as the reviewer pointed out, paper revisions are not allowed at this stage in the process, else we would update the paper with these changes right away). We discuss the updates below.
> > >
> > > > - **Although an inverse mapping is required for isomorphism, $\mathcal{A}\_{i}^{\alpha\_{i}}$ has no inverse mapping because it fixes the value of the $i^{th}$ component to $\alpha\_{i}$.**
> > >
> > > Agreed. We were following prior literature while writing this definition [1], but, unlike that work, we use hard interventions and hence do not need this process to be isomorphic; only the data-generating process needs to be invertible. We will remove the constraint of isomorphism therefore.
> > >
> > > [1] Besserve et al. (https://arxiv.org/abs/1812.03253)
> > >
> > > > - **While authors denote the domain of $\mathcal{E}$ as $\mathcal{X} \times R^{m}$, $R^{m}$ is incorrect because $\widehat{\mathcal{A}}$ is a concatenation of interventions.**
> > >
> > > Agreed. We will change the domain of $R^{m}$ to a composition of $m$ separate reals.
> > >
> > > > - **Authors used square brackets both to represent lists (e.g., $[K]$ in Section 2) and to represent closed intervals ($[0, 1]$ in Section 5).**
> > >
> > > We will explicitly state in Section 2 (preliminaries) that $[K]$ denotes the set of integers from $1$ to $K$. We will also add that $[a, b]$ denotes a closed interval between reals $a, b$.
> > >
> > > > - **Furthermore, some mathematical expressions are used without any definition. For example, they did not define exactly what $[K]$ means. This can only be inferred from context. This issue also applies to the Truncate function in (ii) of equation (3).**
> > >
> > > In the pursuit of saving space, we had to fallback on standardly used mathematical notations in the community (e.g., $[K]$ denotes set of integers from $1$ to $K$). However, as promised above, we will update the preliminaries section to explicitly state the meaning of $[K]$. We will also update the paragraph before Equation 3 to explicitly define a truncated normal distribution.
> > >
> > > > - **The exact mathematical expression of $f_{\gamma}$ in Section 5 is not specified.**
> > >
> > > Please note we never use the notation $f_{\gamma}$ in Section 5 (or in the paper in general). Perhaps the reviewer is pointing to $f_{r}$, which we define as the representation of the model on a given input? While we believe this is relatively clear, we are also happy to add in the preliminaries that a representation denotes the model's output before the classification layer.
> > >
> > > > - **Similarly, $\gamma_{\theta \to \theta_{C}}(t)$ is not a path, but rather a point on a path.**
> > >
> > > We will update the text and use $\gamma_{\theta \to \theta_{C}}(.)$ to denote a path, while reserving $\gamma_{\theta \to \theta_{C}}(t)$ to denote points on a path.
> > >
> > >
> > > > - **Combined with the first weakness, these misuses of mathematical expressions will present a huge barrier for readers unfamiliar with the field.**
> > >
> > > We agree with the reviewer that rectifying these notational issues will help the paper's legibility and **stress that these issues are very easily fixable (again, we would upload a revision right away if it were possible)**. However, as can be seen in our answers above, the issues were relatively minor and the concepts were indeed clear from context (as also noted by the reviewer). Thus, we humbly request the reviewer to discount these notational mistakes for deciding the score for the paper.
> > >
> > >
> > > ------

---

> > > > ### Comment · Reviewer_X6v6 · 2022-11-25
> > > > **Response**
> > > >
> > > > Thank you for the detailed rebuttal.
> > > >
> > > > I still feel this paper is **unfriendly to readers unfamiliar with the subjects** of the paper (e.g., mode connectivity, invariance), which makes me hesitant to raise the score. The authors, for example, argued that *concepts were indeed clear from the context*. This, however, means that **legibility can be sacrificed depending on a reader's level of knowledge**. This issue also applies to Section 5. While the authors point out that *their experimental setup is similar to previous studies on mitigating the reliance on spurious attributes in models [1, 2]*, the authors present detailed assumptions and references of their methods **after explaining the method** (in Evaluating CBFT paragraph). In this case, **readers are confronted with a description of the method (CBFT) without understanding what problem the authors are trying to solve**. Therefore, I recommend presenting the details of assumptions about the method before explaining the method. In addition, I think it would be helpful for readers' understanding if a notational table or pseudo code for CBFT is provided in the Appendix (with appropriate references).

---

> > > > > ### Author Response · Authors · 2022-11-25
> > > > > **Continuing discussion**
> > > > >
> > > > > Thank you for the prompt reply!
> > > > >
> > > > >
> > > > > > **I still feel this paper is unfriendly to readers unfamiliar with the subjects of the paper (e.g., mode connectivity, invariance), which makes me hesitant to raise the score. The authors, for example, argued that concepts were indeed clear from the context.**
> > > > >
> > > > > Please note that **we already provide thorough definitions for each of the concepts listed by the reviewer (Definition 1: mode connectivity and Definition 3: invariance)**. We put a significant effort in the paper to clarify these concepts both formally and intuitively: e.g., see Figure 1, which motivates our lens of mode connectivity, and Figure 3, which intuitively explains invariances and mechanistic similarity. We also provide significant background (see Section 2.1, Preliminaries and Appendix B) and include almost 100 references throughout the paper. However, we are happy to expand this discussion even further in the next revision and include a unified background section. We also note that our remark, "these concepts were clear from context", was **specifically in response to the reviewer's raised notational concerns** (see Response to Reviewer, Part 3). Since the reviewer themselves stated that they could infer the notations from context, the goal of our remark was to highlight that these minimal notational concerns did not hinder the reviewer's scientific understanding of the paper. *We also emphasize again that these concerns were few and easily fixable via a revision; we would happily revise the paper right away if it were possible.*
> > > > >
> > > > >
> > > > >
> > > > > ----
> > > > >
> > > > > > **This, however, means that legibility can be sacrificed depending on a reader's level of knowledge. This issue also applies to Section 5. While the authors point out that their experimental setup is similar to previous studies on mitigating the reliance on spurious attributes in models [1, 2], the authors present detailed assumptions and references of their methods after explaining the method (in Evaluating CBFT paragraph).**
> > > > >
> > > > > In the "Evaluating CBFT" section mentioned by the reviewer, we state that we are going to evaluate CBFT against two recent baselines proposed by Kirichenko et al. and Kumar et al.; quoting from our paper: "We empirically validate the effectiveness of CBFT compared to recent baselines designed for reducing a model’s reliance on spurious attributes (Kirichenko et al., 2022b; Kumar et al., 2022) (see App. A.2 for training details)." **However, these methods were referenced and it was mentioned that we'll be using their experimental setup several times before as well.**
> > > > >
> > > > > Indeed, we clarify our problem setup and motivation at several places: e.g., see Paragraph 2 in Section 1, end of paragraph 3 in Section 1, and Paragraph 1 in Section 5. Further, the references to the mentioned papers are already placed in these spots, i.e., much before the "Evaluating CBFT" section. We also provide a detailed caption in Table 1 to clarify our setup.
> > > > >
> > > > > We acknowledge that due to space constraints, we did defer implementation details to the appendix and only added a forward reference in the main paper. However, we are happy to pull some of this text back to the main paper and further expand the discussion there.
> > > > >
> > > > >
> > > > > > **In addition, I think it would be helpful for readers' understanding if a notational table or pseudo code for CBFT is provided in the Appendix (with appropriate references).**
> > > > >
> > > > > Thank you for the suggestion! We will happily add a pseudocode to the paper's appendix and appropriate references to it in the main text.

---

> > > > > > ### Author Response · Authors · 2022-12-04
> > > > > > **Gentle Reminder**
> > > > > >
> > > > > > Dear Reviewer X6v6,
> > > > > >
> > > > > > We hope that we have justifiably answered your questions and properly accommodated the feedback in your initial review. Since the discussion period ends soon, please let us know if you have any further questions. Thank you!

---

### Author Response · Authors · 2022-11-13
**Summary of Revision**

We thank the four reviewers for their insightful feedback and for unanimously recommending acceptance of our submission. We are pleased to hear that all the reviewers find our newly introduced mechanistic view of mode connectivity, "interesting and an important area" [R fq5i], "quite interesting and well-motivated" [R bN6r], "very interesting and will be useful for the scientific community" [R iiL3], and our proposed fine-tuning algorithm "demonstrates good empirical performance" [R Y4FF].

To fully reflect the reviewers' valuable feedback, we have conducted  new theoretical and empirical analyses in the revised manuscript and attached an extensive code base making sure all the results are reproducible. In summary, major additions include:
* *Release of an extensive codebase* that allows training models on datasets with different proportions of synthetic cues, counterfactual evaluations, finding and evaluating mode connectivity paths, and several fine-tuning methods.
* *Addition of new experimental results*: ablations on CBFT (Appendix C.1, Table 3) and comparisons with training from scratch (Appendix C.2, Table 4)
* *Addition of new theoretical results* (prompted by reviewer iiL3's query on complexity of synthetic cues): a proof for conjecture 1 for a two-layer fully connected network, assuming a data-generating process that captures the essence of our empirical setup in the main paper and involves multiple predictive attributes of different "complexity" (Appendix F.4, Pages 35--39). *We believe this analysis can be of independent interest to the ICLR community:* e.g., in Lemma 2, we demonstrate why one needs to permute two models to observe linear mode connectivity, and in Remark 2 we propose a measure that can be used to predict whether there exists a permutation that allows for two models to be linear-mode connected!


We respond to individual comments and questions below.

---

### Decision · Program_Chairs · 2023-01-20

**Decision:**

Reject

**Justification For Why Not Higher Score:**

interesting hypothesis / findings but no meaningful implications is provided.

**Justification For Why Not Lower Score:**

n/a

**Metareview: Summary, Strengths And Weaknesses:**

The paper hypothesizes and experimentally verifies that mechanistically dissimilar minimizers cannot be linearly mode-connected. Based on this hypothesis, they propose a connectivity-based fine-tuning method that can modify the mechanism of a pre-trained model and correct the effect of spurious attributes during fine-tuning.

All reviewers, including this AC, agreed that the hypothesis itself is scientifically very interesting. However, they have failed to connect to a practically meaningful implication. I think the method proposed in section 5 is somewhat ad-hoc, and lacks experimental verification. Followings are the details:

- First of all, the setting of having clean data is not practical (at least it's less important/serious), and recently, most works are considering situations without bias information. Kumar et al mentioned by the author focuses on OOD, and Kirichenko et al seems to be still an arxiv paper.

- There are indeed some (old) studies such as [1-4] assuming that clean data is available or we know the bias label (of course, not a fine-tuning setting). It would be better if the method is compared against them and if the 'inducing a loss barrier' step can be verified even in these methods as well.

[1] Learning not to learn: Training deep neural networks with biased data, cvpr 2019
[2] Towards fairness in visual recognition: Effective strategies for bias mitigation, cvpr 2020
[3] End: Entangling and disentan- gling deep representations for bias correction, cvpr 2021
[4] Unbiased Classification Through Bias-Contrastive and Bias-Balanced Learning, NeurIPS 2021

- It was verified only on the data generated by the authors with some unrealistic artifact (cue), and the latest models for debiasing were not considered as baselines. Especially verifications on real datasets for debiasing such as corrupted cifar, biased bffhq, BAR, and MetaShift etc. are necessary.

- Or, I think the impact would be greater if it could be shown that CBFT can actually debias a pretrained model trained on a large scale. For example, it would be nice to resolve the bias of CLIP that the authors mentioned in the response to reviewer Y4FF, which will be plausible because fine tuning does not require large resources.


**Summary Of Ac-Reviewer Meeting:**

Further discussion confirmed that there was no misunderstanding from all reviewers, but the disagreement on accept/reject could not be narrowed down. Everyone was positive about the finding of the paper, but this AC and one reviewer leaned toward rejection because it was unclear what the finding actually meant in practice.